# Logarithmic Regret in Preference Learning via Optimistic PAC-Bayesian Particle Ensembles

## Abstract

The remarkable sample efficiency of preference-based reinforcement learning, which underpins the alignment of large language models with human feedback (RLHF), presents a significant theoretical puzzle. Existing analyses often rely on idealized assumptions, such as infinite-particle ensembles or exact, full-batch gradients, that are disconnected from the practical realities of deployed algorithms. This paper provides a statistically grounded abstraction of modern RLHF-style training pipelines. We introduce a unified optimistic PAC-Bayesian framework that distills the statistical essence of complex, multi-stage RLHF pipelines into a single, provably efficient online learning algorithm. Our central result is a high-probability regret bound of $\widetilde{\mathcal{O}}(d_{\mathrm{eluder}} \log T)$ for a rich, non-linear class of reward models, demonstrating when and why logarithmic regret is achievable using *finite* ensembles and noisy *stochastic gradient* updates under preference feedback. This unified theory provides an explanation for the sample efficiency of pairwise preference optimization, extends naturally to full Markov Decision Processes, and establishes a theoretical foundation for the empirical success of methods like RLHF.

## 1 Introduction

The alignment of large language models (LLMs) through preference-based learning has become a cornerstone of modern artificial intelligence, enabling the development of systems that are helpful, harmless, and attuned to human intent (Ouyang et al., 2022; Bai et al., 2022; Dong et al., 2024). A striking empirical observation in this domain is the profound *sample efficiency* of these alignment pipelines. Practitioners routinely steer billion-parameter models toward complex desired behaviors using on the order of only tens of thousands of pairwise human preferences (Rafailov et al., 2023; Christiano et al., 2017). This efficiency stands in stark contrast to the sheer dimensionality of the models and suggests that the correct theoretical target for regret should exhibit a near-logarithmic dependence on the number of interaction rounds, $T$. While classical online learning analyses for expressive function classes typically yield regret bounds of $\widetilde{\mathcal{O}}(\sqrt{T})$ (Russo & Van Roy, 2013; 2014), the empirical reality of RLHF motivates a much sharper theoretical goal. This leads to a pivotal open question: *Can we provide a rigorous theoretical explanation for the sample efficiency of practical preference-based alignment pipelines that yields sharp, near-logarithmic regret guarantees?*

The standard practical pipeline for Reinforcement Learning from Human Feedback (RLHF) is a complex, multi-stage process (Ouyang et al., 2022; Bai et al., 2022). It typically begins with Supervised Fine-Tuning (SFT) on a high-quality dataset, proceeds to the training of a separate reward model on collected human preference data, and culminates in policy optimization via an algorithm like PPO against that static reward model. This multi-stage pipeline, while empirically successful, presents a formidable challenge for unified theoretical analysis, as theoretical work often focuses on specific stages in isolation.

In this work, we move beyond analyzing the pipeline's components separately and instead propose a more fundamental theoretical model, the **Optimistic Langevin Ensemble (OLE)**, that captures the statistical core of preference-based learning in a single, cohesive online process. By analyzing this unified algorithm, we explain the sample efficiency of existing complex pipelines and provide a principled blueprint for a more theoretically grounded approach to alignment.

Bridging the empirical-theoretical divide requires that our unified model remains faithful to the realities of practical implementations. We identify four critical gaps[1] that must be addressed:

- **Gap 1: Mean-Field vs. Finite Ensembles.** Theoretical analyses often study a mean-field (infinite-particle) posterior flow for analytical tractability (Jordan et al., 1998; Sznitman, 2006), whereas practical implementations maintain a (often small) *finite* ensemble of reward models.

- **Gap 2: Exact vs. Stochastic Gradients.** Continuous-time or full-batch gradient derivations obscure the fact that all large-scale implementations rely on noisy mini-batch updates.

- **Gap 3: Continuous-Time vs. Discrete-Time Dynamics.** Mathematical tools like Wasserstein gradient flows offer an elegant continuous-time perspective (Ambrosio et al., 2008), but deployed algorithms operate in discrete time with a finite step size $\eta$.

- **Gap 4: Intractable vs. Tractable Uncertainty.** The principle of optimism requires an upper confidence bound on the true reward, but the exact Bayesian posterior uncertainty is intractable for deep neural networks. Practical algorithms rely on computationally feasible proxies, such as ensemble variance.

In this work, we develop an *optimistic PAC-Bayesian particle* framework for preference-based reinforcement learning that resolves these four gaps within our unified OLE model. Our framework is designed to be faithful to the algorithms used in practice while providing sharp, meaningful performance guarantees. We prove that such procedures attain a cumulative regret that scales as $\widetilde{\mathcal{O}}(d_{\mathrm{eluder}} \log T)$, where $d_{\mathrm{eluder}}$ is the eluder dimension of the function class (Russo & Van Roy, 2013; Li et al., 2022). Our analysis achieves this by coupling a PAC-Bayesian control of generalization (McAllester, 1999; Catoni, 2007) with concentration inequalities for stochastic dynamics (Freedman, 1975) and Wasserstein stability bounds for particle approximations (Fournier & Guillin, 2015), thereby addressing the four gaps within a single, cohesive theory.

**Positioning and Scope.** Our work is complementary to the important and emerging body of theory on *KL-regularized* bandits and RL, which has also achieved logarithmic regret guarantees but in the distinct setting of *numeric rewards* (Zhao et al., 2024; 2025b) for KL-regularized contextual bandits and MDPs under eluder-dimension assumptions. We, in contrast, focus on the more foundational problem of learning from *pairwise preference feedback*, which is the canonical setup for RLHF and DPO where a reward model is itself learned from human comparisons (Christiano et al., 2017; Bradley & Terry, 1952; Luce et al., 1959). Our contribution is an $\mathcal{O}(d_{\mathrm{eluder}} \log T)$ bound for *standard cumulative regret* in the pairwise-preference setting. Our analysis is algorithm-native, deriving guarantees directly from a PAC-Bayesian treatment of particle ensembles, rather than from the specific optimization landscape of a KL-regularized objective. Conceptually, our approach is related to optimism-in-the-face-of-uncertainty and to feel-good Thompson sampling (Zhang, 2022), but our setting, estimators, and guarantees are novel. A comprehensive survey and detailed comparisons appear in Appendix B.

Table 1: Our work achieves logarithmic regret for pairwise preference feedback with general function approximation in a framework that models practical algorithmic constraints. Detailed analysis on the differences in assumptions and problem settings can be found in Appendix B.1.

| Setting | Feedback Model | Key Assumptions | Regret (Leading Term) |
|---|---|---|---|
| **This work (OLE)** | **Pairwise Preference** | **Realizable + Eluder Dim.** | $\widetilde{\mathcal{O}}(d_{\mathrm{eluder}} \log T)$ |
| KL-Reg. Bandits (Zhao et al., 2025a) | Numeric Reward | Realizable + Eluder Dim. | $\widetilde{\mathcal{O}}(d \log T)$ |
| Preference RL (Wang et al., 2023) | Pairwise Preference | Realizable | $\widetilde{\mathcal{O}}(\sqrt{T})$ |
| Dueling Bandits (Yue et al., 2012) | Pairwise Preference | Tabular/Linear | $\widetilde{\mathcal{O}}(\log T)$ or $\widetilde{\mathcal{O}}(\sqrt{T})$ |
| Optimistic Bandits (Russo & Van Roy, 2014) | Numeric Reward | Realizable + Eluder Dim. | $\widetilde{\mathcal{O}}(d\sqrt{T})$ |

We summarize our main results for preference-based learning as follows.

- **Unified PAC-Bayesian Particle Analysis with Logarithmic Regret.** For preference-based contextual bandits, we analyze a practical algorithm using finite ensembles and mini-batch SGD. We prove that, with high probability, the cumulative regret is bounded by $\mathrm{Regret}(T) = \widetilde{\mathcal{O}}(d_{\mathrm{eluder}} \log T)$ + lower-order terms for discretization, finite ensembles, and mini-batching,

---

[1]More discussion on the four gaps in Appendix Section A.3.

where the leading term captures the statistical cost of exploration, and the lower-order terms explicitly quantify the practical algorithmic costs.

- **Optimistic Langevin Ensembles.** We introduce and analyze an optimistic Langevin-style ensemble update that provides exploration bonuses online and connects to standard preference optimization methods in the offline limit. Our analysis combines PAC-Bayesian inequalities with martingale concentration to provide non-asymptotic stability and concentration bounds.

- **Extension to Markov Decision Processes.** We extend our framework to preference-based RL with dynamics (e.g., discounted MDPs), obtaining analogous near-logarithmic regret guarantees. This complements results for numeric-reward MDPs (Zhao et al., 2025a) while operating in the more fundamental pairwise feedback regime.

- **Practical Implications.** Our bounds provide a direct theoretical explanation for the sample efficiency of methods like RLHF and DPO (Rafailov et al., 2023) and offer principled guidance for setting hyperparameters. We also show how parameter-efficient fine-tuning methods like LoRA (Hu et al., 2022) naturally lead to a small eluder dimension, connecting our theory to the practice of large-scale model alignment.

## 2 PROBLEM SETUP AND STRUCTURAL ASSUMPTIONS

This section formally establishes the mathematical foundation[2] for our analysis. We begin by defining the preference-based contextual bandit model and the notion of cumulative preference regret. We then introduce the key structural assumptions[3] on the underlying reward function class that enable efficient, low-regret learning.

### 2.1 THE PREFERENCE-BASED CONTEXTUAL BANDIT MODEL

We consider an online learning problem that unfolds over $T$ rounds. At each round $t \in \{1, \ldots, T\}$, the environment presents a context $x_t \in \mathcal{X}$. The learning agent then selects a pair of actions to be compared, typically to maximize information gain about the optimal action. The agent receives feedback in the form of a pairwise preference. This process models the core interaction loop in RLHF, where a context might be a user prompt and the actions are different model-generated responses (Ouyang et al., 2022; Christiano et al., 2017).

Underlying this preference feedback is a latent, unknown reward function $r^* : \mathcal{X} \times \mathcal{Y} \to \mathbb{R}$. This function represents the true, unobserved quality or utility of an action $y$ in a context $x$. The observed preferences are stochastic manifestations of this latent function. We model this relationship using the standard and widely adopted Bradley-Terry-Luce (BTL) model (Bradley & Terry, 1952; Luce et al., 1959). Given a pair of actions $(y_w, y_\ell)$, the probability that $y_w$ is preferred over $y_\ell$ (denoted $y_w \succ y_\ell$) in context $x$ is given by a logistic link function:

$$p(y_w \succ y_\ell \mid x) = \sigma\left(r^*(x, y_w) - r^*(x, y_\ell)\right). \tag{2.1}$$

Whenever we query a comparison between $(y_w, y_\ell)$ in context $x$, denote by $\text{feedback}_t \in \{0, 1\}$ the resulting binary preference at round $t$, taking value 1 when event $y_w \succ y_\ell$ occurs and 0 otherwise. The likelihood in Equation (2.1) is the BTL model, where $\sigma(z) = (1 + e^{-z})^{-1}$ is the sigmoid function. This model is central to many preference-based algorithms, including Direct Preference Optimization (Rafailov et al., 2023), and forms the basis of our likelihood-based objective.

The agent's goal is to learn a policy $\pi$ that, for any given context $x$, selects actions that have high latent reward $r^*(x, y)$. The performance of the agent is measured by the *cumulative preference regret*, which quantifies the total opportunity cost incurred over $T$ rounds. Let $y_t$ be the action selected by the agent's policy at round $t$ in context $x_t$, and let $y_t^* = \arg\max_{y \in \mathcal{Y}} r^*(x_t, y)$ be the optimal action for that context. The regret at round $t$ is the difference in expected reward between the optimal action and the chosen action. The cumulative regret over $T$ rounds is defined as:

$$\text{Regret}(T) = \sum_{t=1}^{T} \left(r^*(x_t, y_t^*) - r^*(x_t, y_t)\right). \tag{2.2}$$

---

[2] Frequently used symbols are summarized in Table 2 in Appendix Section A.

[3] An assumption checklist appears in Table 3 in Appendix Section A.

We will use Equation (2.2) as our formal notion of cumulative regret throughout the paper. The objective is to design an algorithm whose cumulative regret grows as slowly as possible with $T$. A logarithmic growth rate, $\text{Regret}(T) = \widetilde{\mathcal{O}}(\log T)$, is the theoretical ideal, indicating extremely efficient learning.

## 2.2 STRUCTURAL ASSUMPTIONS ON THE REWARD CLASS

To enable tractable learning from preference data alone, we impose a set of structural assumptions on the class of possible reward functions $\mathcal{R}$. These assumptions are standard in the theoretical analysis of learning with function approximation (Foster & Rakhlin, 2023) and are chosen to be as general as possible while still permitting strong performance guarantees.

**Assumption 2.1** (Realizability and bounded parameter space)**.** *We assume that the true latent reward function $r^*$ belongs to a known, parameterized function class $\mathcal{R} = \{r_\theta : \theta \in \Theta\}$, where each $r_\theta : \mathcal{X} \times \mathcal{Y} \to [0,1]$. The parameter space $\Theta \subset \mathbb{R}^d$ is a closed Euclidean ball $\Theta = \{\theta \in \mathbb{R}^d : \|\theta\| \leq B\}$ for some known radius $B < \infty$, and we assume the prior $\Pi_0$ and all subsequent posteriors $\Pi_t$ are supported on $\Theta$.*

This is a common starting point for theoretical analysis, allowing us to focus on the learning problem without the additional complication of model misspecification (Azar et al., 2024).

**Assumption 2.2** (Lipschitz Continuity)**.** *We assume that the reward function parameterization is smooth. Specifically, the function class is $L$-Lipschitz with respect to the parameters: for all $\theta, \theta' \in \Theta$ and all $(x, y)$, we have:*

$$|r_\theta(x,y) - r_{\theta'}(x,y)| \leq L\|\theta - \theta'\|_2. \tag{2.3}$$

This assumption is satisfied by many practical models, including neural networks with bounded weights and smooth activation functions. It is a crucial property that ensures that small changes in the parameter space lead to correspondingly small changes in the reward space, which is essential for generalization, optimization stability, and for relating parameter-space uncertainty to function-space uncertainty (Zhang, 2023).

**Assumption 2.3.** *This is the most critical assumption for enabling efficient exploration and achieving logarithmic regret. We assume that the function class $\mathcal{R}$ has a finite eluder dimension (Russo & Van Roy, 2013; 2014).*

**Eluder dimension.** We adopt the $\epsilon$-eluder dimension $d_{\text{eluder}}(\mathcal{R}, \epsilon)$ as the intrinsic complexity controlling regret in our analysis. For completeness, a concise definition together with its variance–information connection appears in Appendix D.2. Moreover, for LoRA-parameterized reward classes we establish sharp eluder control; see Proposition D.4 in Appendix D.3.

## 3 PAC-BAYESIAN GENERALIZATION AND WASSERSTEIN GRADIENT FLOW

This section connects PAC-Bayesian generalization objective to a Wasserstein gradient-flow (WGF) description of the learning dynamics. We (i) motivate a PAC-Bayes objective as the optimization target, (ii) introduce a smoothed/projected–KL device that yields a sharpened bound suitable for particle posteriors[4], and (iii) show that steepest descent of this objective in the 2-Wasserstein geometry yields a Langevin diffusion and the associated Fokker–Planck (continuity) equation. Full statements with constants and all proofs are deferred to Section C and Section E.

Let $S = \{z_i\}_{i=1}^m \overset{\text{i.i.d.}}{\sim} \mathcal{D}$, parameter space $\Theta \subseteq \mathbb{R}^d$, prior $\Pi$ on $\Theta$, posterior $\mu \in \mathcal{P}(\Theta)$, and per-example loss $\ell_\theta(z) \in [0,1]$ that is $L$-Lipschitz in $\theta$ for each $z$. We write $\hat{L}_S(\mu) := \frac{1}{m} \sum_{i=1}^m \mathbb{E}_{\theta \sim \mu} \ell_\theta(z_i)$ and $\text{Risk}\mu_{\mathcal{D}} := \mathbb{E}_{z \sim \mathcal{D}} \mathbb{E}_{\theta \sim \mu} \ell_\theta(z)$. For a Markov kernel $\mathsf{S}$ on $\Theta$, $\mathsf{S}_{\#}\mu$ denotes the push forward, and the *projected KL* is

$$D_{\text{KLS}}(\mu\|\Pi) := D_{\text{KL}}(\mathsf{S}_{\#}\mu \,\|\, \mathsf{S}_{\#}\Pi),$$

---

[4]Throughout this section we consider the *Gibbs posterior*, defined by $Q_\lambda(\mathrm{d}\theta) \propto \exp(-\lambda \hat{L}_m(\theta)) P(\mathrm{d}\theta)$, for a fixed prior over parameters $P$ and the empirical preference loss $\hat{L}_m(\theta)$. Note that Gibbs posterior differs the vanilla Bayesian posterior with respect to the environment's generative model.

which satisfies $D_{\mathrm{KLS}}(\mu\|\Pi) \leq D_{\mathrm{KL}}(\mu\|\Pi)$ by data processing (see Theorem C.3 and Section C).

A classical PAC-Bayes inequality for a posterior $\mu$ independent of $S$ reads

$$\mathrm{Risk}_{\mathcal{D}}(\mu) \leq \hat{L}_S(\mu) + \sqrt{\frac{D_{\mathrm{KL}}(\mu\|\Pi) + \ln\frac{2\sqrt{m}}{\delta}}{2m}}. \tag{3.1}$$

This suggests optimizing the right-hand side by trading empirical fit against complexity. Introducing an inverse-temperature parameter $\beta > 0$ yields the variational objective

$$J_{\mathrm{PAC}}(\mu) = \hat{L}_S(\mu) + \beta\, D_{\mathrm{KL}}(\mu\,\|\,\Pi), \tag{3.2}$$

which is the *free energy* associated with empirical risk and prior regularization.

**Per-example loss.** Each feedback example is denoted by $z$ (e.g., a bandit or preference observation), and we define the per-example loss as $\ell_\theta(z) := -\log p_\theta(z)$, the negative log-likelihood of $z$ under the parametric feedback model $p_\theta$. The PAC-Bayesian objective at time $t$ is then $J_t(\theta) := \mathbb{E}_{z\sim D_t}\big[\ell_\theta(z)\big] + \beta\big(\log\mu(\theta) - \log\Pi(\theta)\big)$, where $D_t$ is the dataset (or replay buffer) at time $t$. Our regret analysis only requires that $\ell_\theta(z)$ be bounded and Lipschitz in $\theta$ on $\Theta$, so any choice of loss satisfying these conditions yields the same asymptotic regret rate (the constants depend on the Lipschitz constant and range of $\ell_\theta$ but not on $T$ or $d_{\mathrm{eluder}}$).

### 3.1 SMOOTHED/PROJECTED–KL PAC-BAYES BOUND

We now state the smoothed/projected variant that will be used both for theory (to control finite-particle posteriors) and for algorithms (to motivate noise schedules). The definition is given here, while the full theorem and constants appear in Section C.

**Definition 3.1** (Projected/Smoothed KL). *For $\mu, \Pi \in \mathcal{P}(\Theta)$ and any smoothing kernel (confer Definition C.1)* $\mathsf{S}$*, define the* projected (smoothed) KL *by*

$$D_{\mathrm{KLS}}(\mu\|\Pi) := D_{\mathrm{KL}}\big(\mathsf{S}_{\#}\mu\,\|\,\mathsf{S}_{\#}\Pi\big).$$

*By data processing for $f$-divergences, $D_{\mathrm{KLS}}(\mu\|\Pi) \leq D_{\mathrm{KL}}(\mu\|\Pi)$ when the right-hand side is finite. For the Gaussian kernel, we write $D_{\mathrm{KLS}_h}(\mu\|\Pi) := D_{\mathrm{KL}}(\mathsf{S}_{h,\#}\mu\|\mathsf{S}_{h,\#}\Pi)$.*

**Theorem 3.2** (PAC-Bayes via smoothing). *Assume $\ell_\theta(z) \in [0,1]$ is $L$-Lipschitz in $\theta$. Let $\mu^N = \frac{1}{N}\sum_{i=1}^N \delta_{\theta_i}$ be an $N$-particle posterior and let $\mathsf{S}_h$ denote Gaussian smoothing with variance $h^2 I_d$. For any prior $\Pi$ independent of $S$ and any $h > 0$, with probability at least $1 - \delta$,*

$$\mathrm{Risk}{\mu^N}_{\mathcal{D}} \leq \mathrm{Risk}{\mu^N}_S + Lh\,\mathbb{E}\|Z\| + \sqrt{\frac{D_{\mathrm{KLS}_h}(\mu^N\|\Pi) + \ln(2m/\delta)}{2m}},$$

*where $Z \sim \mathcal{N}(0, I_d)$ so $\mathbb{E}\|Z\| \leq \sqrt{d}$. Moreover, if $\Pi = \mathcal{N}(\theta_0, \sigma_0^2 I_d)$ then*

$$D_{\mathrm{KLS}_h}(\mu^N\|\Pi) \leq \frac{1}{2N(\sigma_0^2+h^2)}\sum_{i=1}^N \|\theta_i - \theta_0\|^2 + \frac{d}{2}\,\phi\Big(\frac{h^2}{\sigma_0^2+h^2}\Big), \text{with } \phi(\rho) = \rho - 1 - \ln\rho.$$

.

### 3.2 OPTIMIZATION DYNAMICS AS A WASSERSTEIN GRADIENT FLOW

Interpreting Equation (3.2) as a free-energy functional on $\mathcal{P}(\Theta)$, the 2-Wasserstein gradient flow of $J_{\mathrm{PAC}}$ is the continuity equation

$$\partial_t \mu_t = \nabla_\theta\cdot\big(\mu_t\,\nabla_\theta V[\mu_t]\big), \tag{3.3}$$

where $V[\mu]$ is any $C^1$ potential whose gradient equals the Wasserstein gradient of $J_{\mathrm{PAC}}$ at $\mu$. Concretely, one may take

$$\nabla_\theta V[\mu](\theta) = \nabla_\theta\,\mathbb{E}_{z\sim S}\,\ell_\theta(z) + \beta\,\nabla_\theta\big(\log\mu(\theta) - \log\Pi(\theta)\big),$$

so that equation 3.3 coincides with the Fokker–Planck equation of the Langevin diffusion

$$d\theta(t) = -\nabla_\theta V[\mu_t]\big(\theta(t)\big)\,dt + \sqrt{2\beta}\,dW(t), \tag{3.4}$$

see, e.g., Jordan et al. (1998); Ambrosio et al. (2008); Villani (2008). Thus, gradient-based training of the free energy $J_{\text{PAC}}$ admits an exact continuum description as WGF.

A first-order time discretization of equation 3.4 (Euler–Maruyama) with step size $\eta > 0$ yields the particle update $\theta_{k+1} = \theta_k - \eta \nabla_\theta V[\mu_k](\theta_k) + \sqrt{2\eta\beta}\,\xi_k$, with $\xi_k \sim \mathcal{N}(0, I_d)$.

Replacing full gradients with mini-batch estimates recovers SGLD. This principled discretizations exposes and quantifies the approximation gaps that drive our regret analysis (precise bounds in Section E):**Finite-ensemble gap** (Monte Carlo drift error): $\widetilde{\mathcal{O}}\big(\sqrt{\sum_t v_t^2/N_t}\big)$. **Stochastic-gradient gap** (mini-batch noise): $\widetilde{\mathcal{O}}\big(\sqrt{\sum_t \sigma_t^2/B_t}\big)$. **Discretization gap** (time stepping): $\widetilde{\mathcal{O}}(\eta T)$. These terms map exactly onto the four sources of error isolated in the Introduction.

# 4 THE OPTIMISTIC LANGEVIN ENSEMBLE (OLE) ALGORITHM

This section translates the theoretical framework developed in the preceding sections into a concrete, self-contained algorithm for preference-based contextual bandits. The algorithm, which we call the **Optimistic Langevin Ensemble (OLE)**, instantiates the discretized Wasserstein gradient flow perspective. It maintains a finite ensemble of reward models, updates them using stochastic Langevin dynamics, and makes decisions using an optimistic selection rule based on ensemble statistics. The specific variant for online contextual bandits is termed Optimistic Thompson Sampling with Langevin Ensembles (O-TSLE).

The OLE algorithm operates in rounds. At each round $t$, it leverages its current posterior belief about the reward function, represented by an ensemble of particles, to optimistically select an action. It then observes the resulting preference feedback and updates its posterior belief using a Langevin step. A discussion on the computational cost of OLE is in Appendix G.1.Pseudo-code of additional variants are provided in Appendix G.2, such as for online contextual bandits and MDP scenarios.

---

**Algorithm 1:** Optimistic Langevin Ensemble (OLE): Generic Template

**Input:** Prior $\Pi_0$; step sizes $\{\eta_t\}$; ensemble sizes $\{N_t\}$; batch sizes $\{B_t\}$; optimism schedule $\{\kappa_t\}$

1 **for** $t = 1, 2, \ldots, T$ **do**
2    Observe context $x_t$;
    // Optimistic Selection
3    Compute ensemble mean $\hat{r}_t(x_t, y)$ and variance $\widehat{\text{Var}}_t(x_t, y)$ for all $y \in \mathcal{Y}$;
4    Construct optimistic index: $I_t(x_t, y) \leftarrow \hat{r}_t(x_t, y) + \kappa_t \sqrt{\widehat{\text{Var}}_t(x_t, y)}$;
5    Select action pair $(y_t^{(w)}, y_t^{(\ell)})$ based on maximizing information gain using $\{I_t(x_t, y)\}_{y \in \mathcal{Y}}$;
6    Receive preference feedback, forming data batch $\mathcal{D}_t$;
    // Posterior Update (SGLD)
7    Sample a mini-batch $B_t \subset D_t$ and compute the stochastic gradient
    $\widehat{\nabla}_t := \frac{1}{|B_t|} \sum_{z \in B_t} \nabla_\theta \ell_\theta(z) + \beta \nabla_\theta \big(\log \mu(\theta) - \log \Pi(\theta)\big)$;
8    Compute mini-batch gradient $\widehat{\nabla}_t$ of $J_{\text{PAC}}(\theta) = \hat{L}_{\mathcal{D}_t}(\theta) + \beta D_{\text{KL}}(\delta_\theta \| \Pi_{t-1})$;
9    **for** $i = 1, \ldots, N_t$ **do**
10       Draw Gaussian noise $\xi_t^{(i)} \sim \mathcal{N}(0, I)$;
11       $\theta_{t+1}^{(i)} \leftarrow \theta_t^{(i)} - \eta_t \widehat{\nabla}_t J_{\text{PAC}}(\theta_t^{(i)}) + \sqrt{2\eta_t\beta}\,\xi_t^{(i)}, \theta_{t+1}^{(i)} \leftarrow \text{Proj}_\Theta\big(\tilde{\theta}_{t+1}^{(i)}\big)$;

---

The core components of the algorithm are as follows:

- **Ensemble Maintenance:** The algorithm's belief about the true reward parameter $\theta^*$ is represented by an ensemble of $N_t$ particles, $\{\theta_t^{(i)}\}_{i=1}^{N_t}$. This ensemble serves as a Monte Carlo approximation of the posterior distribution $\mu_t$. At the start of learning ($t = 0$), these particles are drawn from a prior distribution $\Pi_0$.

- **Langevin Update Step:** This is the learning step of the algorithm. After receiving new preference data $\mathcal{D}_t$, each particle in the ensemble is updated using one step of Stochastic Gradient Langevin Dynamics (SGLD). The gradient is computed with respect to the PAC-Bayesian objective $J_{\text{PAC}}$

on a mini-batch of the new data. This update moves the particles towards regions of the parameter space that better explain the observed preferences, while the injected Gaussian noise ensures that the ensemble continues to represent a distribution and does not collapse to a single point.

- **Optimistic Selection Rule:** This is the exploration mechanism of the algorithm and the component that addresses the fourth implementation gap (intractable uncertainty). To make decisions that efficiently balance exploration and exploitation, the agent needs an upper confidence bound (UCB) on the true, unknown reward function $r^*$. Computing the exact Bayesian UCB is intractable for complex models. The OLE algorithm therefore uses a computationally feasible proxy based on the statistics of its particle ensemble. For each candidate action $y$ in the current context $x_t$, it computes an optimistic index:

$$I_t(x_t, y) = \hat{r}_t(x_t, y) + \kappa_t \cdot \sqrt{\widehat{\mathrm{Var}}_t(x_t, y)}. \tag{4.1}$$

The exploration bonus in Equation (4.1) follows the eluder-dimension view of exploration (Russo & Van Roy, 2013; 2014) and yields the desired logarithmic-regret scaling (Hazan et al., 2007).

- **Projection onto the bounded parameter space:** In the theoretical analysis we interpret the Langevin update as a projected SGLD step. Each unconstrained update is followed by the non-expansive Euclidean projection onto the ball $\Theta = \{\theta : \|\theta\| \leq B\}$. Since $\Pi_0$ is supported on $\Theta$, this ensures that all particles $\theta_t^{(i)}$ remain in $\Theta$ for all $t$, matching Assumption 2.1. In practice, this projection corresponds to weight clipping (or weight decay, softly) to the ball of radius $B$; if any iterate leaves $\Theta$, it is projected back before being used for action selection.

Here, $\hat{r}_t(x_t, y)$ is the mean reward predicted by the ensemble, serving as the best guess for the true reward. $\widehat{\mathrm{Var}}_t(x_t, y)$ is the variance of the reward predictions across the ensemble, which serves as a proxy for the posterior uncertainty about the reward of that action. The parameter $\kappa_t$ is an optimism coefficient that controls the weight given to this uncertainty, effectively determining how much the agent prioritizes exploration. The agent then selects a pair of actions to query for a preference based on these optimistic indices, typically choosing a pair that is expected to be most informative for resolving the current uncertainty. While the exact Bayesian posterior uncertainty is intractable for complex models, we will show in our analysis (Section 5) that the ensemble variance serves as a theoretically sound proxy. This is because of a fundamental duality between variance and information gain , which ensures that exploring regions of high ensemble variance leads to an efficient reduction of uncertainty about the true reward function, thereby enabling logarithmic regret.

**Remark 4.1** (Initialization of particles). *In the theoretical analysis we work with a fixed number of particles $N$ and initialize them i.i.d. from the prior $\Pi_0$ at $t = 1$, so $\theta_1^{(i)} \sim \Pi_0$ for all $i$. In practical variants where the number of particles $N_t$ is allowed to grow with $t$, we initialize any new particle with index $i > N_{t-1}$ from the current empirical posterior approximation $\Pi_{t-1}$ (i.e., by resampling from the existing particles). This implementation choice only affects constant factors in mixing and variance; the regret analysis is stated for the idealized setting with a fixed number of particles initialized from $\Pi_0$.*

## 5 Regret Analysis

This section presents the main theoretical result of the paper: a unified, high-probability regret bound for the Optimistic Langevin Ensemble (OLE) algorithm. The bound demonstrates that the algorithm achieves a cumulative regret that scales logarithmically with the time horizon $T$, plus explicit, sublinear terms that quantify the costs of the practical approximations corresponding to the "four gaps." This result provides a rigorous theoretical explanation for the remarkable sample efficiency of preference-based learning. Full proofs are in Appendix Section E.

Our main theorem bounds the cumulative preference regret of the OLE algorithm. It shows that the regret is controlled by the intrinsic complexity of the reward function class, as measured by the eluder dimension, and by the parameters governing the algorithmic approximations.

**Theorem 5.1.** *Let Assumptions 2.1 (Realizability), 2.2 (Lipschitz Continuity), and 2.3 (Finite Eluder Dimension) hold. For any $\delta \in (0, 1)$, consider the OLE algorithm run for $T$ rounds with step sizes $\{\eta_t\}$, ensemble sizes $\{N_t\}$, mini-batch sizes $\{B_t\}$, and an optimism schedule $\kappa_t = C_0\sqrt{\log(T/\delta)}$ for a suitable constant $C_0$. Let $v_t^2$ be an upper bound on the conditional variance of the Monte Carlo*

*estimate of the optimistic value, and let $\sigma_t^2$ be an upper bound on the conditional variance of the mini-batch gradient estimator. Then with probability at least $1 - \delta$, the cumulative regret satisfies:*

$$\text{Regret}(T) \leq \underbrace{C_1 \, d_{\text{eluder}} \log T}_{\text{Exploration Cost}} + C_2 \left( \underbrace{\sum_{t=1}^{T} \eta_t}_{\text{Discretization}} + \underbrace{\widetilde{\mathcal{O}} \left( \sqrt{\sum_{t=1}^{T} \frac{v_t^2}{N_t}} \right)}_{\text{Finite Ensemble}} + \underbrace{\widetilde{\mathcal{O}} \left( \sqrt{\sum_{t=1}^{T} \frac{\sigma_t^2}{B_t}} \right)}_{\text{Stochastic Gradient}} \right), \quad (5.1)$$

*where $C_1$ and $C_2$ are absolute constants. The eluder dimension $d_{\text{eluder}}$ is evaluated at a precision scale $\epsilon$ that decreases with $t$, such as $\epsilon_t = 1/(1 + t)$.*

**Remark 5.2** (On tightness of the leading term and Uniformity). *Up to polylogarithmic factors, the $\widetilde{\mathcal{O}}(d_{\text{eluder}} \log T)$ leading term in our regret bound matches known lower bounds and optimal algorithms for contextual bandits with rich (e.g., generalized linear) function classes, where the eluder dimension governs sample complexity (Russo & Van Roy, 2013; 2014). In particular, the $\log T$ factor is information-theoretically unavoidable even in parametric bandit settings with well-specified models (Hazan et al., 2007).*

*Our leading $\widetilde{\mathcal{O}}(d_{\text{eluder}} \log T)$ term is a* uniform *guarantee over all instances that satisfy our structural assumptions (realizability, boundedness, Lipschitz continuity, finite eluder dimension, and the Bradley–Terry–Luce preference model). Here $d_{\text{eluder}} = \dim_E(\mathcal{R}, T^{-1})$ is a complexity measure of the function class $\mathcal{R}$, and $T$ is the horizon; the bound does not expose explicit gap or margin parameters. The fast-rate behaviour comes from coupling two ingredients: (i) a variance–information lemma for the BTL model, which shows that the mutual information gained at round $t$ is at least a constant multiple of the squared prediction error; and (ii) an eluder-dimension bound on the cumulative squared widths (Lemma D.7).*

This bound provides a comprehensive picture of the algorithm's performance and completes the narrative arc of bridging the four gaps. Each term has a precise interpretation:

- **The Exploration Term:** $C_1 d_{\text{eluder}} \log T$. This is the leading-order term and represents the fundamental statistical cost of exploration. Its logarithmic dependence on the horizon $T$ is the key result, confirming that the algorithm learns extremely efficiently. The cost scales linearly with the eluder dimension $d_{\text{eluder}}$, which captures the intrinsic complexity of the learning problem. This term arises directly from the use of an optimistic exploration strategy.

- **The Discretization Error:** $\sum_{t=1}^{T} \eta_t$. This term quantifies the cost of Gap 3: approximating the continuous-time Wasserstein gradient flow with a discrete-time algorithm. It represents the cumulative bias from the Euler-Maruyama discretization. For a constant step size $\eta$, this error is $\widetilde{\mathcal{O}}(\eta T)$. However, as shown in the corollary below, this term can be made negligible by using a decreasing step size schedule.

- **The Finite-Ensemble Error:** $\widetilde{\mathcal{O}}(\sqrt{\sum_{t=1}^{T} v_t^2/N_t})$. This term quantifies the cost of Gap 1: approximating the true posterior distribution with a finite ensemble of $N_t$ particles. It represents the accumulated Monte Carlo estimation error. The term grows sub-linearly in $T$ and decreases as the ensemble size $N_t$ increases, explicitly characterizing the trade-off between computational cost and statistical accuracy.

- **The Stochastic Gradient Error:** $\widetilde{\mathcal{O}}(\sqrt{\sum_{t=1}^{T} \sigma_t^2/B_t})$. This term quantifies the cost of Gap 2: using noisy mini-batch gradients instead of exact full-batch gradients. It represents the accumulated noise from the stochastic optimization process. Like the ensemble error, it grows sub-linearly and decreases as the mini-batch size $B_t$ increases.

In the idealized limit where $\eta_t \to 0$, $N_t \to \infty$, and $B_t \to \infty$, all three lower-order terms vanish, and we are left with a purely logarithmic regret bound, $\text{Regret}(T) = \widetilde{\mathcal{O}}(d_{\text{eluder}} \log T)$. Our theorem provides the first analysis that makes this trade-off explicit for preference-based RL.

**Corollary 5.3.** *If the step sizes and resource allocation schedules are chosen such that $\sum_{t=1}^{T} \eta_t = \widetilde{\mathcal{O}}(1)$, $\sum_{t=1}^{T} v_t^2/N_t = \widetilde{\mathcal{O}}(1)$, and $\sum_{t=1}^{T} \sigma_t^2/B_t = \widetilde{\mathcal{O}}(1)$, then under the assumptions of Theorem 5.1, the cumulative regret is:*

$$\text{Regret}(T) = \widetilde{\mathcal{O}}\left(d_{\text{eluder}} \log T\right). \quad (5.2)$$

This corollary shows that by using standard schedules, such as a decreasing step size $\eta_t \propto 1/t$ and geometrically increasing ensemble and batch sizes, the approximation errors can be rendered into constant, lower-order terms, achieving the theoretical ideal.

**Remark 5.4.** *As discussed in Section 2, the eluder dimension can be related to the intrinsic dimensionality of the learning task. For models fine-tuned with low-rank adaptation (LoRA), the eluder dimension $d_{\mathrm{eluder}}$ is controlled not by the total number of parameters $d$, but by the much smaller intrinsic rank $d_*$ (Hu et al., 2022; Yang et al., 2023). Consequently, the regret bounds in Theorem 5.1 and Corollary 5.3 scale as $\widetilde{\mathcal{O}}(d_* \log T)$. This provides a direct and rigorous theoretical explanation for the empirical observation that parameter-efficient fine-tuning methods can achieve high sample efficiency even on massive models.*

## 6 EXTENSIONS TO MARKOV DECISION PROCESSES

To demonstrate the versatility and power of our theoretical framework, we extend the analysis from the contextual bandit setting to the more general and challenging setting of Markov Decision Processes (MDPs). This extension requires handling temporal dependencies, long-term credit assignment, and the propagation of uncertainty through Bellman updates. We show that our optimistic PAC-Bayesian ensemble approach can be naturally adapted to both finite-horizon and discounted MDPs, yielding analogous logarithmic regret guarantees. Proofs in Appendix Section F.

### 6.1 SETUP FOR PREFERENCE-BASED MDPS

A finite-horizon MDP is defined by a tuple $(\mathcal{S}, \mathcal{A}, H, P, r^*, \rho_0)$, where $\mathcal{S}$ is the state space, $\mathcal{A}$ is the action space, $H$ is the horizon, $P$ are the transition dynamics, $r^*$ is the latent reward function, and $\rho_0$ is the initial state distribution. In the preference-based RL setting, the agent does not observe the numeric rewards $r^*(s, a)$. Instead, it receives preference feedback, typically comparing entire trajectories or state-action pairs. The agent's objective is to learn a policy $\pi = \{\pi_h\}_{h=1}^H$ that maximizes the expected cumulative latent reward.

To enable value-based learning algorithms, we require an additional structural assumption beyond those for the bandit case.

**Assumption 6.1.** *We assume the function class for the action-value function (Q-function) is approximately closed under the Bellman optimality operator. That is, for any Q-function in our class, applying one step of Bellman backup results in a function that is still close to (or within) the class (Agarwal et al., 2023; Jin et al., 2021). This is a standard assumption in the theory of RL with function approximation, ensuring that the value functions produced during learning remain representable within our chosen model class.*

### 6.2 THE O-TDLE ALGORITHM FOR MDPS

We adapt our OLE algorithm to the MDP setting, resulting in a method we call Optimistic TD with Langevin Ensembles (O-TDLE). The core idea remains the same: maintain an ensemble of models to represent the posterior distribution and use optimistic exploration. The key difference is that the ensemble now represents the Q-function, and the updates are driven by temporal difference errors.

The O-TDLE algorithm (detailed in Algorithm 5 )proceeds in episodes. At each step $h$ within an episode, the agent is in state $s_h$. It uses its ensemble of Q-function models, $\{Q_{\theta^{(i)}}\}_{i=1}^N$, to compute an optimistic index for each action $a \in \mathcal{A}$:

$$I_h(s_h, a) = \hat{Q}_h(s_h, a) + \kappa_h \cdot \sqrt{\widehat{\mathrm{Var}}_h(Q(s_h, a))}, \tag{6.1}$$

where $\hat{Q}_h$ and $\widehat{\mathrm{Var}}_h$ are the mean and variance of the Q-value predictions across the ensemble. The agent then selects the action $a_h = \arg\max_{a \in \mathcal{A}} I_h(s_h, a)$. After executing the action and observing the next state $s_{h+1}$, the agent collects preference data (e.g., by comparing the executed trajectory segment to a reference—such as a SFT model). This data is then used to perform an SGLD update on the ensemble parameters $\{\theta^{(i)}\}$, using a loss derived from a Bellman-style TD error consistent with the preference feedback.

**Using the learned reward in MDPs.** In the MDP setting we never assume access to the environment's numeric single-step rewards. Instead, as in standard preference-based RL, we posit a latent

per-step reward function $r_{\theta^*}(x, a)$ such that preferences over finite trajectories are induced by their cumulative latent return. Given the observed pairwise preferences, our PAC-Bayesian update on $\theta$ produces a posterior distribution over reward models $r_\theta(\cdot, \cdot)$. At any time $t$, for a sampled parameter $\theta_t$ we can evaluate the *pseudo-reward* $\tilde{R}_t := r_{\theta_t}(x_t, A_t)$ on the visited state–action pair $(x_t, A_t)$. The TD targets in our MDP extension are defined in terms of these pseudo-rewards, e.g. $y_t = \tilde{R}_t + \gamma V_{\phi_t}(x_{t+1}) = r_{\theta_t}(x_t, A_t) + \gamma V_{\phi_t}(x_{t+1})$, for a value function $V_{\phi_t}$ with parameters $\phi_t$. Thus the algorithm is implementable from preference feedback: the environment is queried only for pairwise comparisons, which are used to update the posterior over $\theta$, and all numeric quantities required by TD are supplied by the learned reward model $r_\theta$.

## 6.3 REGRET ANALYSIS FOR MDPs

We prove that the O-TDLE algorithm achieves a logarithmic regret bound in the MDP setting. The bound now includes a polynomial dependence on the horizon $H$, which is expected as errors can propagate and compound over the steps of an episode.

**Theorem 6.2.** *Under Assumptions 2.1-2.3 and 6.1, the* O-TDLE *algorithm, run for $T$ episodes, achieves a cumulative regret that satisfies, with high probability:*

$$\text{Regret}(T) = \widetilde{\mathcal{O}}\left(H^2 \cdot d_{\text{eluder}} \cdot \log T\right) + \textit{lower-order approximation terms}. \tag{6.2}$$

*The lower-order terms for discretization, finite-ensemble, and stochastic gradient errors have a similar structure to the bandit case, now summed over all steps and episodes.*

**Remark 6.3** (On the $H$-dependence). *Our bound incurs an $H^2$ factor in the leading term, which is standard for episodic finite-horizon analyses under function approximation. Improving the $H$-dependence typically requires stronger structural assumptions (e.g., linear MDPs or Bellman completeness with additional mixing/realizability properties) or refined variance decompositions; see, e.g., Azar et al. (2024); Jin et al. (2021).*

Our proof for the MDP setting employs a powerful policy decomposition technique, inspired by recent advances in the analysis of KL-regularized RL with numeric rewards Zhao et al. (2025a). This technique allows us to reduce the multi-step credit assignment problem to a sequence of bandit-like analyses, to which our core optimistic exploration argument can be applied. The novelty of our approach lies in adapting this tool to the preference-based feedback setting and integrating it within our PAC-Bayesian particle ensemble framework. A similar analysis can be performed for the infinite-horizon discounted MDP setting, yielding a regret bound with a polynomial dependence on the effective horizon $(1 - \gamma)^{-1}$.

## 7 CONCLUSION, LIMITATIONS, AND FUTURE WORK

In this work, we developed a unified optimistic PAC-Bayesian framework for preference-based learning that closes several critical gaps between theory and practice. Our analysis provides the first theoretical explanation for the sample efficiency of modern alignment pipelines by establishing a near-logarithmic regret bound, $\widetilde{\mathcal{O}}(d_{\text{eluder}} \log T)$, that explicitly accounts for the algorithmic costs of using finite ensembles, stochastic gradients, and discrete-time updates. Our framework provides a firm theoretical foundation for the empirical success of methods like DPO (Rafailov et al., 2023) and connects the complexity of exploration to the intrinsic dimensionality of parameter-efficient fine-tuning (Aghajanyan et al., 2020; Hu et al., 2022).

**Limitations and Future works.** Our theoretical guarantees rely on standard but strong structural assumptions. The realizability assumption, which posits that the true reward function lies within the model class, is a significant idealization for complex models like LLMs, which are likely to be misspecified (Foster & Rakhlin, 2023). Similarly, our extension to MDPs requires Bellman completeness, a condition known to be restrictive for reinforcement learning with general function approximation (Agarwal et al., 2023; Golowich & Moitra, 2024; Wu et al., 2024). Finally, the decoupled structure of our regret bound opens the door to designing adaptive algorithms that can dynamically schedule computational resources, such as ensemble and mini-batch sizes, to optimally balance the statistical and computational trade-offs inherent in practical alignment. Our analysis is purely theoretical. A systematic empirical evaluation of OLE on preference-based RL benchmarks, as well as large-scale RLHF pipelines, is an important direction for future work.

## ETHICS STATEMENT

This work is theoretical, focusing on the algorithmic foundations of preference learning for the alignment of large language models. As with any alignment methodology, the practical application of our framework carries potential risks. These include *over-optimization* to the learned reward model, which may not perfectly capture nuanced human intent, and the potential for malicious *reward hacking*. We emphasize that our algorithms are designed for statistical and computational efficiency in optimizing a given preference model; they do not define the values inherent in that model. The collection and curation of the preference data that serves as the source of these values must be approached with care to respect privacy and mitigate the encoding and amplification of societal biases. Appropriate guardrails, diverse data sourcing, and multi-faceted evaluation of aligned models remain necessary to mitigate unintended consequences.

## THE USE OF LARGE LANGUAGE MODELS

In this work, the authors used generative AI tools (ChatGPT-5) to aid in and polish the writing of this paper. We use the following prompt to check the language section by section (including abstract): "Check the following statement, examine if the narrative is professional and understandable for broader audience in the area of machine learning community, and examine if the language meets native speaker standard. If not, generate feedback on how should I modify my narratives." All LLM-generated content was thoroughly reviewed and verified by the authors prior to inclusion. Research design, critical analyses, and all final decisions were carried out independently by the authors.

## REPRODUCIBILITY STATEMENT

This work is entirely theoretical. To ensure the reproducibility of our results, we provide complete and self-contained proofs for all theorems, propositions, and lemmas in the appendix. The appendix also contains detailed pseudocode for our proposed algorithms (Appendix G), a full discussion of the structural assumptions (Appendix A), and guidance on the hyperparameter schedules required to achieve the stated regret bounds. All cross-references within the document are hyperlinked for ease of navigation.

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

## APPENDIX CONTENTS

## A  NOTATION AND ADDITIONAL BACKGROUND

This appendix provides the complete theoretical underpinnings for the results presented in the main paper. We begin by establishing a unified notational system and providing a deeper discussion of the foundational concepts that motivate our work. This ensures the appendix is self-contained and accessible to readers with background in machine learning.

## A.1 Notation

We summarize the most frequently used symbols throughout the paper and this appendix in Table 2 for ease of reference. This consistent notation is crucial for maintaining clarity throughout the complex derivations that follow.

Table 2: Notation used throughout the paper and appendix.

| Symbol | Meaning |
|---|---|
| $\mathcal{X}, \mathcal{Y}$ | Context and candidate/output spaces |
| $\mathcal{S}, \mathcal{A}$ | State and action spaces (for MDPs) |
| $r^*(\cdot)$ | Ground-truth latent reward function, parameterized by $\theta^*$ |
| $\Theta$ | Parameter space for the reward models |
| $\mathcal{R} = \{r_\theta : \theta \in \Theta\}$ | Realizable reward function class |
| $\pi, \pi_t$ | Policy (at round $t$) |
| $\Pi_t, \mu_t$ | Posterior distribution over parameters $\theta$ at round $t$ |
| $\mu_t^N$ | Empirical measure of the $N$-particle ensemble at time $t$ |
| $(\mathcal{F}_t)_{t \geq 0}$ | Natural filtration (history) up to the end of round $t$ |
| $\text{feedback}_t$ | Preference feedback observed at round $t$ |
| $N_t, B_t, \eta_t$ | Ensemble size, mini-batch size, and step size at round $t$ |
| $w_t$ | Width of the confidence set $\mathcal{G}_t$ at the queried pair at round $t$ |
| $V_t$ | Posterior predictive variance of the queried logit difference at round $t$ |
| $v_t^2, \sigma_t^2$ | Conditional variance and sub-Gaussian noise proxy at round $t$ |
| $d_{\text{eluder}}$ | Eluder dimension of the reward function class $\mathcal{R}$ |
| $\gamma$ | Discount factor (for discounted MDPs) |
| $\beta$ | Inverse temperature in the PAC-Bayesian objective and SGLD updates |
| $\kappa_t$ | Optimism/bonus coefficient at round $t$ |
| $\text{Regret}(T)$ | Cumulative preference regret up to time $T$ |
| $W_2(\cdot, \cdot)$ | 2-Wasserstein distance between probability measures |

## A.2 Assumption Checklist

*How to read Table 3.* Each row states an assumption (or group of related assumptions), its informal meaning, and the main theorems/lemmas where it is used. This makes it easier to trace which structural conditions drive each part of the regret analysis.

*How to read Table 4.* We separate the bandit and finite-horizon MDP settings and indicate which assumptions are required in each case. This helps clarify which structural conditions are specific to the MDP extension (e.g., Bellman completeness) versus those already present in the bandit analysis.

## A.3 Detailed Discussion of Theoretical Gaps

The introduction highlighted four critical gaps between idealized theory and practical RLHF implementations. Here, we elaborate on why each gap presents a formidable theoretical challenge and how their interplay necessitates a unified analysis.

- **Gap 1 (Finite Ensembles vs. Mean-Field):** Many theoretical analyses of particle-based systems, especially those leveraging tools from optimal transport (Jordan et al., 1998; Ambrosio et al., 2008), operate in the mean-field limit where the number of particles $N \to \infty$. In this limit, the empirical distribution of particles converges to the solution of a deterministic partial differential equation (the Fokker-Planck equation), a phenomenon known as propagation of chaos (Sznitman, 2006). However, practical implementations use small, finite ensembles ($N$ is often less than 10). This introduces a non-trivial Monte Carlo sampling error at each step, as the interaction term in the particle dynamics depends on the empirical measure, not the true mean-field distribution. Our analysis must quantify this error and ensure it does not accumulate uncontrollably.

- **Gap 2 (Stochastic vs. Exact Gradients):** Large-scale model training is computationally infeasible without mini-batch stochastic gradients. While the noise introduced by mini-batching is

Table 3: Assumptions at a glance: informal summary and where they enter the analysis.

| Name | Informal content | Used in |
|---|---|---|
| Realizability and bounded $\Theta$ | Rewards lie in the model class $\mathcal{R}$; parameters lie in a bounded ball $\Theta$ | Theorem 5.1, Theorem D.8, Theorem D.1 |
| Lipschitz continuity | Reward model (and loss) are $L$-Lipschitz in $\theta$ on $\Theta$ | Theorem 3.2, Theorem D.8, Theorems D.10 and E.2, Theorem 5.1 |
| Finite eluder dimension | $\mathcal{R}$ has finite $\varepsilon$-eluder dimension | Theorem D.8, Theorem E.4, Theorem 5.1 |
| Langevin drift regularity | Drift of the mean-field Langevin SDE is Lipschitz and coercive | Theorem E.3 |
| Martingale / variance control | Martingale increments are sub-Gaussian with bounded conditional variances | Theorems D.10, D.11, E.1 and E.2 |

Table 4: Assumptions by setting.

| Setting | Active assumptions |
|---|---|
| Contextual bandits / preference bandits | Theorems 2.1 to 2.3 |
| Finite-horizon MDPs with preference feedback | Theorems 2.1, 2.2 and 6.1 |

zero-mean, its cumulative effect over $T$ rounds is a significant source of error. The variance of this noise depends on the batch size $B_t$ and the local curvature of the loss landscape. A rigorous analysis cannot simply assume gradients are exact; it must employ tools like martingale concentration inequalities to bound the accumulated deviation caused by this stochasticity.

- **Gap 3 (Discrete-Time vs. Continuous-Time):** The Wasserstein gradient flow perspective provides a powerful, continuous-time picture of the ideal optimization path. However, algorithms are implemented with a discrete step size $\eta_t$. The standard method for discretizing the underlying Langevin SDE is the Euler-Maruyama scheme. This introduces a discretization bias at each step, and the cumulative bias can grow linearly with $T$ if not carefully controlled, potentially overwhelming the desired logarithmic regret term. Our analysis must explicitly account for this weak error and show how to manage it with a proper step-size schedule.

- **Gap 4 (Tractable vs. Intractable Uncertainty):** The principle of optimism requires an upper confidence bound on the true reward function. For complex models like neural networks, the true Bayesian posterior variance is intractable to compute. Practical algorithms use the variance of predictions across the finite ensemble as a proxy for uncertainty. While intuitive, it is not a priori guaranteed that this ensemble variance is a valid upper bound on the true posterior uncertainty. A central part of our theoretical contribution is to formally justify this proxy and prove that it is sufficient to drive efficient exploration.

A crucial point is the interdependence of these gaps. The noise from stochastic gradients (Gap 2) can interact with and amplify the discretization error (Gap 3). The quality of the finite-ensemble approximation (Gap 1) directly determines the reliability of the uncertainty proxy used for exploration (Gap 4). A successful theory, therefore, cannot analyze these in isolation. Our unified framework is designed to bound the sum of these interacting error terms, demonstrating that their interplay does not lead to a catastrophic amplification of regret.

## A.4 CONTRIBUTIONS TO FORMAL RESULTS MAP

To provide a clear roadmap for the reader, Table 5 explicitly links the main contributions of this work to the formal theorems and proofs contained within this appendix. This table serves as a guide to verifying each of our central claims.

Table 5: Map of contributions to their formal statements and proofs in the appendix.

| Contribution | Formal statement (proof location) |
|---|---|
| Unified PAC–Bayesian particle theory | Theorem D.1 (App. D.1) |
| Unified regret bound for bandits | Statement Theorem 5.1 (App. E.2) |
| Finite-sample approximation error decomposition | Theorems E.1 to E.3 and equation E.16 (App. D.4) |
| Extension to finite-horizon MDPs with preferences | Theorem 6.2 and Section F.1 (App. F.2) |
| Extension to discounted MDPs with preferences | Section F.1 (App. F.3) |
| Eluder dimension for LoRA-style parametrizations | Theorem D.4 (App. D.3) |
| Fast-rate exploration term (logarithmic regret mechanism) | Theorem E.4 (App. E.2) |
| Algorithmic pseudocode (OLE / OTSLE / OTDLE) | Algorithms 2, 3 and 5 (App. G.2) |

## B  EXTENDED RELATED WORK

Our work connects to and builds upon several distinct but related lines of research in machine learning theory and practice.

**RLHF and Direct Preference Optimization.**  The modern paradigm of aligning LLMs was established by large-scale RLHF pipelines (Ouyang et al., 2022; Bai et al., 2022; Dong et al., 2024), which combine preference data collection, reward modeling, and policy optimization. More recent direct preference optimization methods, such as DPO and its variants (Rafailov et al., 2023; Meng et al., 2024), have streamlined this process and demonstrated strong empirical performance. Our work provides a foundational theoretical explanation for the remarkable sample efficiency observed in these practical systems, showing that near-logarithmic regret is achievable.

**Preference Learning, Dueling Bandits, and RL with Preferences.**  The problem of learning from comparative feedback has a long history, rooted in foundational statistical models like the Bradley-Terry-Luce model (Bradley & Terry, 1952; Luce et al., 1959; Thurstone, 2017). In the online setting, this problem is formalized as the *dueling bandits* problem, for which a rich body of literature provides sample complexity guarantees, typically achieving $\widetilde{\mathcal{O}}(\sqrt{T})$ regret in general settings and $\widetilde{\mathcal{O}}(\log T)$ in more restricted tabular or linear cases (Yue & Joachims, 2009; Yue et al., 2012). Extensions to reinforcement learning with preferences have been studied, but these analyses often yield sub-optimal $\widetilde{\mathcal{O}}(\sqrt{T})$ regret for general function classes (Wang et al., 2023; Pacchiano et al., 2021). Our work is the first to establish a near-logarithmic regret bound for preference-based RL with general non-linear function approximation.

**Relation to contextual dueling bandits.**  In the linear contextual dueling bandit setting of Bengs et al. (2022), the learner chooses a *pair* of actions at each round and receives a noisy comparison between them. They study weak/strong dueling regret, defined in terms of how often the chosen pair loses (or fails to win) against the best arm, and show a minimax $\Omega(d\sqrt{T})$ lower bound for this pairwise regret. In contrast, our setting is single-action selection with pairwise feedback: the learner chooses a single action $y_t$, may query preferences involving $y_t$, and we measure *standard* single-action cumulative regret $\mathrm{Regret}(T) = \sum_{t=1}^{T}\big(r^*(x_t, y^*(x_t)) - r^*(x_t, y_t)\big)$. Our $\widetilde{\mathcal{O}}(d_{\mathrm{eluder}}\log T)$ bound is a uniform fast-rate guarantee over all instances that satisfy our structural assumptions (realizability, boundedness, Lipschitz continuity, finite eluder dimension, and a BTL preference model), for this single-action regret. Since the action space and regret notion are different, the $\Omega(d\sqrt{T})$ dueling lower bound does not apply directly to our setting, and there is no contradiction between their result and our minimax bound.

**KL-Regularized Bandits and RL (Numeric Rewards).**  Our work is complementary to the important and emerging body of theory on KL-regularized bandits and RL, which has also achieved logarithmic regret guarantees but in the distinct setting of *numeric rewards* (Xiong et al., 2024; Zhao et al., 2024; 2025a;b) and often under additional structural assumptions like data coverage. While this parallel line of work provides deep insights into policy optimization given a numeric reward, our work addresses the more foundational problem of learning the reward function itself from *pairwise preference feedback*. This is the canonical setup for RLHF and DPO, where the reward model is the primary object to be learned from human comparisons. Our analysis is therefore algorithm-native, deriving guarantees directly from a PAC-Bayesian treatment of particle ensembles, rather than from the specific optimization landscape of a KL-regularized objective.

Specifically, we would like o highlight the difference between our result and the result developed byZhao et al. (2025a). Zhao et al. (2025a) has established $\widetilde{\mathcal{O}}(\log T)$ bounds for the *KL-regularized regret*, namely the suboptimality of the KL-regularized objective itself. Our results are complementary: we obtain a $\widetilde{\mathcal{O}}(d_{\mathrm{eluder}}\log T)$ bound for the *standard cumulative regret* $\mathrm{Regret}(T)$ in a pairwise-preference setting under realizability condition. We reference Zhao et al. (2025a) to highlight a shared eluder-dimension mechanism—in both cases, a sum-of-squares uncertainty term controls the cumulative suboptimality—rather than to equate their KL-regularized objective with our standard regret.

**PAC-Bayes, Optimism, and Thompson Sampling.** Our theoretical approach is built on the foundations of PAC-Bayesian learning theory, which provides powerful, high-probability generalization bounds for randomized predictors (McAllester, 1999; Catoni, 2007; Alquier, 2021; Guedj, 2019). Recent work has shown the power of PAC-Bayesian analysis for explaining generalization in deep learning (Lotfi et al., 2022; Haddouche et al., 2024). We combine these tools with the classical principle of optimism-in-the-face-of-uncertainty from the bandit literature (Hazan et al., 2007). The complexity of exploration in our framework is measured by the eluder dimension (Russo & Van Roy, 2013; 2014), a concept central to achieving logarithmic regret in benign regimes. Our optimistic posterior update mechanism is conceptually related to feel-good Thompson sampling (Zhang, 2022), but is tailored to the preference-based setting and analyzed via PAC-Bayesian tools.

**Particle Approximations and Optimal-Transport Tools.** To rigorously analyze the behavior of our finite-ensemble algorithm, we interpret its dynamics as a discretization of a Wasserstein gradient flow on the space of probability measures (Jordan et al., 1998). We control the approximation error introduced by the finite number of particles using tools from optimal transport theory and the study of empirical measures (Ambrosio et al., 2008; Villani, 2008; Fournier & Guillin, 2015; Sznitman, 2006). The analysis of the stochastic gradient and discretization errors is informed by the literature on the convergence of stochastic-gradient Langevin-type methods (Liu et al., 2023; Suzuki et al., 2023), allowing us to derive explicit, non-asymptotic lower-order terms in our regret bound.

In summary, prior analyses for preference-based learning typically achieve $\widetilde{\mathcal{O}}(\sqrt{T})$ regret for general function classes. In parallel, analyses of KL-regularized learning with numeric rewards have achieved $\widetilde{\mathcal{O}}(\log T)$ regret, sometimes under strong assumptions. Our work is the first to deliver a near-logarithmic regret bound for the fundamental problem of *pairwise preference feedback* within a framework that is faithful to the practical algorithms used in RLHF, thereby helps bridge the gap between theory and practice by providing logarithmic regret guarantees for preference-based RL in a framework that mirrors key aspects of RLHF-style pipelines (KL-regularized objectives, pairwise feedback, finite ensembles, and noisy stochastic gradients), while leaving a full empirical study for future work.

### B.1 COMPARISON AGAINST CLOSELY RELATED WORKS

**Comparison with the works in Table B.1.** Table B.1 collects the most closely related results and makes explicit that they differ along three axes: (i) the *setting and feedback model* (dueling vs. single–action, non–contextual vs. contextual, bandit vs. MDP, absolute rewards vs. preference feedback), (ii) the *objective / regret notion* (single–action regret, dueling regret, KL–regularized regret, Bayesian regret, or $\varepsilon$–optimality sample complexity), and (iii) the *assumptions* (realizability and bounded eluder dimension, stochastic transitivity, coverage conditions, etc.).

**Yue et al. (2012).** They study a non–contextual $K$–armed dueling bandit problem where the *action is a pair of arms* and the feedback is a noisy comparison between them. Regret is defined in terms of the probability that the unique best arm would win a duel against the chosen pair, with separate notions of strong and weak dueling regret. Under strong stochastic transitivity and a stochastic triangle inequality on pairwise win probabilities, they obtain expected regret $\mathbb{E}[R_T] = O\big(K\varepsilon_{1,2}^{-1}\log T\big)$ and prove a matching lower bound $\Omega\big(K\varepsilon^{-1}\log T\big)$. Our setting is contextual and optimizes *single–action regret* (the gap in latent reward between the chosen action and the optimal action), while only the *observations* are pairwise. We do not assume a finite $K$ or a total order over a fixed set of arms.

**Wang et al. (2023).** This work analyzes RLHF with preference feedback and gives *sample–complexity* guarantees for learning an $\varepsilon$–optimal policy (or a von Neumann winner) via reductions from preference–based RL to standard reward–based RL. Their bounds scale as $\tilde{O}\big(H^2 d_P|\Pi_{\exp}|^2\log|\mathcal{P}|/\varepsilon^2 + Hd_R|\Pi_{\exp}|/\varepsilon\big)$ episodes (plus a separate query–complexity term), and they do not study online regret as a function of $T$. By contrast, our focus is on online regret in our preference–based model; the induced sample complexity follows from standard online–to–batch conversion.

**Zhao et al. (2025).** Zhao et al. consider contextual bandits and MDPs with *absolute reward* feedback and optimize the *KL–regularized objective* $J(\pi) = \mathbb{E}[R^*(x,a)] - \eta^{-1}\mathrm{KL}(\pi(\cdot|x)\|\pi_{\mathrm{ref}}(\cdot|x))$. They

Table 6: Comparison of our results with closely related studies. The rows differ in setting/feedback, objective/regret notion, and assumptions. Our main contribution is a logarithmic–in–$T$ *single–action regret* bound under preference feedback and bounded eluder dimension.

| Work | Setting & feedback | Objective / regret notion | Key assumptions & guarantee (in $T$ or $\varepsilon$) |
|---|---|---|---|
| This paper | Contextual bandit / episodic; single action chosen, *pairwise* (preference) observations. | Standard cumulative *single–action regret* (latent reward gap between chosen and optimal action). | Realizability of reward/preference in a function class with bounded eluder dimension; mild curvature/low–noise condition on the link. Regret $\tilde{O}(d_{\mathrm{E}} \log T)$ in $T$. |
| Yue et al. (2012) | Non–contextual $K$–armed dueling bandits; action is a *pair* of arms with noisy comparison feedback. | Strong/weak *dueling regret* w.r.t. win probability of the best arm vs. chosen pair. | Strong stochastic transitivity and stochastic triangle inequality on pairwise win probabilities. $\mathbb{E}[R_T] = O\big(K\varepsilon_{1,2}^{-1} \log T\big)$ and lower bound $\Omega\big(K\varepsilon^{-1} \log T\big)$. |
| Wang et al. (2023) | General RLHF (MDPs with $H > 1$); trajectory or $(s,a)$–level preference feedback. | *Sample complexity* to obtain an $\varepsilon$–optimal policy (or von Neumann winner); no explicit regret in $T$. | Realizability of reward or preference classes; Bellman– or generalized–eluder dimension bounds. Sample complexity $\tilde{O}\big(H^2 d_P |\Pi_{\exp}|^2 \log |\mathcal{P}|/\varepsilon^2 + H d_R |\Pi_{\exp}|/\varepsilon\big)$ (plus query complexity). |
| Zhao et al. (2025) | Contextual bandits and MDPs with *absolute* rewards and known reference policy. | *KL–regularized* regret in $J(\pi) = \mathbb{E}[R^*] - \eta^{-1}\mathrm{KL}(\pi\|\pi_{\mathrm{ref}})$. | Realizability for reward class with bounded eluder dimension. Regret $O\big(\eta\, d_{\mathcal{R}} \log(N_{\mathcal{R}} T)\big)$ for bandits, and analogous bound with $H$–dependence for MDPs. |
| Russo & Van Roy (2014) | Stochastic bandits (including contextual) with general function approximation; absolute rewards. | *Bayesian* cumulative regret under a prior; no preferences or KL–regularization. | Eluder dimension and Kolmogorov dimension of reward class. $\mathrm{BayesRegret}(T) \leq \tilde{O}\big(\sigma\sqrt{d_{\mathrm{E}}(F, 1/T)\,T}\big)$, giving $\tilde{O}(d\sqrt{T})$ for linear models (not $O(d \log T)$). |

prove that their KL–UCB and KL–LSVI–UCB algorithms achieve $O\big(\eta\, d_{\mathcal{R}} \log(N_{\mathcal{R}} T)\big)$ regret in this KL–regularized objective (with additional $H$–dependence in MDPs). We cite this work because it also uses eluder–based *sum–of–squares* arguments to obtain logarithmic dependence on $T$, but the objective differs: our main theorems are stated for standard cumulative regret in our preference–based model.

**Russo & Van Roy (2014).** Russo and Van Roy introduce the eluder dimension and analyze posterior sampling (Thompson sampling) for general stochastic bandit models. Their main results are *Bayesian regret* bounds of the form $\mathrm{BayesRegret}(T) \leq \tilde{O}\big(\sigma\sqrt{d_{\mathrm{E}}(F, 1/T)\,T}\big)$, which specialize to $\tilde{O}(d\sqrt{T})$ (up to logarithms) for linear models. We only use their notion of eluder dimension as a complexity measure; our logarithmic dependence on $T$ arises from a different squared–gap decomposition that is specific to our model. In the revised version, we will correct our earlier informal summary from $O(d \log T)$ to $\tilde{O}(d\sqrt{T})$.

## C  SMOOTHED/PROJECTED–KL PAC-BAYES AND WGF: FULL STATEMENTS AND PROOFS

This section collects the technical results that underlie the smoothed/projected–KL PAC-Bayes bound and its Wasserstein gradient-flow interpretation used in the main text. We organize the material as follows:

(i) In Theorems C.1 and C.3 we formalize the smoothing kernel $\mathsf{S}$ and the induced projected KL divergence $D_{\mathrm{KLS}}$, and we record the basic properties needed later (chiefly the data-processing inequality).

(ii) Theorem C.4 states and proves the full smoothed/projected–KL PAC-Bayes generalization bound, including the Gaussian specialization that we plug into the regret analysis.

(iii) In the final subsection we spell out the Wasserstein gradient-flow calculus for the PAC-Bayesian free-energy functional $J_{\mathrm{PAC}}$, and we show how it gives the Fokker–Planck equation tracked by our idealized particle dynamics.

Purely measure-theoretic details and the episode budget/scheduling lemmas used in the regret proof are deferred to Section D and Section E.

## C.1 PROJECTED–KL SMOOTHING AND BASIC PROPERTIES

We recall the projected divergence used in the main text.

**Definition C.1** (Smoothing kernel and pushforward). *Let $(\Theta, \mathcal{B})$ be a measurable parameter space. A smoothing kernel is a Markov kernel $\mathsf{S} : \Theta \times \mathcal{B} \to [0, 1]$, i.e., for each $\theta \in \Theta$, $\mathsf{S}(\theta, \cdot)$ is a probability measure and for each $A \in \mathcal{B}$, $\theta \mapsto \mathsf{S}(\theta, A)$ is measurable. For a probability measure $\mu \in \mathcal{P}(\Theta)$, its pushforward by $\mathsf{S}$ is*

$$(\mathsf{S}_{\#}\mu)(A) := \int_{\Theta} \mathsf{S}(\theta, A)\, \mu(d\theta), \qquad A \in \mathcal{B}.$$

*When $\Theta = \mathbb{R}^d$ and $h > 0$, the Gaussian smoothing kernel is $\mathsf{S}_h(\theta, \cdot) := \mathcal{N}(\theta, h^2 I_d)$, in which case $\mathsf{S}_{h,\#}\mu = \mu * \mathcal{N}(0, h^2 I_d)$ is the usual Gaussian convolution. We write $\mathsf{S}_h := \mathsf{S}_h$ for brevity.*

**Remark C.2** (Interpretation of smoothing and projected KL). *Intuitively, the kernel $\mathsf{S}(\theta, \cdot)$ replaces a deterministic parameter $\theta$ by a small cloud of nearby parameters. Sampling $\theta \sim \mu$ and then $\tilde{\theta} \sim \mathsf{S}(\theta, \cdot)$ produces a random "smoothed parameter" $\tilde{\theta}$ with law $\mathsf{S}_{\#}\mu$. The projected KL divergence*

$$D_{\mathrm{KLS}}(\mu\|\Pi) = D_{\mathrm{KL}}\big(\mathsf{S}_{\#}\mu \,\|\, \mathsf{S}_{\#}\Pi\big)$$

*therefore compares $\mu$ and $\Pi$ only through their smoothed versions. By the data-processing inequality we always have $D_{\mathrm{KLS}}(\mu\|\Pi) \le D_{\mathrm{KL}}(\mu\|\Pi)$ whenever the latter is finite, so $D_{\mathrm{KLS}}$ is a more forgiving complexity term. This is precisely the divergence that appears in the smoothed PAC-Bayes bound of Theorem C.4.*

**Definition C.3** (Projected/Smoothed KL). *For $\mu, \Pi \in \mathcal{P}(\Theta)$ and any smoothing kernel $\mathsf{S}$, define the projected (smoothed) KL by*

$$D_{\mathrm{KLS}}(\mu\|\Pi) := D_{\mathrm{KL}}\big(\mathsf{S}_{\#}\mu \,\|\, \mathsf{S}_{\#}\Pi\big).$$

*By data processing for $f$-divergences, $D_{\mathrm{KLS}}(\mu\|\Pi) \le D_{\mathrm{KL}}(\mu\|\Pi)$ when the right-hand side is finite. For the Gaussian kernel of Definition C.1, we write $D_{\mathrm{KLS}_h}(\mu\|\Pi) := D_{\mathrm{KL}}(\mathsf{S}_{h,\#}\mu\|\mathsf{S}_{h,\#}\Pi)$.*

**Risk notation (for convenience).** For a distribution $\mu$ over parameters, a dataset $S = (z_1, \ldots, z_m)$ of size $m$, and a data distribution $\mathcal{D}$ over examples $z$, we recall the randomized predictor risks

$$\mathrm{Risk}\mu_S := \frac{1}{m} \sum_{i=1}^{m} \mathbb{E}_{\theta \sim \mu}\, \ell_\theta(z_i), \qquad \mathrm{Risk}\mu_{\mathcal{D}} := \mathbb{E}_{z \sim \mathcal{D}} \mathbb{E}_{\theta \sim \mu}\, \ell_\theta(z).$$

These coincide with the empirical and population risks used in the main text.

## C.2 SMOOTHED/PROJECTED–KL PAC-BAYES BOUND: FULL STATEMENT AND PROOF

We now give the full version of Theorem 3.2 including constants and a convenient specialization for Gaussian priors.

**Theorem C.4** (PAC-Bayes via smoothing; full). *Assume $\ell_\theta(z) \in [0, 1]$ is $L$-Lipschitz in $\theta$ for each $z$. Let $S = \{z_i\}_{i=1}^m \overset{i.i.d.}{\sim} \mathcal{D}$, and let $\mu^N = \frac{1}{N} \sum_{i=1}^{N} \delta_{\theta_i}$ be any $N$-particle posterior (possibly data-dependent). For any prior $\Pi$ independent of $S$, any $h > 0$, and any $\delta \in (0, 1)$, with probability at least $1 - \delta$ over $S$,*

$$\mathrm{Risk}\mu^N{}_{\mathcal{D}} \le \mathrm{Risk}\mu^N{}_S + Lh\,\mathbb{E}\|Z\| + \sqrt{\frac{D_{\mathrm{KLS}_h}(\mu^N\|\Pi) + \ln(2m/\delta)}{2m}},$$

*where $Z \sim \mathcal{N}(0, I_d)$ so that $\mathbb{E}\|Z\| \leq \sqrt{d}$. Moreover, if $\Pi = \mathcal{N}(\theta_0, \sigma_0^2 I_d)$, then*

$$D_{\mathrm{KLS}_h}(\mu^N \| \Pi) \ \leq \ \frac{1}{2N(\sigma_0^2 + h^2)} \sum_{i=1}^{N} \|\theta_i - \theta_0\|^2 \ + \ \frac{d}{2} \phi\Big(\frac{h^2}{\sigma_0^2 + h^2}\Big),$$

*with $\phi(\rho) = \rho - 1 - \ln \rho$.*

**Remark C.5** (Connection to the main regret bound)**.** *In the regret analysis of Section E.1 we would apply Theorem C.4 with $\mu^N$ equal to the empirical measure of the $N$ particles at the beginning of an episode, and with the smoothing scale $h$ chosen according to the schedule specified in Section E. The Gaussian specialization controls the complexity term $D_{\mathrm{KLS}_h}(\mu^N \| \Pi)$ by the squared distance of the particles from the Gaussian prior mean $\theta_0$:*

$$D_{\mathrm{KLS}_h}(\mu^N \| \Pi) \ \lesssim \ \frac{1}{N(\sigma_0^2 + h^2)} \sum_{i=1}^{N} \|\theta_i - \theta_0\|^2 \ + \ d,$$

*which is in turn bounded along the dynamics using the stability and step-size conditions proved in Section E. This is the only place where the explicit form of $D_{\mathrm{KLS}_h}$ for Gaussian priors enters the regret bound.*

*Proof.* Apply a standard PAC-Bayes bound for bounded losses (e.g., *empirical Bernstein*/McAllester-style) to the *smoothed* posterior $\mathsf{S}_{h,\#}\mu^N$ and prior $\mathsf{S}_{h,\#}\Pi$:

$$\mathrm{Risk}\mathsf{S}_{h,\#}\mu^N{}_{\mathcal{D}} \ \leq \ \mathrm{Risk}\mathsf{S}_{h,\#}\mu^N{}_S \ + \ \sqrt{\frac{D_{\mathrm{KL}}\big(\mathsf{S}_{h,\#}\mu^N \| \mathsf{S}_{h,\#}\Pi\big) + \ln(2m/\delta)}{2m}}.$$

Lipschitzness and Gaussian smoothing yield the bias control $\mathrm{Risk}\mu^N{}_{\mathcal{D}} \ \leq \ \mathrm{Risk}\mathsf{S}_{h,\#}\mu^N{}_{\mathcal{D}} + Lh\,\mathbb{E}\|Z\|$ and $\mathrm{Risk}\mathsf{S}_{h,\#}\mu^N{}_S \leq \mathrm{Risk}\mu^N{}_S + Lh\,\mathbb{E}\|Z\|$, whence

$$\mathrm{Risk}\mu^N{}_{\mathcal{D}} \ \leq \ \mathrm{Risk}\mu^N{}_S \ + \ Lh\,\mathbb{E}\|Z\| \ + \ \sqrt{\frac{D_{\mathrm{KLS}_h}(\mu^N \| \Pi) + \ln(2m/\delta)}{2m}},$$

using $D_{\mathrm{KLS}_h}(\mu^N \| \Pi) = D_{\mathrm{KL}}(\mathsf{S}_{h,\#}\mu^N \| \mathsf{S}_{h,\#}\Pi)$ (definition) and $\mathbb{E}\|Z\| \leq \sqrt{d}$. For the Gaussian-prior specialization, compute the KL between Gaussians:

$$D_{\mathrm{KL}}\Big(\mathcal{N}(\theta_i, h^2 I_d) \,\Big\|\, \mathcal{N}(\theta_0, (\sigma_0^2 + h^2) I_d)\Big) \ = \ \frac{\|\theta_i - \theta_0\|^2}{2(\sigma_0^2 + h^2)} \ + \ \frac{d}{2} \phi\Big(\frac{h^2}{\sigma_0^2 + h^2}\Big),$$

and average over $i = 1, \ldots, N$. This proves the claim. $\qquad\square$

**Gaussian prior specialization.**   If $\Pi = \mathcal{N}(\theta_0, \sigma_0^2 I_d)$ and $\mu^N = \frac{1}{N} \sum_{i=1}^{N} \delta_{\theta_i}$, then

$$D_{\mathrm{KLS}_h}(\mu^N \| \Pi) \ = \ \frac{1}{N} \sum_{i=1}^{N} D_{\mathrm{KL}}\Big(\mathcal{N}(\theta_i, h^2 I_d) \,\Big\|\, \mathcal{N}(\theta_0, (\sigma_0^2 + h^2) I_d)\Big)$$

with

$$D_{\mathrm{KL}}\Big(\mathcal{N}(\theta_i, h^2 I_d) \,\Big\|\, \mathcal{N}(\theta_0, (\sigma_0^2 + h^2) I_d)\Big) = \frac{\|\theta_i - \theta_0\|^2}{2(\sigma_0^2 + h^2)} + \frac{d}{2} \phi\Big(\frac{h^2}{\sigma_0^2 + h^2}\Big), \quad \phi(\rho) = \rho - 1 - \ln \rho.$$

## C.3   WASSERSTEIN GRADIENT-FLOW CALCULUS USED IN THE MAIN TEXT

For completeness we record the standard Wasserstein gradient-flow formulation of the PAC-Bayesian objective used in Section 3. Recall that the PAC-Bayesian free-energy functional is

$$J_{\mathrm{PAC}}(\mu) \ := \ \hat{L}_S(\mu) + \beta\, D_{\mathrm{KL}}(\mu \| \Pi) \ = \ \int_{\Theta} \Big( \mathbb{E}_{z \sim S}\big[\ell_\theta(z)\big] \Big) \mu(d\theta) \ + \ \beta \int_{\Theta} \log\Big(\frac{\mu(\theta)}{\Pi(\theta)}\Big) \mu(d\theta),$$

$$\tag{C.1}$$

where $\mu$ is a probability measure on $\Theta$ with density (still denoted by $\mu$) with respect to Lebesgue measure, and $\Pi$ is a fixed prior with a strictly positive density on the support of $\mu$.

Standard results in optimal transport (see, e.g., Jordan et al. (1998); Ambrosio et al. (2008); Villani (2008)) imply that the 2-Wasserstein gradient flow of $J_{\mathrm{PAC}}$ is governed by the continuity equation

$$\partial_t \mu_t(\theta) \;=\; \nabla_\theta \cdot \Big( \mu_t(\theta) \, \nabla_\theta \frac{\delta J_{\mathrm{PAC}}}{\delta \mu}(\theta) \Big), \tag{C.2}$$

where the first variation of $J_{\mathrm{PAC}}$ is given by

$$\frac{\delta J_{\mathrm{PAC}}}{\delta \mu}(\theta) \;=\; \mathbb{E}_{z \sim S}\big[\ell_\theta(z)\big] \;+\; \beta\big(\log \mu(\theta) - \log \Pi(\theta)\big) \;+\; c_t. \tag{C.3}$$

Here $c_t$ is an arbitrary time-dependent constant (arising from the normalization of $\mu_t$) whose gradient is zero and hence does not affect the flow in equation C.2.

Expanding the divergence in equation C.2 using equation C.3 yields

$$\partial_t \mu_t \;=\; \nabla_\theta \cdot \Big( \mu_t \nabla_\theta \mathbb{E}_{z \sim S}\big[\ell_\theta(z)\big] \Big) \;+\; \beta \, \Delta_\theta \mu_t \;-\; \beta \, \nabla_\theta \cdot \Big( \mu_t \nabla_\theta \log \Pi(\theta) \Big),$$

which is exactly the Fokker–Planck equation Equation (3.3) associated with the Langevin diffusion targeting the Gibbs posterior with density proportional to $\exp(-\mathbb{E}_{z \sim S}[\ell_\theta(z)])\Pi(\theta)$. This is the correspondence used in the main text to connect the population-level idealized dynamics to the particle algorithm.

**Where to find the end-to-end regret analysis.** The budget allocation across episodes/iterations and the root-time Monte Carlo accumulation lemmas used for our final regret bounds appear in Section D and Section E. This avoids duplicating those results here while keeping this appendix focused on the PAC-Bayes smoothing and the WGF calculus.

## D   TECHNICAL LEMMAS AND AUXILIARY RESULTS

This section gathers technical lemmas (variance–information coupling, discretization, stochastic gradients, Monte Carlo concentration) used by Section E.

We start by recalling the PAC-Bayesian objective and its connection to the Wasserstein gradient flow, and then proceed to rigorously analyze each source of approximation error.

### D.1   PAC-BAYESIAN GENERALIZATION AND THE LEARNING OBJECTIVE

The PAC-Bayesian framework provides high-probability guarantees on the population loss of a randomized predictor $Q$ in terms of its empirical loss and its divergence to a fixed, data-independent prior $P$. Let $S = \{z_i\}_{i=1}^m$ be drawn i.i.d. from $\mathcal{D}$, define the population loss

$$L(\theta) \;:=\; \mathbb{E}_{z \sim \mathcal{D}}\big[\ell_\theta(z)\big],$$

and the empirical loss

$$\hat{L}_m(\theta) \;:=\; \frac{1}{m} \sum_{i=1}^m \ell_\theta(z_i) \quad \text{on the dataset } S.$$

A standard PAC-Bayes bound (see, e.g., Catoni (2007)) states that for any prior distribution $P$ on $\Theta$, any $\delta \in (0,1)$, and any (possibly data-dependent) posterior $Q$ on $\Theta$, with probability at least $1 - \delta$ over the draw of $S$,

$$\mathbb{E}_{\theta \sim Q}\big[L(\theta)\big] \;\le\; \mathbb{E}_{\theta \sim Q}\big[\hat{L}_m(\theta)\big] \;+\; \sqrt{\frac{D_{\mathrm{KL}}(Q\|P) + \ln(m/\delta)}{2m}}. \tag{D.1}$$

In words, the true risk of $Q$ is controlled by its empirical risk plus a complexity term depending on how far $Q$ deviates from the prior $P$.

It is convenient to collect the empirical-risk and complexity terms into the *PAC-Bayesian free-energy functional*

$$J_{\mathrm{PAC}}(\mu) \;:=\; \hat{L}_S(\mu) \;+\; \beta \, D_{\mathrm{KL}}(\mu\|P), \tag{D.2}$$

where $\hat{L}_S(\mu) := \mathbb{E}_{\theta \sim \mu}\big[\hat{L}_m(\theta)\big]$ and $\beta > 0$ plays the role of an inverse temperature. For an appropriate choice of $\beta$, minimizing the right-hand side of equation D.1 over $Q$ is equivalent (up to

additive constants independent of $Q$) to minimizing $J_{\mathrm{PAC}}$, and the unique minimizer of $J_{\mathrm{PAC}}$ is the Gibbs posterior

$$Q_\lambda(\mathrm{d}\theta) \;\propto\; \exp\!\big(-\lambda\,\hat{L}_m(\theta)\big)\,P(\mathrm{d}\theta), \qquad \lambda = 1/\beta.$$

The Langevin update step in our OLE algorithm is therefore a noisy gradient step on the functional $J_{\mathrm{PAC}}$, with the measure $\mu$ represented in practice by the empirical distribution of the particle ensemble.

In the smoothed PAC-Bayesian and Wasserstein gradient-flow analysis below we denote the prior by $\Pi$ and write $(\Pi_t)_{t\geq 0}$ for the time-indexed posteriors generated by the idealized dynamics. In our setting we simply take $\Pi = P$ and use the notation $\Pi_0$ for the initial prior and $\Pi_t$ for its evolution over time.

**Theorem D.1.** *The posterior distribution $\Pi_t$ maintained by the idealized (continuous-time, infinite-particle) Langevin dynamics minimizes the PAC-Bayesian functional $J_{\mathrm{PAC}}(\mu)$ over the space of probability measures. The finite-ensemble, discrete-time, stochastic-gradient implementation approximates this ideal posterior, and its generalization error is controlled by the sum of the PAC-Bayesian objective and the approximation error terms.*

*Proof.* The proof follows from the variational characterization of the Fokker-Planck equation as the Wasserstein gradient flow of the free energy functional, which in our case is $J_{\mathrm{PAC}}(\mu)$ (Jordan et al., 1998). The practical algorithm is a numerical approximation of this flow, and its deviation from the ideal posterior is bounded by the lemmas in Section D.4. $\square$

**Informal interpretation of Theorem D.1** Informally, Theorem D.1 says that the empirical particle posterior produced by OLE tracks the Wasserstein gradient flow of the PAC-Bayesian objective $J_{\mathrm{PAC}}$ up to controlled approximation errors. As a consequence, any decrease of $J_{\mathrm{PAC}}$ along the idealized continuous-time dynamics is mirrored (up to the bounds established in Section D.4) by the finite-particle algorithm used in our implementation. This is the bridge between the PAC-Bayesian generalization theory and the actual learning dynamics analyzed in the regret bound.

## D.2 ELUDER DIMENSION AND THE VARIANCE-INFORMATION BOUND

The key to bounding the exploration cost is the eluder dimension (Russo & Van Roy, 2013; 2014).

Throughout this section, we write $(\mathcal{F}_t)_{t\geq 0}$ for the natural filtration generated by all randomness up to the end of round $t$ (contexts $x_s$, actions $y_s$, preference observations $\mathrm{feedback}_s$, and the internal randomness of the algorithm for $s \leq t$). We denote by $\mathrm{feedback}_t \in \{0,1\}$ the binary preference feedback observed at round $t$, with $\mathrm{feedback}_t = 1$ corresponding to the event that the "winning" action $y_t^{(w)}$ is preferred to the "losing" action $y_t^{(\ell)}$ under the Bradley–Terry–Luce model in Equation (2.1).

**Definition D.2.** *A sequence of context-action pairs $(x_1, y_1), \ldots, (x_k, y_k)$ is $\epsilon$-independent for a function class $\mathcal{R}$ if for every $i \in \{1, \ldots, k\}$, there exist two functions $r_1, r_2 \in \mathcal{R}$ such that $|r_1(x_j, y_j) - r_2(x_j, y_j)| \leq \epsilon$ for all $j < i$, but $|r_1(x_i, y_i) - r_2(x_i, y_i)| > \epsilon$. The $\epsilon$-eluder dimension, $d_{\mathrm{eluder}}(\mathcal{R}, \epsilon)$, is the length of the longest such sequence.*

A low eluder dimension means that after a few queries, any two functions consistent with the observations must be close everywhere, enabling efficient learning. This complexity measure is connected to regret via the following lemma.

**Lemma D.3** (Variance–information lemma, Restated emphasizing the BTL model)**.** *Fix a round $t$ and condition on the $\sigma$-algebra $\mathcal{F}_{t-1}$ and on the chosen context $x_t$ and comparison pair $(y_t^{(w)}, y_t^{(\ell)})$. Let $\mu_t$ denote the posterior distribution of $\theta^*$ given $\mathcal{F}_{t-1}$, and define the posterior predictive variance of the logit difference at the queried pair by*

$$V_t \;:=\; \mathrm{Var}_{\theta\sim\mu_t}\Big(r_\theta(x_t, y_t^{(w)}) - r_\theta(x_t, y_t^{(\ell)})\Big).$$

*Under Assumption 2.1 and the Bradley–Terry–Luce preference model equation 2.1, there exists a constant $\lambda_{\mathrm{BTL}} > 0$, depending only on the logistic link and the reward range, such that*

$$I\big(\theta^*; \mathrm{feedback}_t \mid \mathcal{F}_{t-1}\big) \;\geq\; \lambda_{\mathrm{BTL}}\, V_t. \tag{D.3}$$

This inequality connects the conditional mutual information at round $t$ to the posterior predictive variance $V_t$ of the queried logit difference, and will be combined with the eluder-dimension analysis below to control the cumulative predictive variances $\sum_{t=1}^{T} V_t$.

*Proof.* Throughout the proof we work conditionally on $\mathcal{F}_{t-1}$, $x_t$ and $(y_t^{(w)}, y_t^{(\ell)})$ and suppress this conditioning from the notation.

**Step 1: Reducing to the random preference probability.**    Define, for each parameter $\theta$,

$$z_\theta := r_\theta(x_t, y_t^{(w)}) - r_\theta(x_t, y_t^{(\ell)}), \qquad p_\theta := \sigma(z_\theta),$$

and let $Z_t$ and $P_t$ denote the random variables obtained by drawing $\theta \sim \mu_t$ and applying these maps. By Assumption 2.1 we have $r_\theta(x, y) \in [0, 1]$ for all $(x, y)$ and $\theta$, so $z_\theta \in [-1, 1]$ and hence $P_t \in [\sigma(-1), \sigma(1)] \subset (0, 1)$.

Under the BTL model equation 2.1, the binary preference feedback $\text{feedback}_t \in \{0, 1\}$ satisfies

$$\mathbb{P}\big(\text{feedback}_t = 1 \,\big|\, \theta\big) = p_\theta, \qquad \mathbb{P}\big(\text{feedback}_t = 0 \,\big|\, \theta\big) = 1 - p_\theta.$$

In particular, $\text{feedback}_t$ depends on $\theta$ only through the scalar $P_t$, and we have the Markov chain

$$\theta^* \;\to\; Z_t \;\to\; P_t \;\to\; \text{feedback}_t.$$

Because $P_t$ is a deterministic function of $\theta^*$ and $\text{feedback}_t$ is conditionally independent of $\theta^*$ given $P_t$, standard properties of mutual information give

$$I\big(\theta^*; \text{feedback}_t \mid \mathcal{F}_{t-1}\big) = I\big(P_t; \text{feedback}_t \mid \mathcal{F}_{t-1}\big).$$

It therefore suffices to lower bound the mutual information between the random Bernoulli parameter $P_t$ and the feedback.

**Step 2: Mutual information as entropy drop.**    Let $H(p) = -p \log p - (1 - p) \log(1 - p)$ denote the binary entropy (in nats). Conditioning on $\mathcal{F}_{t-1}$, write

$$\bar{p}_t := \mathbb{E}[P_t],$$

where the expectation is with respect to $\theta \sim \mu_t$. Since $\text{feedback}_t \mid P_t \sim \text{Bernoulli}(P_t)$, we have

$$H\big(\text{feedback}_t \mid P_t\big) = H(P_t), \qquad H\big(\text{feedback}_t\big) = H(\bar{p}_t),$$

and thus

$$I\big(P_t; \text{feedback}_t \mid \mathcal{F}_{t-1}\big) = H(\bar{p}_t) - \mathbb{E}\big[H(P_t)\big], \tag{D.4}$$

where the expectation is over $P_t$.

**Step 3: Strong concavity of binary entropy.**    The binary entropy is twice differentiable on $(0, 1)$ with

$$H''(p) = -\frac{1}{p} - \frac{1}{1 - p}, \qquad p \in (0, 1).$$

For all $p \in (0, 1)$ we have $H''(p) \leq -4$, with equality at $p = 1/2$. Hence $H$ is 4-strongly concave on any compact subinterval of $(0, 1)$, and in particular on $[\sigma(-1), \sigma(1)]$.

We now recall a standard fact about strongly concave functions.

**Claim.** Let $f$ be twice differentiable and $\lambda$-strongly concave on an interval $I \subset \mathbb{R}$, that is, $f''(x) \leq -\lambda$ for all $x \in I$. If $X$ is a real random variable taking values in $I$ with mean $m = \mathbb{E}[X]$, then

$$f(m) - \mathbb{E}[f(X)] \;\geq\; \frac{\lambda}{2} \text{Var}(X).$$

*Proof of the claim.* For each realization $X = x$ there exists (by Taylor's theorem with Lagrange remainder) a point $\xi_x$ on the line segment between $m$ and $x$ such that

$$f(x) = f(m) + f'(m)(x - m) + \tfrac{1}{2} f''(\xi_x)(x - m)^2.$$

Taking expectations, and using $\mathbb{E}[X - m] = 0$, we obtain

$$\mathbb{E}[f(X)] = f(m) + \tfrac{1}{2}\,\mathbb{E}\big[f''(\xi_x)(X - m)^2\big].$$

Since $f''(\xi_x) \leq -\lambda$ for all $\xi_x \in I$, we conclude

$$f(m) - \mathbb{E}[f(X)] = -\tfrac{1}{2}\,\mathbb{E}\big[f''(\xi_x)(X-m)^2\big] \;\geq\; \tfrac{\lambda}{2}\,\mathbb{E}[(X-m)^2] = \tfrac{\lambda}{2}\,\mathrm{Var}(X),$$

which proves the claim. $\qquad\square$

Applying the claim with $f = H$, $\lambda = 4$ and $X = P_t$ (which is supported on $[\sigma(-1), \sigma(1)]$) yields

$$H(\bar{p}_t) - \mathbb{E}\big[H(P_t)\big] \;\geq\; 2\,\mathrm{Var}(P_t). \tag{D.5}$$

Combining equation D.4 and equation D.5 we obtain

$$I\big(\theta^*;\,\mathrm{feedback}_t \mid \mathcal{F}_{t-1}\big) = I\big(P_t;\,\mathrm{feedback}_t \mid \mathcal{F}_{t-1}\big) \;\geq\; 2\,\mathrm{Var}(P_t). \tag{D.6}$$

**Step 4: Relating variance of $P_t$ to variance of the logit.** The sigmoid function $\sigma(z) = (1 + e^{-z})^{-1}$ is continuously differentiable on $\mathbb{R}$ with derivative $\sigma'(z) = \sigma(z)\big(1 - \sigma(z)\big)$. On the compact interval $[-1, 1]$ we have

$$0 < c_{\min} \;\leq\; \sigma'(z) \;\leq\; c_{\max} < 1/4, \qquad z \in [-1, 1],$$

where $c_{\min} := \min_{z \in [-1,1]} \sigma'(z) = \sigma(-1)\big(1 - \sigma(-1)\big) > 0$. Thus $\sigma$ is strictly increasing with derivative bounded away from 0 on $[-1, 1]$, and its inverse $g := \sigma^{-1}$ is well-defined and Lipschitz on $[\sigma(-1), \sigma(1)]$ with Lipschitz constant $L = 1/c_{\min}$.

By definition $P_t = \sigma(Z_t)$ and $Z_t \in [-1, 1]$, so $Z_t = g(P_t)$ and

$$\begin{aligned}
\mathrm{Var}(Z_t) = \mathrm{Var}\big(g(P_t)\big) &= \mathbb{E}\big[(g(P_t) - \mathbb{E}[g(P_t)])^2\big] \\
&\leq \mathbb{E}\big[(g(P_t) - g(\mathbb{E}[P_t]))^2\big] \leq L^2\,\mathbb{E}\big[(P_t - \mathbb{E}[P_t])^2\big] = L^2\,\mathrm{Var}(P_t),
\end{aligned}$$

where we used the Lipschitz property of $g$ and the fact that the variance is upper bounded by the second moment around any fixed reference point. Rearranging yields

$$\mathrm{Var}(P_t) \;\geq\; c_{\min}^2\,\mathrm{Var}(Z_t).$$

Recalling that $Z_t = r_\theta(x_t, y_t^{(w)}) - r_\theta(x_t, y_t^{(\ell)})$, we conclude that

$$\mathrm{Var}(P_t) \;\geq\; c_{\min}^2\,\mathrm{Var}_{\theta \sim \mu_t}\Big(r_\theta(x_t, y_t^{(w)}) - r_\theta(x_t, y_t^{(\ell)})\Big) = c_{\min}^2\,V_t. \tag{D.7}$$

**Step 5: Combine.** Combining equation D.6 and equation D.7 we obtain

$$I\big(\theta^*;\,\mathrm{feedback}_t \mid \mathcal{F}_{t-1}\big) \;\geq\; 2\,\mathrm{Var}(P_t) \;\geq\; 2c_{\min}^2\,V_t.$$

Defining

$$\lambda_{\mathrm{BTL}} := 2\,c_{\min}^2 = 2\Big(\sigma(-1)\big(1 - \sigma(-1)\big)\Big)^2 > 0,$$

we arrive at the desired inequality $I(\theta^*;\,\mathrm{feedback}_t \mid \mathcal{F}_{t-1}) \geq \lambda_{\mathrm{BTL}} V_t$. This constant depends only on the BTL link function $\sigma$ and the reward range $r_\theta(x, y) \in [0, 1]$ (which ensures $Z_t \in [-1, 1]$). $\quad\square$

**Curvature of the BTL link.** By Assumption 2.1 we have $r_\theta(x, y) \in [0, 1]$ for all $(x, y)$ and $\theta$, so the logit differences $z_\theta(x, y_w, y_\ell) := r_\theta(x, y_w) - r_\theta(x, y_\ell)$ lie in $[-1, 1]$. On this compact interval the negative log-likelihood of the Bradley–Terry–Luce model equation 2.1 is uniformly strongly convex. Consequently, there exists a constant $\lambda_{\mathrm{BTL}} > 0$ such that the Kullback–Leibler divergence between any two preference models is bounded below by $\lambda_{\mathrm{BTL}}$ times the squared difference in their logits. This curvature is exactly what underlies the variance–information Lemma D.3.

### D.3 SHARP ELUDER-DIMENSION CONTROL (FOR LoRA-BASED MODELS)

A key argument for the practical relevance of our theory is that the eluder dimension for massive models is not as large as their parameter count might suggest, especially when using parameter-efficient fine-tuning methods like LoRA (Hu et al., 2022).

**Proposition D.4.** *Consider a reward function class $\mathcal{R}$ parameterized by a large neural network with weights $W_0 \in \mathbb{R}^{d \times d'}$. Let the fine-tuning be restricted to a LoRA update $W = W_0 + AB$, where $A \in \mathbb{R}^{d \times d_*}$, $B \in \mathbb{R}^{d_* \times d'}$, and $d_* \ll d, d'$. The trainable parameters are the entries of $A$ and $B$. Under standard smoothness assumptions on the network architecture, the eluder dimension of this class scales as $d_{\mathrm{eluder}}(\mathcal{R}, \epsilon) = \tilde{\mathcal{O}}(d_*(d + d') \log(1/\epsilon))$, not with the full parameter count $d \times d'$.*

*Proof.* The proof follows from the observation that the reward function $r_{A,B}(x, y)$ is a smooth function of the low-rank matrices $A$ and $B$. The effective number of parameters is $d_*(d + d')$. Applying standard covering number arguments for Lipschitz function classes to this lower-dimensional parameter space yields the stated bound on the eluder dimension. This result formalizes the intuition that the intrinsic dimensionality of the fine-tuning task is what governs the exploration complexity (Aghajanyan et al., 2020; Li et al., 2022). $\square$

**Assumption D.5** (Blockwise Lipschitzness for LoRA layers). *For each modified layer $\ell \in [L]$ with base weight $W_\ell \in \mathbb{R}^{m_\ell \times n_\ell}$ and low-rank update $A_\ell B_\ell^\top$ with rank $r_\ell$, we assume the reward (or preference log-likelihood) is $L_\ell$-Lipschitz in each block parameter and smooth in the base activations, uniformly over the input domain. That is, for all admissible inputs, perturbations $(\Delta A_\ell, \Delta B_\ell)$ satisfy*

$$\left| \mathcal{R}\big(W_\ell + (A_\ell + \Delta A_\ell)(B_\ell + \Delta B_\ell)^\top\big) - \mathcal{R}\big(W_\ell + A_\ell B_\ell^\top\big) \right| \leq L_\ell \big(\|\Delta A_\ell\|_F + \|\Delta B_\ell\|_F\big).$$

**Corollary D.6** (Intrinsic dimension under blockwise Lipschitz LoRA). *Under Theorem D.5, the eluder dimension of the LoRA-parameterized reward class satisfies, for any $\epsilon \in (0, 1]$,*

$$d_{\mathrm{eluder}}(\epsilon; \mathcal{R}_{LoRA}) \leq C \left( \sum_{\ell=1}^{L} r_\ell \big(m_\ell + n_\ell - r_\ell\big) \right) \log \frac{C'}{\epsilon},$$

*for universal positive constants $C, C'$. In particular, the effective intrinsic dimension scales with the rank budget rather than the ambient parameter count, aligning with empirical observations on parameter-efficient fine-tuning (Hu et al., 2022; Aghajanyan et al., 2020).*

*Proof.* **Step 1 (Model class and parameterization).** Let $\mathcal{R}$ denote the LoRA-parameterized reward class obtained by freezing a base network and adding, in each layer $\ell \in [L]$, a rank-$r_\ell$ update of the form $U_\ell V_\ell^\top$ with $U_\ell \in \mathbb{R}^{m_\ell \times r_\ell}$, $V_\ell \in \mathbb{R}^{n_\ell \times r_\ell}$. Assumption Theorem D.5 ensures *blockwise Lipschitzness*: for any two parameter tuples $\Theta, \Theta'$,

$$\sup_{(x,y)} \left| r_\Theta(x, y) - r_{\Theta'}(x, y) \right| \leq \sum_{\ell=1}^{L} L_\ell \big\| [U_\ell, V_\ell] - [U_\ell', V_\ell'] \big\|_F.$$

**Step 2 (Covering numbers for low-rank blocks).** Fix radii $R_\ell$ so that $\|(U_\ell, V_\ell)\|_F \leq R_\ell$ for all admissible parameters (w.l.o.g. finite by compactness assumptions). For each block $\ell$, the parameter set lives on a smooth manifold of dimension $d_\ell = r_\ell(m_\ell + n_\ell - r_\ell)$. Standard volumetric bounds give an $\epsilon_\ell$-net of size at most $(CR_\ell/\epsilon_\ell)^{d_\ell}$ in Frobenius norm. By the blockwise Lipschitzness, an $(\epsilon_\ell/L_\ell)$-cover in parameters induces an $\epsilon_\ell$-cover in function sup-norm. Taking the product over blocks and distributing a total accuracy $\epsilon$ across blocks (e.g., $\epsilon_\ell = \epsilon/L$) yields the *function-class covering bound*

$$\mathcal{N}(\epsilon, \mathcal{R}, \|\cdot\|_\infty) \leq \prod_{\ell=1}^{L} \left( \frac{C_\ell}{\epsilon} \right)^{d_\ell} = \left( \frac{C}{\epsilon} \right)^{\sum_{\ell=1}^{L} d_\ell}, \tag{D.8}$$

for constants $C_\ell$ depending on $(L_\ell, R_\ell)$ and a universal $C = \prod_\ell C_\ell$. (See, e.g., standard covering-number bounds for low-rank matrix manifolds.)

**Step 3 (From covering numbers to eluder dimension).** By the growth-function argument of Russo & Van Roy (2013; 2014) (see also Lemma Theorem D.3), for any $\epsilon \in (0, 1]$ there exists a universal $C' > 0$ such that

$$d_{\text{eluder}}(\mathcal{R}, \epsilon) \leq C' \sup_{\delta \in [\epsilon, 1]} \log \mathcal{N}(\delta, \mathcal{R}, \|\cdot\|_\infty). \tag{D.9}$$

Combining equation D.8 and equation D.9 gives

$$d_{\text{eluder}}(\mathcal{R}, \epsilon) \leq C' \Big( \sum_{\ell=1}^{L} d_\ell \Big) \log \frac{C}{\epsilon} = C' \Big( \sum_{\ell=1}^{L} r_\ell \big( m_\ell + n_\ell - r_\ell \big) \Big) \log \frac{C}{\epsilon},$$

which is precisely the claimed bound (absorbing constants into $C, C'$).

**Step 4 (Interpretation).** The dependence is *intrinsic*: it scales with the low-rank degrees of freedom and is independent of the ambient widths except through the block dimensions $(m_\ell, n_\ell)$ and Lipschitz constants $L_\ell$. This matches the intuition that parameter-efficient fine-tuning reduces the exploration burden. $\qquad\square$

**Lemma D.7** (Cumulative squared widths). *Let $\{\mathcal{G}_t\}_{t=1}^{T}$ be confidence sets over the reward class $\mathcal{R}$ with radii $\{\beta_t\}_{t=1}^{T}$, and define the width at the chosen action $A_t = (x_t, y_t)$ by*

$$w_t := w_{\mathcal{G}_t}(A_t) := \sup_{f, f' \in \mathcal{G}_t} \big| f(x_t, y_t) - f'(x_t, y_t) \big|.$$

*Assume rewards are bounded in $[0, 1]$, so $w_t \in [0, 1]$ for all $t$, and set*

$$d_{\text{eluder}} := \dim_E\big(\mathcal{R}, T^{-1}\big), \qquad \beta_T := \max_{1 \leq t \leq T} \beta_t.$$

*Then for all $T \geq 2$ there exists a universal constant $C_{\text{w}} > 0$ such that*

$$\sum_{t=1}^{T} w_t^2 \leq C_{\text{w}} \, d_{\text{eluder}} \, \beta_T \, \log(eT), \tag{D.10}$$

*and the same inequality holds for the expectations $\sum_{t=1}^{T} \mathbb{E}[w_t^2]$.*

*Proof.* We follow a dyadic decomposition over scales $\varepsilon \in [T^{-1}, 1]$ combined with Proposition 3 of Russo & Van Roy (2013).

**Step 1: Split very small widths.** We first separate rounds with tiny width:

$$\sum_{t=1}^{T} w_t^2 = \sum_{t=1}^{T} w_t^2 \mathbf{1}\{w_t \leq T^{-1}\} + \sum_{t=1}^{T} w_t^2 \mathbf{1}\{w_t > T^{-1}\}.$$

Since $w_t \leq 1$, the first term is trivially bounded by $T\,(T^{-1})^2 = 1/T$.

**Step 2: Dyadic partition of the nontrivial widths.** Let $K := \lfloor \log_2 T \rfloor$ and define dyadic scales $\varepsilon_k := 2^{-k}$ for $k = 0, 1, \ldots, K$. For each $k$ define

$$S_k := \{ t \leq T : \varepsilon_{k+1} < w_t \leq \varepsilon_k \}.$$

Then each round with $w_t > T^{-1}$ belongs to some $S_k$, and

$$\sum_{t=1}^{T} w_t^2 \mathbf{1}\{w_t > T^{-1}\} \leq \sum_{k=0}^{K} \varepsilon_k^2 \, |S_k|. \tag{D.11}$$

**Step 3: Apply Proposition 3 at each scale.** For $t \in S_k$ we have $w_t > \varepsilon_{k+1}$, hence

$$|S_k| \leq \sum_{t=1}^{T} \mathbf{1}\{w_t > \varepsilon_{k+1}\}.$$

Proposition 3 of Russo & Van Roy (2013) states that, for any $\varepsilon > 0$,

$$\sum_{t=1}^{T} \mathbf{1}\{w_t > \varepsilon\} \ \leq \ \left(\frac{4\beta_T}{\varepsilon^2} + 1\right) \dim_E(\mathcal{R}, \varepsilon) \qquad \text{almost surely.}$$

By monotonicity of the eluder dimension in its scale parameter and the definition of $d_{\text{eluder}}$,

$$\dim_E(\mathcal{R}, \varepsilon_{k+1}) \ \leq \ \dim_E(\mathcal{R}, T^{-1}) \ = \ d_{\text{eluder}}$$

for all $k$ such that $\varepsilon_{k+1} \geq T^{-1}$, which holds for $k = 0, \ldots, K$. Therefore

$$|S_k| \ \leq \ d_{\text{eluder}}\left(\frac{4\beta_T}{\varepsilon_{k+1}^2} + 1\right).$$

Multiplying by $\varepsilon_k^2$ and using $\varepsilon_k = 2\varepsilon_{k+1}$ gives

$$\varepsilon_k^2 |S_k| \ \leq \ d_{\text{eluder}}\left(4\beta_T \frac{\varepsilon_k^2}{\varepsilon_{k+1}^2} + \varepsilon_k^2\right) = d_{\text{eluder}}\left(16\beta_T + \varepsilon_k^2\right).$$

**Step 4: Sum over dyadic scales and combine.** Summing over $k$ and using $\sum_{k=0}^{\infty} \varepsilon_k^2 = \sum_{k=0}^{\infty} 4^{-k} < 2$, we obtain

$$\sum_{t=1}^{T} w_t^2 \mathbf{1}\{w_t > T^{-1}\} \ \leq \ d_{\text{eluder}}\left(16\beta_T(K+1) + 2\right) \ \leq \ d_{\text{eluder}}\left(16\beta_T \log_2(2T) + 2\right).$$

Combining with the $1/T$ bound from Step 1 and absorbing constants into a universal $C_{\text{w}} > 0$ yields equation D.10. The bound is deterministic conditional on $\{w_t\}$, so it also holds for $\sum_{t=1}^{T} \mathbb{E}[w_t^2]$. $\qquad \square$

**Lemma D.8** (Information–eluder-dimension bound). *Let Assumption 2.1 hold and suppose the preference feedback is generated according to the Bradley–Terry–Luce model in equation 2.1. Let $\mathcal{R} = \{r_\theta : \theta \in \Theta\}$ and denote by $\dim_E(\mathcal{R}, \varepsilon)$ the $\varepsilon$-eluder dimension of $\mathcal{R}$ (Russo & Van Roy, 2013). Assume:*

1. *(Bounded rewards) $r_\theta(x, y) \in [0, 1]$ for all $(x, y)$ and all $\theta \in \Theta$.*

2. *(Bounded preference noise) Conditional on $(\theta^*, \mathcal{F}_{t-1}, x_t, y_t^{(w)}, y_t^{(\ell)})$, the binary preference feedback $\text{feedback}_t \in \{0, 1\}$ has mean $\mathbb{E}[\text{feedback}_t \mid \theta^*, \mathcal{F}_{t-1}] = p_{\theta^*}(x_t, y_t^{(w)}, y_t^{(\ell)})$ and is $\sigma$-sub-Gaussian for some $\sigma > 0$.*

3. *(Metric entropy growth) There exists $C_{\text{cov}} > 0$ such that for all $\varepsilon \in (0, 1]$,*

$$\log N\left(\mathcal{R}, \varepsilon, \|\cdot\|_\infty\right) \ \leq \ C_{\text{cov}} \log(1/\varepsilon),$$

*where $N(\mathcal{R}, \varepsilon, \|\cdot\|_\infty)$ is the $\varepsilon$-covering number of $\mathcal{R}$ in the sup-norm.*

*Let $d_{\text{eluder}} := \dim_E(\mathcal{R}, T^{-1})$. Then there exists a constant $C_{\text{info}} > 0$, depending only on $(\sigma, C_{\text{cov}})$ and the constants in Assumption 2.1, such that for any horizon $T \geq 2$,*

$$I\left(\theta^*; \text{feedback}_{1:T}\right) \ \leq \ C_{\text{info}} \, d_{\text{eluder}} \log^2(eT), \qquad \text{(D.12)}$$

*where $\text{feedback}_{1:T} = (\text{feedback}_1, \ldots, \text{feedback}_T)$.*

*Proof.* We follow the confidence-set and width analysis of Russo & Van Roy (2013), adapting it to our preference-learning setting and to mutual information.

**Step 1: Confidence sets and widths.** Let $(\hat{f}_t^{\text{LS}})_{t \geq 1}$ be least-squares predictors based on past data and define confidence sets

$$\mathcal{G}_t := \left\{f \in \mathcal{R} : \|f - \hat{f}_t^{\text{LS}}\|_{2, E_t} \leq \sqrt{\beta_t}\right\},$$

where $\|\cdot\|_{2,E_t}$ is the empirical 2-norm and $\beta_t$ is chosen as in equation (4) of Russo & Van Roy (2013). Their Proposition 2 implies that, with probability at least $1 - 1/T$, we have $f_{\theta^*} \in \mathcal{G}_t$ for all $t \leq T$.

Define the width of $\mathcal{G}_t$ at the selected pair $(x_t, y_t^{(w)}, y_t^{(\ell)})$ by

$$w_t := \sup_{f, f' \in \mathcal{G}_t} \left| f(x_t, y_t^{(w)}, y_t^{(\ell)}) - f'(x_t, y_t^{(w)}, y_t^{(\ell)}) \right|.$$

**Step 2: From widths to information.** Under the Bradley–Terry–Luce model, the preference probability $p_\theta(x_t, y_t^{(w)}, y_t^{(\ell)})$ is a smooth and bounded function of the reward difference $r_\theta(x_t, y_t^{(w)}) - r_\theta(x_t, y_t^{(\ell)})$. Combining the variance–information lemma D.3 with the boundedness and Lipschitz properties of the logistic link, one obtains an information–width inequality: there exists $c_0 > 0$, depending only on the link function and the reward range, such that

$$I\big(\theta^*; \text{feedback}_t \mid \mathcal{F}_{t-1}\big) \ \leq \ c_0 \, \mathbb{E}\big[w_t^2 \mid \mathcal{F}_{t-1}\big].$$

Summing over $t$ and applying the tower property gives

$$I\big(\theta^*; \text{feedback}_{1:T}\big) = \sum_{t=1}^{T} I\big(\theta^*; \text{feedback}_t \mid \mathcal{F}_{t-1}\big) \ \leq \ c_0 \sum_{t=1}^{T} \mathbb{E}[w_t^2].$$

**Step 3: Bounding the cumulative squared widths.** Applying Lemma D.7 to the confidence sets $\{\mathcal{F}_t\}$ and widths $w_t = w_{\mathcal{F}_t}(A_t)$ constructed in Step 1, we obtain

$$\sum_{t=1}^{T} \mathbb{E}[w_t^2] \ \leq \ C_{\mathrm{w}} \, d_{\mathrm{eluder}} \, \beta_T \, \log(eT),$$

where $d_{\mathrm{eluder}} = \dim_E(\mathcal{R}, T^{-1})$ and $\beta_T = \max_{1 \leq t \leq T} \beta_t$ is the confidence radius.

**Step 4: Controlling $\beta_T$ via metric entropy.** Our metric-entropy assumption implies $\log N(\mathcal{R}, \varepsilon, \|\cdot\|_\infty) \leq C_{\mathrm{cov}} \log(1/\varepsilon)$. Substituting this into the definition of $\beta_T$ (equation (4) in Russo & Van Roy (2013)) with $\varepsilon = T^{-2}$ shows that there exists $C_\beta > 0$ such that

$$\beta_T \ \leq \ C_\beta \, \log T.$$

Combining the displays from Steps 2 and 3 with this bound on $\beta_T$ yields

$$I\big(\theta^*; \text{feedback}_{1:T}\big) \ \leq \ C_{\mathrm{info}} \, d_{\mathrm{eluder}} \, \log^2(eT)$$

for a constant $C_{\mathrm{info}}$ depending only on $(\sigma, C_{\mathrm{cov}})$ and the problem constants, as claimed. $\qquad\square$

## D.4 Approximation of Wasserstein Gradient Flow

This section outlines the approximation of the idealized Wasserstein gradient flow dynamics by a finite ensemble of particles. The core idea is that the particles provide a discrete representation of the continuous flow of probability measures, tracking the evolution of the PAC-Bayesian free-energy functional $J_{\mathrm{PAC}}$.

Let the particles $\{\theta_i\}_{i=1}^{N}$ represent an empirical distribution of parameters at time $t$. We approximate the gradient flow of the PAC-Bayesian objective $J_{\mathrm{PAC}}$ by evolving these particles according to the following update rule:

$$\theta_i^{t+1} = \theta_i^t - \eta \nabla_{\theta_i} \left( \frac{1}{N} \sum_{j=1}^{N} \ell_{\theta_j}(z) + \beta \log \frac{\mu_{\theta_j}}{\Pi(\theta_j)} \right) + \mathcal{N}(0, \sigma^2).$$

Here, $\eta$ is the step size, $\ell_{\theta_j}(z)$ is the loss, and $\mu_{\theta_j}$ represents the empirical measure at time $t$.

The error bound for this approximation depends on the step size $\eta$ and the number of particles $N$, which is formalized in the subsequent lemmas.

**Lemma D.9** (Finite-Particle Approximation Error). *Let $\mu_t$ be the mean-field law and $\mu_t^N$ be the empirical measure of $N$ particles. For any $L$-Lipschitz function $\phi$, the error in estimating its expectation is bounded in probability: $|\int \phi d\mu_t^N - \int \phi d\mu_t| = \widetilde{\mathcal{O}}(1/\sqrt{N})$ (Fournier & Guillin, 2015).*

*Proof.* This follows from classical results on the convergence rate of the empirical measure in Wasserstein distance and the duality between Wasserstein distance and expectations of Lipschitz functions. The error from approximating the interaction term in the SGLD update accumulates, leading to the term in the final regret bound. $\square$

**Lemma D.10.** *Let $\hat{g}_t(\theta)$ be an unbiased mini-batch gradient estimator of the true gradient $g_t(\theta)$ with conditional variance $\mathrm{Var}(\hat{g}_t - g_t \mid \mathcal{F}_{t-1}) \leq \sigma_t^2/B_t$. The cumulative error from the noise sequence $\xi_t = \eta_t(\hat{g}_t - g_t)$ is bounded with high probability by $\widetilde{\mathcal{O}}(\sqrt{\sum_{t=1}^T \eta_t^2 \sigma_t^2/B_t})$.*

*Proof.* Let $\widehat{g}_t(\theta)$ be an unbiased mini-batch estimator of the population gradient $g_t(\theta)$ with $\mathbb{E}[\widehat{g}_t(\theta) \mid \mathcal{F}_{t-1}] = g_t(\theta)$ and conditional covariance $\mathbb{E}\big[\|\widehat{g}_t(\theta) - g_t(\theta)\|^2 \mid \mathcal{F}_{t-1}\big] \leq \sigma_t^2/B_t$. Consider the parameter update $\theta_{t+1} = \theta_t - \eta_t \widehat{g}_t(\theta_t) + $ (other terms) and track the noise contribution to the PAC objective $J(\theta)$ through the descent lemma. Define the noise martingale $\zeta_t := \langle \nabla J(\theta_t), \widehat{g}_t(\theta_t) - g_t(\theta_t)\rangle$ with $\mathbb{E}[\zeta_t \mid \mathcal{F}_{t-1}] = 0$. Then

$$\sum_{t=1}^T \eta_t \, \zeta_t$$

is a martingale with predictable quadratic variation bounded by

$$\sum_{t=1}^T \eta_t^2 \, \mathbb{E}[\zeta_t^2 \mid \mathcal{F}_{t-1}] \; \leq \; \sum_{t=1}^T \eta_t^2 \, \|\nabla J(\theta_t)\|^2 \, \frac{\sigma_t^2}{B_t} \; \leq \; G^2 \sum_{t=1}^T \eta_t^2 \frac{\sigma_t^2}{B_t},$$

where $G$ bounds $\|\nabla J(\theta)\|$ on the iterates (ensured by standard coercivity/compacity arguments in our setting). Applying Freedman's inequality (or Azuma–Hoeffding with conditional variances) yields, with probability at least $1 - \delta$,

$$\Big|\sum_{t=1}^T \eta_t \, \zeta_t\Big| \; \leq \; c_1 \, G \, \sqrt{\log \tfrac{2}{\delta}} \, \sqrt{\sum_{t=1}^T \eta_t^2 \frac{\sigma_t^2}{B_t}} \; + \; c_2 \, G \, \log \tfrac{2}{\delta} \, \max_t \eta_t \frac{\sigma_t}{\sqrt{B_t}},$$

establishing the stated $\widetilde{\mathcal{O}}\big(\sqrt{\sum_t \eta_t^2 \sigma_t^2/B_t}\big)$ high-probability control on the cumulative stochastic-gradient error. $\square$

**Lemma D.11** (Finite-Ensemble Monte Carlo Error). *Let the Monte Carlo error in estimating the optimistic index be $\xi_t = \hat{I}_t - I_t$, with $\mathbb{E}[\xi_t \mid \mathcal{F}_{t-1}] = 0$ and $\mathrm{Var}(\xi_t \mid \mathcal{F}_{t-1}) \leq v_t^2/N_t$. The cumulative error $\sum_{t=1}^T \xi_t$ is bounded with high probability by $\widetilde{\mathcal{O}}(\sqrt{\sum_{t=1}^T v_t^2/N_t})$.*

*Proofs of Theorem D.10 and Theorem D.11.* Both proofs rely on the same core argument. The error sequences $\{\xi_t\}$ in both cases are martingale difference sequences with respect to the filtration $\mathcal{F}_{t-1}$. We can therefore apply a concentration inequality for martingales. Freedman's inequality is particularly well-suited as it handles predictable, time-varying variance bounds (Freedman, 1975). Let $S_T = \sum_{t=1}^T \xi_t$. Let $V_T = \sum_{t=1}^T \mathbb{E}[\xi_t^2 \mid \mathcal{F}_{t-1}]$ be the predictable quadratic variation. Freedman's inequality states that for any $u, v > 0$:

$$\Pr(S_T \geq u \text{ and } V_T \leq v) \leq \exp\left(-\frac{u^2/2}{v + cu/3}\right)$$

where $c$ is a uniform bound on $|\xi_t|$. Setting $v$ to be the sum of our variance bounds (e.g., $v = \sum v_t^2/N_t$) and solving for $u$ for a given probability level $\delta$ yields the stated $\widetilde{\mathcal{O}}(\sqrt{\cdot})$ bounds. $\square$

**Variance and noise parameters.** Throughout this paper we assume bounded rewards and sub-Gaussian noise. In particular, there exist finite constants $v^2$ and $\sigma^2$ such that for all $t$,

$$\text{Var}(\Delta M_t \mid \mathcal{F}_t) \leq v^2, \qquad \mathbb{E}\big[\exp(\lambda \varepsilon_t) \mid \mathcal{F}_t\big] \leq \exp(\lambda^2 \sigma^2/2) \quad \forall \lambda \in \mathbb{R},$$

where $(\mathcal{F}_t)_{t\geq 0}$ is the natural filtration introduced above, $\Delta M_t$ is the martingale increment in the regret decomposition, and $\varepsilon_t$ is the reward noise. We denote by $v_t^2$ and $\sigma_t^2$ the corresponding conditional variance and sub-Gaussian proxy at time $t$, and the above assumptions imply $v_t^2 \leq v^2$ and $\sigma_t^2 \leq \sigma^2$ for all $t$.

## E    MAIN THEOREMS: FULL STATEMENTS AND PROOFS (BANDITS)

This section contains the *full* proofs of the main results. It relies on auxiliary tools in Sections C and D.

### E.1    RESTATEMENT OF MAIN THEOREMS

Let Assumptions 2.1, 2.2, and 2.3 hold. For any $\delta \in (0,1)$, consider the OLE algorithm run for $T$ rounds with step sizes $\{\eta_t\}$, ensemble sizes $\{N_t\}$, mini-batch sizes $\{B_t\}$, and an optimism schedule $\kappa_t = C_0 \sqrt{\log(T/\delta)}$ for a suitable constant $C_0$. Let $v_t^2$ be an upper bound on the conditional variance of the Monte Carlo estimate of the optimistic value, and let $\sigma_t^2$ be an upper bound on the conditional variance of the mini-batch gradient estimator. Then with probability at least $1 - \delta$, the cumulative regret satisfies:

$$\text{Regret}(T) \leq \underbrace{C_1\, d_{\text{eluder}} \log T}_{\text{Exploration Cost}} + C_2 \underbrace{\sum_{t=1}^{T} \eta_t}_{\text{Discretization}} + \underbrace{\widetilde{\mathcal{O}}\left(\sqrt{\sum_{t=1}^{T} \frac{v_t^2}{N_t}}\right)}_{\text{Finite Ensemble}} + \underbrace{\widetilde{\mathcal{O}}\left(\sqrt{\sum_{t=1}^{T} \frac{\sigma_t^2}{B_t}}\right)}_{\text{Stochastic Gradient}}, \qquad \text{(E.1)}$$

where $C_1$ and $C_2$ are absolute constants depending on model parameters like the Lipschitz constant $L$. The eluder dimension $d_{\text{eluder}}$ is evaluated at a precision scale that decreases with $t$.

### E.2    PROOF OF THE UNIFIED REGRET BOUND (SECTION E.1)

We now prove Section E.1. Throughout, we let $(\mathcal{F}_t)_{t\geq 0}$ denote the natural filtration generated by all randomness up to the end of round $t$ (contexts, actions, preference feedback, and the internal randomness of the ensemble and SGLD).

Recall that at each round $t$, the OLE algorithm computes, for each candidate action $y \in \mathcal{Y}$ in context $x_t$, an optimistic index

$$I_t(x_t, y) = \hat{r}_t(x_t, y) + \kappa_t \sqrt{\widehat{\text{Var}}_t(x_t, y)}, \qquad \text{(E.2)}$$

where $\hat{r}_t(x_t, y)$ and $\widehat{\text{Var}}_t(x_t, y)$ are the ensemble mean and variance, respectively, and $\kappa_t = C_0 \sqrt{\log(T/\delta)}$. We will bound the regret by: (i) decomposing the instantaneous regret at each round into an optimism term and an estimation term; (ii) controlling the sum of optimism terms using variance–information duality and the eluder dimension; and (iii) bounding the estimation term via a careful decomposition into discretization, finite-ensemble, and stochastic-gradient contributions.

**Step 1: Instantaneous regret decomposition.** Let $y_t^* \in \arg\max_{y\in\mathcal{Y}} r^*(x_t, y)$ denote an optimal action in context $x_t$, and let $y_t$ denote the action chosen by the OLE policy induced by the index $I_t$ (for simplicity, we write the regret in terms of the deployed action $y_t$, which is one element of the selected comparison pair). The instantaneous regret is

$$r^*(x_t, y_t^*) - r^*(x_t, y_t).$$

Introduce the shorthand $I_t^* := I_t(x_t, y_t^*), I_t := I_t(x_t, y_t), \hat{r}_t := \hat{r}_t(x_t, y_t), r_t^* := r^*(x_t, y_t)$, and note that by definition of $y_t$, we have $I_t \geq I_t^*$. Then

$$r^*(x_t, y_t^*) - r^*(x_t, y_t) = \underbrace{\left(I_t^* - r^*(x_t, y_t)\right)}_{\text{(I)}} - \underbrace{\left(I_t^* - r^*(x_t, y_t^*)\right)}_{\text{(II)}} \tag{E.3}$$

$$\leq \underbrace{\left(I_t - \hat{r}_t\right)}_{\text{Optimism term}} + \underbrace{\left(\hat{r}_t - r_t^*\right)}_{\text{Estimation error}} -\text{(II)}. \tag{E.4}$$

The inequality uses $I_t \geq I_t^*$ and inserts and subtracts $\hat{r}_t$. Define the "good optimism event"

$$\mathcal{E}_t := \{ r^*(x, y) \leq I_t(x, y) \text{ for all } (x, y) \in \mathcal{X} \times \mathcal{Y} \}.$$

By the PAC-Bayes and concentration arguments developed in Sections C and D, together with the choice $\kappa_t = C_0 \sqrt{\log(T/\delta)}$, we can ensure that

$$\Pr\left(\bigcap_{t=1}^T \mathcal{E}_t\right) \geq 1 - \delta. \tag{E.5}$$

On the event $\mathcal{E}_t$, we have $r^*(x_t, y_t^*) \leq I_t^*$, so term (II) in equation E.3 is non-negative. Hence, on $\bigcap_{t=1}^T \mathcal{E}_t$,

$$r^*(x_t, y_t^*) - r^*(x_t, y_t) \leq \underbrace{\kappa_t \sqrt{\widehat{\mathrm{Var}}_t(x_t, y_t)}}_{\Delta_t^{\mathrm{opt}}} + \underbrace{\left(\hat{r}_t(x_t, y_t) - r^*(x_t, y_t)\right)}_{\epsilon_t}, \tag{E.6}$$

where we have set $\Delta_t^{\mathrm{opt}} := I_t(x_t, y_t) - \hat{r}_t(x_t, y_t) = \kappa_t \sqrt{\widehat{\mathrm{Var}}_t(x_t, y_t)}$ and $\epsilon_t := \hat{r}_t(x_t, y_t) - r^*(x_t, y_t)$.

Summing over $t = 1, \ldots, T$ and working on the event $\mathcal{E} := \bigcap_{t=1}^T \mathcal{E}_t$ yields

$$\mathrm{Regret}(T) \leq \sum_{t=1}^T \Delta_t^{\mathrm{opt}} + \sum_{t=1}^T \epsilon_t. \tag{E.7}$$

We will bound the two sums on the right-hand side separately.

**Step 2: Bounding the cumulative optimism term.** Write

$$V_t := \widehat{\mathrm{Var}}_t(x_t, y_t), \qquad \Delta_t^{\mathrm{opt}} = \kappa_t \sqrt{V_t}.$$

By Cauchy–Schwarz,

$$\sum_{t=1}^T \Delta_t^{\mathrm{opt}} = \sum_{t=1}^T \kappa_t \sqrt{V_t} \leq \sqrt{\sum_{t=1}^T 1} \sqrt{\sum_{t=1}^T \kappa_t^2 V_t} = \sqrt{T} \sqrt{\sum_{t=1}^T \kappa_t^2 V_t}. \tag{E.8}$$

Using the choice $\kappa_t^2 = C_0^2 \log(T/\delta)$, it remains to control $\sum_{t=1}^T V_t$.

The variance–information lemma (Theorem D.3) states that, under the Bradley–Terry–Luce preference model equation 2.1, for any posterior distribution $\mu_t$ over parameters at round $t$ and any context $x_t$ and chosen pair $(y_t^{(w)}, y_t^{(\ell)})$, the conditional mutual information satisfies

$$I\left(\theta^*; \mathrm{feedback}_t \mid \mathcal{F}_{t-1}\right) \geq C_{\mathrm{var}} \cdot V_t, \tag{E.9}$$

for some universal constant $C_{\mathrm{var}} > 0$ (depending only on the fact that the logistic link keeps preferences bounded away from 0 and 1).

Summing equation E.9 over $t = 1, \ldots, T$ and using the chain rule for mutual information, we obtain

$$\sum_{t=1}^T V_t \leq C_{\mathrm{var}}^{-1} \sum_{t=1}^T I\left(\theta^*; \mathrm{feedback}_t \mid \mathcal{F}_{t-1}\right) = C_{\mathrm{var}}^{-1} I\left(\theta^*; \mathrm{feedback}_{1:T}\right). \tag{E.10}$$

The remaining ingredient is to bound the total information gain in terms of the eluder dimension. This is guaranteed by the lemma D.8. Combining equation E.10 and lemma D.8 yields

$$\sum_{t=1}^{T} V_t \;\leq\; C_{\mathrm{var}}^{-1} C_{\mathrm{info}}\, d_{\mathrm{eluder}}\, \log^2(eT) \;=:\; C_V\, d_{\mathrm{eluder}}\, \log^2(eT). \tag{E.11}$$

Substituting equation E.11 and $\kappa_t^2 = C_0^2 \log(T/\delta)$ into equation E.8 gives

$$\sum_{t=1}^{T} \Delta_t^{\mathrm{opt}} \leq \sqrt{T}\, \sqrt{\sum_{t=1}^{T} \kappa_t^2 V_t} = \sqrt{T}\, \sqrt{C_0^2 \log(T/\delta) \sum_{t=1}^{T} V_t}$$

$$\leq \sqrt{T}\, \sqrt{C_0^2 \log(T/\delta)\, C_V\, d_{\mathrm{eluder}}\, \log^2(eT)}$$

$$\leq C_1 \sqrt{d_{\mathrm{eluder}} T}\, \log T, \tag{E.12}$$

for a suitable constant $C_1 > 0$ (absorbing $\log(T/\delta)$ into the $\log T$ factor via the $\widetilde{\mathcal{O}}(\cdot)$ notation).

**Step 3: Bounding the cumulative estimation error.** We now bound the second sum in equation E.7, $\sum_{t=1}^{T} \epsilon_t$, which captures the discrepancy between the ensemble prediction $\hat{r}_t$ and the true reward $r^*$ evaluated at $(x_t, y_t)$.

Let $\bar{r}_t(x, y)$ denote the prediction of the *ideal*, continuous-time, infinite-particle mean-field posterior at round $t$. Then at the deployed action $(x_t, y_t)$ we can write

$$\epsilon_t = \hat{r}_t(x_t, y_t) - r^*(x_t, y_t) = \big(\hat{r}_t(x_t, y_t) - \bar{r}_t(x_t, y_t)\big) + \big(\bar{r}_t(x_t, y_t) - r^*(x_t, y_t)\big). \tag{E.13}$$

Summing over $t$ we obtain

$$\sum_{t=1}^{T} \epsilon_t = \underbrace{\sum_{t=1}^{T} \big(\hat{r}_t(x_t, y_t) - \bar{r}_t(x_t, y_t)\big)}_{S_{\mathrm{approx}}} + \underbrace{\sum_{t=1}^{T} \big(\bar{r}_t(x_t, y_t) - r^*(x_t, y_t)\big)}_{S_{\mathrm{ideal}}}. \tag{E.14}$$

*Step 3(a): Ideal mean-field prediction error.* The term $S_{\mathrm{ideal}}$ measures the deviation of the ideal mean-field posterior mean from the true reward. This is controlled by the PAC-Bayesian generalization bounds for the smoothed posterior in Section C, which imply that, under Assumption 2.1, the mean-field posterior concentrates around $\theta^*$ and the average generalization error is small. In particular, standard arguments (see Theorem 3.2 and subsequent discussion) yield

$$|S_{\mathrm{ideal}}| \;\leq\; C_{\mathrm{ideal}} \sqrt{T} \tag{E.15}$$

for some constant $C_{\mathrm{ideal}}$ depending on the Lipschitz constant of the loss and the prior, and this term is dominated by the exploration term equation E.12 when $T$ is large. For simplicity, we absorb $S_{\mathrm{ideal}}$ into the overall $\widetilde{\mathcal{O}}(\sqrt{d_{\mathrm{eluder}} T} \log T)$ term.

*Step 3(b): Approximation error from discretization, finite ensemble, and stochastic gradients.* The term $S_{\mathrm{approx}}$ in equation E.14 captures the effect of replacing the ideal mean-field posterior with a finite ensemble, with discrete-time SGLD dynamics and mini-batch gradients. We now decompose it into a martingale term and a bias term using the filtration $(\mathcal{F}_t)$.

By definition, both $\hat{r}_t$ and $\bar{r}_t$ are $\mathcal{F}_t$-measurable. We write

$$S_{\mathrm{approx}} = \sum_{t=1}^{T} \big(\hat{r}_t(x_t, y_t) - \bar{r}_t(x_t, y_t)\big)$$

$$= \underbrace{\sum_{t=1}^{T} \big(\hat{r}_t(x_t, y_t) - \mathbb{E}[\hat{r}_t(x_t, y_t) \mid \mathcal{F}_{t-1}]\big)}_{S_{\mathrm{mart}}: \ \text{martingale noise}} + \underbrace{\sum_{t=1}^{T} \big(\mathbb{E}[\hat{r}_t(x_t, y_t) \mid \mathcal{F}_{t-1}] - \bar{r}_t(x_t, y_t)\big)}_{S_{\mathrm{bias}}: \ \text{bias/approximation term}}.$$

The first term $S_{\mathrm{mart}}$ is a martingale difference sequence with conditionally sub-Gaussian increments whose conditional variances are bounded by $v_t^2/N_t$ (finite-ensemble Monte Carlo noise) and by $\eta_t^2 \sigma_t^2/B_t$ (stochastic-gradient noise). The second term $S_{\mathrm{bias}}$ collects the approximation bias arising from discretizing the Langevin dynamics.

The following lemmas, proved in Appendix D.4, give high-probability bounds for each contribution (we quote them here for convenience):

**Lemma E.1** (Finite-ensemble Monte Carlo error). *Under the conditions of Theorems 2.1 and 2.2, the finite-ensemble Monte Carlo fluctuations satisfy*

$$\left| \sum_{t=1}^{T} \Big( \hat{r}_t^{\mathrm{MC}}(x_t, y_t) - \mathbb{E}[\hat{r}_t^{\mathrm{MC}}(x_t, y_t) \mid \mathcal{F}_{t-1}] \Big) \right| \leq C_{\mathrm{ens}} \sqrt{ \sum_{t=1}^{T} \frac{v_t^2}{N_t} }$$

*with probability at least $1 - \delta/4$, for some constant $C_{\mathrm{ens}} > 0$. (Here $\hat{r}_t^{\mathrm{MC}}$ denotes the Monte Carlo estimate of the ensemble mean.)*

**Lemma E.2** (Stochastic-gradient error). *Let $\hat{g}_t(\theta)$ be an unbiased mini-batch gradient estimator with conditional variance bounded by $\sigma_t^2/B_t$. Then the cumulative error induced by using $\hat{g}_t$ instead of the exact gradient in SGLD satisfies*

$$\left| \sum_{t=1}^{T} \eta_t \Big( \hat{g}_t(\theta_t) - \mathbb{E}[\hat{g}_t(\theta_t) \mid \mathcal{F}_{t-1}] \Big) \right| \leq C_{\mathrm{sg}} \sqrt{ \sum_{t=1}^{T} \eta_t^2 \frac{\sigma_t^2}{B_t} }$$

*with probability at least $1 - \delta/4$.*

**Lemma E.3** (Discretization bias). *Suppose the drift of the mean-field Langevin SDE is $L$-Lipschitz and satisfies the standard coercivity conditions ensuring existence of a unique invariant measure. Then the cumulative bias induced by using a time step $\eta_t$ in the Euler–Maruyama discretization satisfies*

$$|S_{\mathrm{bias}}| \leq C_{\mathrm{disc}} \sum_{t=1}^{T} \eta_t,$$

*for some constant $C_{\mathrm{disc}} > 0$.*

Combining equation E.2 with Theorems E.1 to E.3 and applying a union bound over the associated high-probability events yields

$$|S_{\mathrm{approx}}| \leq C_2 \sum_{t=1}^{T} \eta_t + C_3 \sqrt{ \sum_{t=1}^{T} \frac{v_t^2}{N_t} } + C_4 \sqrt{ \sum_{t=1}^{T} \frac{\sigma_t^2}{B_t} }, \tag{E.16}$$

for appropriate constants $C_2, C_3, C_4 > 0$.

**Step 4: Combine.** Combining the instantaneous decomposition equation E.7, the optimism bound equation E.12, and the bounds equation E.15 and equation E.16, and applying a final union bound over all the high-probability events ($\mathcal{E}$, the PAC-Bayes generalization bound, and the martingale concentration events), we obtain that with probability at least $1 - \delta$,

$$\mathrm{Regret}(T) \leq C_1 \sqrt{d_{\mathrm{eluder}} T} \log T + C_2 \sum_{t=1}^{T} \eta_t + C_3 \sqrt{ \sum_{t=1}^{T} \frac{v_t^2}{N_t} } + C_4 \sqrt{ \sum_{t=1}^{T} \frac{\sigma_t^2}{B_t} },$$

which is exactly the statement of Section E.1. This completes the proof.

**Proposition E.4** (Exploration term and logarithmic regret). *Under Assumptions 2.1, 2.2, and 2.3, and on the high-probability optimism event equation E.5, there exists a constant $C_{\mathrm{opt}} > 0$ such that for all $T \geq 2$,*

$$\sum_{t=1}^{T} \kappa_t \sqrt{\widehat{\mathrm{Var}}_t(x_t, y_t)} \leq C_{\mathrm{opt}} \, d_{\mathrm{eluder}} \, \log T. \tag{E.17}$$

*Proof.* Recall that $\mathcal{F}_{t-1}$ denotes the $\sigma$-algebra generated by all randomness up to round $t-1$ (contexts, actions, feedback, and algorithm randomness), and that $\text{feedback}_t$ denotes the preference observation at round $t$. For each $t$, we define the per-round mutual information

$$I_t := I\big(\theta^*; \text{feedback}_t \mid \mathcal{F}_{t-1}\big) = \mathbb{E}\Big[D_{\text{KL}}\big(P(\text{feedback}_t \mid \theta^*, \mathcal{F}_{t-1}) \,\big\|\, P(\text{feedback}_t \mid \mathcal{F}_{t-1})\big)\Big],$$

where the expectation is taken over $(\theta^*, \text{feedback}_t)$ and $\mathcal{F}_{t-1}$.

**Step 1: Chain rule for mutual information.** By the chain rule for mutual information applied to the sequence $\text{feedback}_{1:T} = (\text{feedback}_1, \ldots, \text{feedback}_T)$ we have

$$I\big(\theta^*; \text{feedback}_{1:T}\big) = \sum_{t=1}^{T} I\big(\theta^*; \text{feedback}_t \mid \text{feedback}_{1:t-1}\big) = \sum_{t=1}^{T} I_t. \tag{E.18}$$

Since $\mathcal{F}_{t-1}$ is the $\sigma$-algebra generated by $\text{feedback}_{1:t-1}$ together with the other past randomness of the algorithm, we can (and will) work with the conditional information $I(\theta^*; \text{feedback}_t \mid \mathcal{F}_{t-1})$.

**Step 2: Information–width inequality for the BTL model.** Under the Bradley–Terry–Luce preference model with bounded rewards, the mutual information gained at round $t$ can be controlled by the squared width of the corresponding confidence set. More precisely, combining the variance–information lemma (Lemma D.3) with the construction of the confidence sets $\{\mathcal{G}_t\}$ and the Lipschitz properties of the logistic link, there exists a constant $c_0 > 0$, depending only on the link and the reward range, such that for every $t$ and every realization of $\mathcal{F}_{t-1}$,

$$I\big(\theta^*; \text{feedback}_t \mid \mathcal{F}_{t-1}\big) \leq c_0\, \mathbb{E}\big[w_t^2 \mid \mathcal{F}_{t-1}\big], \tag{E.19}$$

where $w_t$ is the width at time $t$,

$$w_t := \sup_{f,f' \in \mathcal{G}_t} \big|f(x_t, y_t^{(w)}, y_t^{(\ell)}) - f'(x_t, y_t^{(w)}, y_t^{(\ell)})\big|.$$

(Informally, equation E.19 says that, given the past, the amount of information we can gain at round $t$ is controlled by the squared width of the current confidence set at the queried comparison.)

Taking expectations of both sides of equation E.19 and using the tower property yields

$$\mathbb{E}[I_t] = \mathbb{E}\big[I(\theta^*; \text{feedback}_t \mid \mathcal{F}_{t-1})\big] \leq c_0\, \mathbb{E}\big[\mathbb{E}[w_t^2 \mid \mathcal{F}_{t-1}]\big] = c_0\, \mathbb{E}[w_t^2]. \tag{E.20}$$

**Step 3: Summing over $t$ and invoking cumulative squared widths.** Plugging equation E.20 into the chain rule equation E.18 and using linearity of expectation, we obtain

$$I\big(\theta^*; \text{feedback}_{1:T}\big) = \sum_{t=1}^{T} I_t = \sum_{t=1}^{T} \mathbb{E}[I_t] \tag{E.21}$$

$$\leq \sum_{t=1}^{T} c_0\, \mathbb{E}[w_t^2] = c_0 \sum_{t=1}^{T} \mathbb{E}[w_t^2]. \tag{E.22}$$

Lemma D.7 (cumulative squared widths) now gives

$$\sum_{t=1}^{T} \mathbb{E}[w_t^2] \leq C_{\text{w}}\, d_{\text{eluder}}\, \beta_T\, \log(eT),$$

where $d_{\text{eluder}} := \dim_E(\mathcal{R}, T^{-1})$ is the $T^{-1}$-eluder dimension of the reward class. Combining this with equation E.22 yields

$$I\big(\theta^*; \text{feedback}_{1:T}\big) \leq c_0 C_{\text{w}}\, d_{\text{eluder}}\, \beta_T\, \log(eT). \tag{E.23}$$

Defining $C_{\text{info}} := c_0 C_{\text{w}}$ gives the claimed information bound.

The remainder of the proof proceeds by combining this information bound with the variance-based optimism inequality (bounding the instantaneous regret by $\kappa_t \sqrt{\widehat{\text{Var}}_t(x_t, y_t)}$) and a dyadic decomposition argument on the predictive variances. This shows that the cumulative optimism term $\sum_{t=1}^{T} \kappa_t \sqrt{\widehat{\text{Var}}_t(x_t, y_t)}$ is at most $C_{\text{opt}} d_{\text{eluder}} \log(eT)$, which is exactly the statement of Proposition E.4. $\qquad\square$

# F FULL PROOFS FOR THE MDP EXTENSION

This section extends our preference-based analysis from contextual bandits to Markov Decision Processes (MDPs) and provides full proofs for the finite-horizon and discounted regret bounds stated in Theorem 6.2 and Section F.1.

## MDP SETUP AND NOTATION

We consider a finite-horizon MDP

$$\mathcal{M} = (\mathcal{S}, \mathcal{A}, P, r^*, \rho, H),$$

with state space $\mathcal{S}$, action space $\mathcal{A}$, transition kernel $P$, horizon $H$, and initial state distribution $\rho$ over $\mathcal{S}$. The latent single-step reward function $r^* : \mathcal{S} \times \mathcal{A} \to [0, 1]$ is unknown but assumed realizable in our reward model class $\mathcal{R} = \{r_\theta : \theta \in \Theta\}$ as in the bandit setting.

A (possibly non-stationary) policy $\pi$ is a sequence $\pi = (\pi_h)_{h=1}^H$ with $\pi_h(\cdot \mid s) \in \Delta(\mathcal{A})$. We write $\pi_h(s) \in \mathcal{A}$ when $\pi_h$ is deterministic. For any policy $\pi$ we define the value and action-value functions in the usual way:

$$V_h^\pi(s) := \mathbb{E}_\pi\Big[\sum_{t=h}^H r^*(S_t, A_t) \;\Big|\; S_h = s\Big], \qquad Q_h^\pi(s, a) := r^*(s, a) + \mathbb{E}_{S_{h+1} \sim P(\cdot|s,a)}\big[V_{h+1}^\pi(S_{h+1})\big].$$

Let $d_h^\pi$ denote the marginal distribution of $S_h$ when $S_1 \sim \rho$ and the trajectory is generated by $P$ under policy $\pi$ for steps $1, \ldots, h - 1$. Note that $d_h^\pi$ does not depend on the episode index.

The OTD–LE algorithm maintains an ensemble of reward models $\{r_\theta : \theta \in \Theta\}$ updated from pairwise preferences using the same PAC–Bayesian machinery as in the bandit case. The environment never reveals numeric rewards; instead, in episode $e$ the algorithm uses a particle $\theta_e$ to form *pseudo-rewards*

$$\tilde{R}_{e,h} := r_{\theta_e}(S_{e,h}, A_{e,h}),$$

which enter the temporal-difference targets used to update the value or $Q$-function parameters. For example, in a value-based implementation we may use

$$Y_{e,h} := \tilde{R}_{e,h} + \gamma V_{\varphi_e, h+1}(S_{e,h+1}),$$

with $\gamma = 1$ in the finite-horizon case and $\gamma \in (0, 1)$ in the discounted case. All numeric quantities in the TD updates are therefore computed from the learned reward model, while the environment provides only preference feedback.[5] We write $\pi_e = (\pi_{e,h})_{h=1}^H$ for the (non-stationary) policy executed in episode $e$ and $\pi^* = (\pi_h^*)_{h=1}^H$ for an optimal policy for $r^*$.

The episodic MDP regret after $T$ episodes is

$$\mathrm{Regret}(T) := \sum_{e=1}^T \big(V_1^{\pi^*}(\rho) - V_1^{\pi_e}(\rho)\big),$$

which coincides with the contextual-bandit regret when $H = 1$.

**Remark F.1** (MDP extension and preference feedback). *In the MDP extension, the environment never reveals ground-truth numeric rewards. We assume a latent single-step reward function $r^* : \mathcal{S} \times \mathcal{A} \to [0, 1]$ that induces the Bradley–Terry–Luce preference model in Equation (2.1), and we fit a posterior over reward models $\{r_\theta : \theta \in \Theta\}$ from pairwise preferences exactly as in the bandit setting. In each episode $e$ and at each stage $h$, OTD–LE forms a pseudo-reward*

$$\tilde{R}_{e,h} := r_{\theta_e}(S_{e,h}, A_{e,h}),$$

*where $\theta_e$ is the particle used to act in episode $e$, and constructs TD targets (for value- or $Q$-function updates) only from these pseudo-rewards together with next-state value estimates. In particular, TD targets only ever use pseudo-rewards from $r_\theta$; the environment is queried solely for preference feedback, not for numeric rewards.*

---

[5]This mirrors the standard "reward-model + RL" pipeline used in preference-based RL and RLHF; see the discussion in App. I.

## F.1 MDP REGRET BOUNDS

Under Assumptions 2.1-2.3 and 6.1, the O-TDLE algorithm, run for $T$ episodes, achieves a cumulative regret that satisfies, with high probability:

$$\text{Regret}(T) = \widetilde{\mathcal{O}}\left(H^2 \cdot d_{\text{eluder}} \cdot \log T\right) + \text{lower-order approximation terms}, \tag{F.1}$$

where the lower-order terms have a similar structure to the bandit case, summed over all $T \times H$ steps.

Under the same assumptions, for an infinite-horizon discounted MDP, the O-DQLE algorithm run for $T$ steps achieves a cumulative regret that satisfies, with high probability:

$$\text{Regret}(T) = \widetilde{\mathcal{O}}\left(\frac{d_{\text{eluder}}}{(1 - \gamma)^3} \cdot \log T\right) + \text{lower-order approximation terms}. \tag{F.2}$$

## F.2 PROOF FOR FINITE-HORIZON MDPs (SECTION F.1)

The proof requires adapting the regret decomposition to handle temporal dependencies. A naive application of the value-difference lemma can lead to errors compounding exponentially in the horizon $H$. To avoid this, we employ a more sophisticated policy decomposition technique.

**Step 1: Regret decomposition via hybrid policies.** Fix an episode $e$ and write $\pi_e = (\pi_{e,h})_{h=1}^H$ for the policy executed by OTD–LE and $\pi^* = (\pi_h^*)_{h=1}^H$ for an optimal policy. For $h = 1, \ldots, H+1$ define the hybrid policies $\pi^{(h)}$ by

$$\pi_t^{(h)}(s) := \begin{cases} \pi_{e,t}(s), & t < h, \\ \pi_t^*(s), & t \geq h, \end{cases} \quad t = 1, \ldots, H, \; s \in \mathcal{S}.$$

Thus $\pi^{(1)} = \pi^*$ (all steps optimal) and $\pi^{(H+1)} = \pi_e$ (all steps follow the learned policy). By telescoping we obtain

$$V_1^{\pi^*}(\rho) - V_1^{\pi_e}(\rho) = \sum_{h=1}^H \left(V_1^{\pi^{(h)}}(\rho) - V_1^{\pi^{(h+1)}}(\rho)\right). \tag{F.3}$$

For each $h$, the three policies $\pi_e, \pi^{(h)}, \pi^{(h+1)}$ agree on steps $1, \ldots, h-1$, so they induce the same state distribution $d_h^{\pi_e}$ at step $h$. Moreover $\pi^{(h)}$ and $\pi^{(h+1)}$ both follow $\pi^*$ from step $h+1$ onward, so their $Q$-functions at step $h$ coincide with $Q_h^{\pi^*}$. Applying the standard finite-horizon performance-difference lemma with these observations yields, for every $h \in \{1, \ldots, H\}$,

$$V_1^{\pi^{(h)}}(\rho) - V_1^{\pi^{(h+1)}}(\rho) = \mathbb{E}_{S_{e,h} \sim d_h^{\pi_e}}\left[Q_h^{\pi^*}\left(S_{e,h}, \pi_h^*(S_{e,h})\right) - Q_h^{\pi^*}\left(S_{e,h}, \pi_{e,h}(S_{e,h})\right)\right]. \tag{F.4}$$

Equations equation F.3 and equation F.4 reduce the regret comparison between $\pi_e$ and $\pi^*$ to a sum of $H$ single-step advantage terms, one for each stage $h$.

**Step 2: Bounding the single-step deviations.** For a fixed episode $e$ and stage $h$, define the instantaneous MDP regret

$$\Delta_{e,h} := Q_h^{\pi^*}\left(S_{e,h}, \pi_h^*(S_{e,h})\right) - Q_h^{\pi^*}\left(S_{e,h}, \pi_{e,h}(S_{e,h})\right).$$

By equation F.4 we have $V_1^{\pi^{(h)}}(\rho) - V_1^{\pi^{(h+1)}}(\rho) = \mathbb{E}[\Delta_{e,h}]$. Unrolling the Bellman recursion shows that $\Delta_{e,h}$ is a bounded linear functional of the per-step latent reward function $r^*$ along the suffix of the trajectory, so it can be written as the difference of two evaluations of $r_\theta$ at an $(\mathcal{S} \times \mathcal{A})$-valued input. Consequently, the PAC–Bayesian posterior control, variance–information lemma, and cumulative squared-width bound developed for the contextual bandit setting apply to each pair $(e, h)$ with the same eluder dimension $d_{\text{eluder}}$.

On the high-probability optimism event from Theorem E.4, the same argument as in the bandit case yields

$$\Delta_{e,h} \leq \kappa_{e,h}\sqrt{V_{e,h}} + (\text{finite-ensemble, discretization, and stochastic-gradient terms}),$$

where $V_{e,h}$ is the posterior predictive variance of the relevant logit difference at $(S_{e,h}, \pi_{e,h}(S_{e,h}))$, and $\kappa_{e,h}$ is an exploration coefficient of order $\log(TH)$. Summing this inequality over $e = 1, \ldots, T$ and $h = 1, \ldots, H$ and applying the variance–information and cumulative squared-width bounds from Sections D.2 and E.2 gives a leading exploration term of order $\widetilde{\mathcal{O}}(H^2 d_{\mathrm{eluder}} \log T)$; the extra factor $H$ comes from the hybrid-policy decomposition equation F.3.

**Step 3: Bounding Approximation Errors.** The approximation errors from discretization, finite ensembles, and stochastic gradients are summed over all $T \times H$ steps. The martingale concentration arguments still apply, leading to lower-order terms of the form $\widetilde{\mathcal{O}}(\sqrt{TH(\cdot)})$. With appropriate scheduling of $N_e$ and $B_e$, these can be controlled. $\qquad\square$

### F.3 Proof for discounted MDPs (Section F.1)

We now consider an infinite-horizon $\gamma$-discounted MDP with the same state and action spaces $(\mathcal{S}, \mathcal{A})$ and latent reward model $r^* : \mathcal{S} \times \mathcal{A} \to [0, 1]$. For a policy $\pi$ and initial distribution $\rho$ we define

$$V^\pi(\rho) := \mathbb{E}_\pi\left[\sum_{t=0}^{\infty} \gamma^t r^*(S_t, A_t) \mid S_0 \sim \rho\right].$$

The $\gamma$-discounted state-occupancy measure of $\pi$ is

$$d^\pi(s) := (1 - \gamma) \sum_{t=0}^{\infty} \gamma^t \Pr_\pi(S_t = s \mid S_0 \sim \rho), \qquad s \in \mathcal{S}.$$

This is a probability distribution on $\mathcal{S}$. Let $Q^\pi(s, a)$ denote the usual $\gamma$-discounted action-value function of $\pi$. The performance-difference lemma for discounted MDPs then states that for any two policies $\pi$ and $\pi'$,

$$V^{\pi'}(\rho) - V^\pi(\rho) = \frac{1}{1 - \gamma} \mathbb{E}_{s \sim d^\pi}\left[Q^{\pi'}(s, \pi'(s)) - Q^{\pi'}(s, \pi(s))\right]. \tag{F.5}$$

In our setting, the algorithm produces a sequence of policies $\pi_1, \pi_2, \ldots, \pi_T$ via OTD–LE using pseudo-rewards $r_{\theta_t}(s, a)$ from the learned reward model, exactly as described in the finite-horizon case; the environment again supplies only preference feedback. Define the instantaneous regret at round $t$ by

$$\Delta_t := Q^{\pi^*}(S_t, \pi^*(S_t)) - Q^{\pi^*}(S_t, \pi_t(S_t)),$$

so that equation F.5 implies

$$V^{\pi^*}(\rho) - V^{\pi_t}(\rho) = \frac{1}{1 - \gamma} \mathbb{E}[\Delta_t].$$

The same PAC–Bayesian, variance–information, and cumulative squared-width analysis as in the contextual bandit setting shows that, on a high-probability event,

$$\Delta_t \leq \kappa_t \sqrt{V_t} + \text{(finite-ensemble, discretization, and stochastic-gradient terms)},$$

where $V_t$ is the posterior predictive variance of the queried logit difference at round $t$. Summing over $t = 1, \ldots, T$ and using the variance–information lemma together with the information–eluder bound from Section D.2 yields

$$\sum_{t=1}^{T} \mathbb{E}[\Delta_t] = \widetilde{\mathcal{O}}(d_{\mathrm{eluder}} \log T).$$

Combining this with the factor $1/(1 - \gamma)$ from equation F.5 and the finite-ensemble / discretization / stochastic-gradient bounds from Section E.2 gives the discounted MDP regret bound stated in Section F.1, with the leading term of order $\widetilde{\mathcal{O}}\big(d_{\mathrm{eluder}}(1-\gamma)^{-2} \log T\big)$ and lower-order approximation terms analogous to the contextual bandit case. $\qquad\square$

# G  Implementation Details and Additional Pseudocode

This section provides the computation cost discussion of OLE, necessary details of pseudocode for the proposed algorithms and a discussion of hyperparameter schedules that achieve the optimal regret rates.

## G.1  Computational Cost of OLE.

At each round $t$, Algorithm 2 performs a single projected SGLD update for each of the $N_t$ particles:

$$\tilde{\theta}_{t+1}^{(i)} = \theta_t^{(i)} - \eta_t \widehat{\nabla} J_{\text{PAC}}(\theta_t^{(i)}) + \sqrt{2\eta_t \beta}\, \xi_t^{(i)}, \qquad \theta_{t+1}^{(i)} = \Pi_\Theta\big(\tilde{\theta}_{t+1}^{(i)}\big).$$

The stochastic gradient $\widehat{\nabla} J_{\text{PAC}}(\theta_t^{(i)})$ is computed on a mini-batch $B_t$ of size $|B_t|$ from the replay buffer $D_t$, so its cost is $O(|B_t| \cdot \dim(\theta))$, exactly as in a standard SGD update on the same model. The additional Gaussian-noise and projection operations are $O(\dim(\theta))$ and therefore negligible compared to the gradient computation. Hence the overall per-round complexity of OLE is

$$O\big(N_t\,|B_t| \cdot \dim(\theta)\big),$$

and the total cost up to horizon $T$ is $O(N\,|B|\,T \cdot \dim(\theta))$ when $N := \sup_t N_t$ and $|B| := \sup_t |B_t|$. In our regret analysis, $N_t$ and $|B_t|$ are taken to be fixed (or at most polylogarithmic in $T$); the corresponding approximation errors appear only in the lower-order "Finite Ensemble" and "Stochastic Gradient" terms of Theorem 5.1 and do not affect the leading dependence on $T$. Thus, OLE is computationally comparable to training a small ensemble of reward models with mini-batch SGD, and all operations are polynomial-time in the problem parameters.

## G.2  Complete Pseudocode

The following algorithms formalize the procedures analyzed in this paper. Algorithm 2 provides the generic template, Algorithm 4 and Algorithm 3 specifies the contextual bandit variant online contextual bandit variant respectively, and Algorithm 5 details the extension to MDPs using temporal-difference learning.

---

**Algorithm 2:** Optimistic Langevin Ensemble (OLE): Generic Template

**Input:** Prior $\Pi_0$; step sizes $\{\eta_t\}$; ensemble sizes $\{N_t\}$; batch sizes $\{B_t\}$; optimism schedule $\{\kappa_t\}$

1 **for** $t = 1, 2, \ldots, T$ **do**
2     Observe context $x_t$;
    // Optimistic Selection
3     Compute ensemble mean $\hat{r}_t(x_t, y)$ and variance $\widehat{\text{Var}}_t(x_t, y)$ for all $y \in \mathcal{Y}$;
4     Construct optimistic index: $I_t(x_t, y) \leftarrow \hat{r}_t(x_t, y) + \kappa_t \sqrt{\widehat{\text{Var}}_t(x_t, y)}$;
5     Select action pair $(y_t^{(w)}, y_t^{(\ell)})$ based on maximizing information gain using $\{I_t(x_t, y)\}_{y \in \mathcal{Y}}$;
6     Receive preference feedback, forming data batch $\mathcal{D}_t$;
    // Posterior Update (SGLD)
7     Compute mini-batch gradient $\widehat{\nabla}_t$ of $J_{\text{PAC}}(\theta) = \hat{L}_{\mathcal{D}_t}(\theta) + \beta D_{\text{KL}}(\delta_\theta \| \Pi_{t-1})$;
8     **for** $i = 1, \ldots, N_t$ **do**
9         Draw Gaussian noise $\xi_t^{(i)} \sim \mathcal{N}(0, I)$;
10         $\theta_{t+1}^{(i)} \leftarrow \theta_t^{(i)} - \eta_t \widehat{\nabla}_t J_{\text{PAC}}(\theta_t^{(i)}) + \sqrt{2\eta_t \beta}\, \xi_t^{(i)}$;
11         $\theta_{t+1}^{(i)} \leftarrow \text{Proj}_\Theta\big(\tilde{\theta}_{t+1}^{(i)}\big)$;

---

---

**Algorithm 3:** Optimistic Thompson Sampling with Langevin Ensembles (O-TSLE)

---

**Input:** Prior $\Pi_0$, step size $\eta$, particles $N_t$, batch size $B_t$, optimism schedule $\kappa_t$.

1 **for** $t = 1, 2, \ldots, T$ **do**

2     Draw $\{\theta_t^{(i)}\}_{i=1}^{N_t}$ by 1 SGLD step from $\Pi_{t-1}$ using $B_t$ samples;

3     Compute predictive mean $\hat{r}_t(y)$ and uncertainty $\hat{\sigma}_t(y)$ over candidates $y \in \mathcal{Y}$;

4     Select action $y_t \in \arg\max_y \ \hat{r}_t(y) + \kappa_t \hat{\sigma}_t(y)$;

5     Observe (pairwise) feedback at $y_t$ and update posterior to $\Pi_t$ (PAC-Bayes loss);

---

---

**Algorithm 4: Optimistic Langevin Ensemble (OLE)** — Contextual Bandit Variant (O-TSLE)

---

**Input:** Prior $\Pi_0$; step sizes $\{\eta_t\}$; ensemble sizes $\{N_t\}$; batch sizes $\{B_t\}$; optimism schedule $\{\kappa_t\}$

1 **for** $t = 1, 2, \ldots, T$ **do**

2     Observe context $x_t$;

    // Optimistic Selection

3     Compute ensemble mean and variance for all $y \in \mathcal{Y}$:

4     $\hat{r}_t(x_t, y) \leftarrow \frac{1}{N_t} \sum_{i=1}^{N_t} r_{\theta_t^{(i)}}(x_t, y)$;

5     $\widehat{\mathrm{Var}}_t(x_t, y) \leftarrow \frac{1}{N_t - 1} \sum_{i=1}^{N_t} (r_{\theta_t^{(i)}}(x_t, y) - \hat{r}_t(x_t, y))^2$;

6     Construct optimistic index: $I_t(x_t, y) \leftarrow \hat{r}_t(x_t, y) + \kappa_t \sqrt{\widehat{\mathrm{Var}}_t(x_t, y)}$;

7     Select action pair $(y_t^{(w)}, y_t^{(\ell)})$ to query, based on maximizing information gain using $\{I_t(x_t, y)\}_{y \in \mathcal{Y}}$;

8     Receive preference feedback for the selected pair, forming data batch $\mathcal{D}_t$;

    // Posterior Update

9     Compute mini-batch gradient $\widehat{\nabla}_t$ of $J_{\mathrm{PAC}}$ using $\mathcal{D}_t$ (batch size $B_t$);

10     **for** $i = 1, \ldots, N_t$ **do**

11        Draw Gaussian noise $\xi_t^{(i)} \sim \mathcal{N}(0, I)$;

12        **Langevin step:** $\theta_{t+1}^{(i)} \leftarrow \theta_t^{(i)} - \eta_t \widehat{\nabla}_t J_{\mathrm{PAC}}(\theta_t^{(i)}) + \sqrt{2\eta_t \beta}\, \xi_t^{(i)}$;

13        $\theta_{t+1}^{(i)} \leftarrow \mathrm{Proj}_\Theta\left(\tilde{\theta}_{t+1}^{(i)}\right)$;

---

---

**Algorithm 5:** Optimistic TD with Langevin Ensembles (O-TDLE) for MDPs

---

**Input:** Prior $\Pi_0$ on Q-function parameters; step sizes $\{\eta_e\}$; ensemble sizes $\{N_e\}$; batch sizes $\{B_e\}$; optimism schedule $\{\kappa_h\}$

1 **for** *episode* $e = 1, 2, \ldots, T$ **do**

2     Initialize state $s_1$;

3     **for** *step* $h = 1, 2, \ldots, H$ **do**

       // Optimistic Action Selection

4        Compute ensemble mean $\hat{Q}_{e,h}(s_h, a)$ and variance $\widehat{\mathrm{Var}}_{e,h}(s_h, a)$ for all $a \in \mathcal{A}$;

5        Select action $a_h = \arg\max_{a \in \mathcal{A}} \left( \hat{Q}_{e,h}(s_h, a) + \kappa_h \sqrt{\widehat{\mathrm{Var}}_{e,h}(s_h, a)} \right)$;

6        Execute $a_h$, observe next state $s_{h+1}$ and collect preference data for the transition;

7        Form a pseudo-reward $\tilde{R}_{e,h} \leftarrow r_{\theta_e}(s_h, a_h)$ using the current reward model ensemble;

    // Posterior Update (after episode)

8     Form a batch of transitions and preferences $\mathcal{D}_e$ from the episode;

9     Compute TD targets $y_h = r(s_h, a_h) + \gamma \max_{a'} \hat{Q}_{e,H}(s_{h+1}, a')$ (using ensemble mean);

10     Compute mini-batch gradient $\widehat{\nabla}_e$ of a TD-based loss on $\mathcal{D}_e$ regularized by $D_{\mathrm{KL}}(\cdot \| \Pi_{e-1})$;

11     Update all particles $\{\theta_e^{(i)}\}$ to $\{\theta_{e+1}^{(i)}\}$ using one or more SGLD steps with gradient $\widehat{\nabla}_e$;

---

### G.3 DISCUSSION OF HYPERPARAMETER SCHEDULES

Corollary 5.3 states that if the algorithmic parameters are scheduled appropriately, the lower-order approximation error terms in the regret bound become asymptotically negligible, leaving a purely logarithmic regret. Here we specify schedules that achieve this.

- **Step Size** ($\eta_t$)**:** To ensure the cumulative discretization error $\sum \eta_t$ remains bounded, a decreasing step size schedule is required. A standard choice is $\eta_t = \eta_0/t$ or $\eta_t = \eta_0/\sqrt{t}$. With such schedules, the sum converges or grows slower than any linear function, making the $\widetilde{\mathcal{O}}(\sum \eta_t)$ term sub-leading.

- **Ensemble Size** ($N_t$) **and Batch Size** ($B_t$)**:** To control the finite-ensemble and stochastic gradient errors, whose cumulative sums scale as $\widetilde{\mathcal{O}}(\sqrt{\sum 1/N_t})$ and $\widetilde{\mathcal{O}}(\sqrt{\sum 1/B_t})$ respectively (assuming bounded variances), we need the sums $\sum 1/N_t$ and $\sum 1/B_t$ to be bounded. This can be achieved by increasing $N_t$ and $B_t$ over time. For example, setting $N_t = \lceil N_0 \log(t+1) \rceil$ and $B_t = \lceil B_0 \log(t+1) \rceil$ would suffice. A practical alternative is an episodic schedule where $N_t$ and $B_t$ are increased (e.g., doubled) at the start of geometrically spaced episodes. This ensures the approximation errors are effectively "paid for" by the logarithmic exploration term.

These schedules demonstrate that our theory provides an asymptotic guarantee, and offers concrete, practical guidance for algorithm design, directly connecting the theoretical results to the desired performance outcome.

## H EXPERIMENT

**Experiment Settings.** **(1) Datasets** . We evaluate our methods on the grade school math dataset *GSM8K* Cobbe et al. (2021), a collection of 8.5K high-quality, linguistically diverse word problems that test basic mathematical skills requiring multi-step reasoning. In addition, we adopt zero-shot prompts and rule-based evaluators to automatically assess the performance of LLMs. **(2) Backbones.** We use *Qwen2.5-1.5B-Instruct*, *Qwen2.5-3B-Instruct* Bai et al. (2023) as language model backbones. **(3) Baselines.** Among widely adopted on-policy RL methods, **GRPO** Shao et al. (2024), **DAPO** Yu et al. (2025) and **GPG** Chu et al. (2025) share a common framework derived from PPO Schulman et al. (2017). Instead of using generalized advantage estimation (GAE), they adopt a group-wise relative estimation strategy. Concretely, a policy $\pi_\theta$ generates a group $G_s$ of candidate rollouts for a given input, and the model is optimized to maximize the expected group-level reward. We combine these three baselines with OLE to test our performance. **(4) Evaluation.** In the experimental data processing phase, we strictly adhere to the original training-test set splits provided for the GSM8K dataset to ensure the reproducibility of results and comparability with prior studies. Specifically, for the original training set of each dataset, we further employ stratified random sampling to partition it into a training subset and a validation subset at an 80%:20% ratio. **(6) Hyper-parameters Details.** The maximum input sequence length is set to 512 tokens, and the maximum number of generated tokens is 2048. The learning rate is $1 \times 10^{-6}$. The number of rollouts $G_s$ is set to 4. The ole threshold percent is set to 0.8. The rank of lora $r$ is set to 16. For DAPO specifically, we set $\epsilon_{\text{low}} = 0.2$ and $\epsilon_{\text{high}} = 0.28$, with the number of resampling steps set to 3. **(6) Implement Details.** To ensure reproducibility, all experiments are implemented in PyTorch with Python 3.11. Training and inference are conducted on 8×A800-80G GPUs. All on-policy RL baselines are implemented using the VeRL framework Sheng et al. (2025). All baselines are carefully re-implemented and hyperparameter-tuned to ensure fair comparisons. Code is available at `https://anonymous.4open.science/r/ICLR_OLE-B243`.

From the experimental results, we observe a consistent pattern across different model sizes (Qwen2.5-Instruct-1.5B and 3B) and optimization paradigms (GRPO, DAPO, GPG): after introduc-

Table 7: GSM8K results on Qwen2.5-Instruct models with OLE performance gain.

| Model | Method | Base (↑) | Drops (↑) | Base+OLE (↑) | Drops (↑) | Performance Gain (↑) |
|-------|--------|----------|-----------|--------------|-----------|----------------------|
| 1.5B | GRPO | 0.596 | 0 | **0.612** | 3944 | 2.69% |
|  | DAPO | 0.497 | 0 | **0.596** | 4344 | 19.9% |
|  | GPG | 0.596 | 0 | **0.613** | 4032 | 2.85% |
| 3B | GRPO | 0.667 | 0 | **0.704** | 5964 | 5.55% |
|  | DAPO | 0.707 | 0 | **0.712** | 5080 | 0.71% |
|  | GPG | 0.635 | 0 | **0.680** | 6408 | 7.09% |

Table 8: GSM8K results of different training schedules for Qwen2.5-3B under GRPO.

| Method | Acc (↑) | Drops (↑) | Performance Gain (↑) |
|--------|---------|-----------|----------------------|
| GRPO Only | 0.667 | 0 | – |
| GRPO + OLE (full steps) | 0.704 | 5964 | +5.55% |
| 20-step GRPO → OLE-enabled GRPO | **0.722** | 5820 | **+8.25%** |

ing OLE, all method–model combinations achieve positive performance gains on GSM8K, while simultaneously discarding a substantial number of training samples.

OLE works by estimating the *marginal contribution* of each sample to the overall optimization objective and selectively dropping those that provide limited benefit or introduce training noise. This allows the training process to concentrate gradient updates on **higher-value samples** under the same compute budget. On the efficiency side, the number of samples participating in backpropagation is significantly reduced (e.g., thousands of samples are dropped for each configuration), which effectively increases the number of informative updates per unit time. On the effectiveness side, we see consistent improvements across both small and large models. The 1.5B+DAPO setting achieves the largest relative gain of 19.9%, indicating that removing low-value samples is particularly beneficial when the base optimizer is weaker or the model capacity is more constrained. Notably, even the strong 3B+GRPO configuration now benefits from OLE, with a 5.55% relative improvement, showing that sample filtering can still enhance performance in already competitive regimes.

Overall, these results support our theoretical hypothesis: **by estimating sample value online and dynamically discarding low-gain examples, OLE increases the "purity" of the training signal, leading to both higher training efficiency and better final model performance, without increasing computational cost.**

**Remark H.1** (Empirical validation and connection to rejection sampling in RL). *The GSM8K experiments with Qwen2.5–Instruct backbones and three on-policy RL baselines (GRPO, DAPO, GPG) exhibit exactly the qualitative behavior predicted by our theory. Across all configurations in the table, plugging OLE on top of the base RL optimizer yields consistent performance gains in the* Base+OLE *column, while the* Drops *column shows that thousands of training updates are skipped by the OLE filtering rule. This is consistent with the regret decomposition in Theorem 5.1: the leading $\tilde{\mathcal{O}}(d_{\mathrm{eluder}} \log T)$ exploration term depends on how quickly the optimistic posterior concentrates, not on using* every *on-policy sample. Discarding low-gain updates primarily shrinks the lower-order stochastic gradient and approximation terms without changing the asymptotic rate, so we expect to see* better *empirical performance at a comparable computational budget, which is exactly what the table reports.*

*From an RL perspective, the OLE filter can be interpreted as a principled form of* rejection sampling *over on-policy rollouts. For each input, the base policy and RL optimizer (GRPO, DAPO, or GPG) generate a small group of candidate responses. OLE then evaluates the* marginal contribution *of each candidate to the PAC-Bayesian objective and keeps only those above the OLE threshold, while rejecting the rest. This accept/reject step plays the same structural role as the heuristic rejection sampling used in many practical RLHF pipelines (e.g., discarding low-reward or low-score trajectories), but here the acceptance rule is derived directly from the PAC-Bayes/Wasserstein gradient-*

*flow analysis rather than chosen ad hoc. The fact that Base+OLE dominates the base RL methods in all settings, despite the substantial number of rejected samples reported in the* Drops *column, empirically corroborates our theoretical claim that optimally accepting only the most informative preference updates can improve both generalization and sample efficiency in preference-based RL.*

**Optimism-Schedule Experiment.** To further test whether OLE serves as an implicit optimism mechanism, we run an additional experiment on Qwen2.5-3B + GRPO. Instead of enabling OLE from the start, we first train with standard GRPO for 20 steps, allowing the model to *exploit* the data uniformly and stabilize its initial representations. We then activate GRPO+OLE for the remaining steps. In this phase, OLE prioritizes higher-uncertainty samples that provide larger information gain, effectively shifting the training dynamics toward *exploration*.

This staged strategy achieves the best accuracy of **0.722**, outperforming both pure GRPO and full-length GRPO+OLE. The result indicates that activating OLE later in training allows the model to explore informative, high-uncertainty samples more effectively once a stable baseline has been formed. Empirically, this supports the design in Algorithm 2, where both the number of particles $N$ and the optimism coefficient $\kappa_t$ are gradually increased to achieve a practical and effective balance between early exploitation and later exploration.

# I   LOWER BOUND AND OPTIMALITY

**Remark I.1** (On lower bounds and optimality in $T$). *We demonstrate with a Proposition I.2 showing that even in the non-contextual, finite-action special case of our model, with the same Bradley–Terry–Luce (BTL) preference structure and bounded rewards as in Theorem 5.1, any uniformly good algorithm must incur expected regret at least of order $\log T$. More precisely, for each fixed instance with positive gaps $\Delta_y = r^*(y^\star) - r^*(y) > 0$ one has $\mathbb{E}[\mathrm{Regret}(T)] \geq c_{\mathrm{low}}(r^*) \log T$ for all sufficiently large $T$, and on a gap-separated subclass with minimum gap $\Delta_{\min} > 0$ there exists a constant $c_{\mathrm{low}}(\Delta_{\min}) > 0$ such that*

$$\sup_{\text{instances with gaps} \geq \Delta_{\min}} \mathbb{E}[\mathrm{Regret}(T)] \geq c_{\mathrm{low}}(\Delta_{\min}) \log T \quad \text{for all sufficiently large } T.$$

*Within this structural class, the dependence on the horizon $T$ in Theorem 5.1 can therefore not be improved below logarithmic order: up to absolute constants, polylogarithmic factors, and the eluder-dimension factor $d_{\mathrm{eluder}}$, our upper bound is optimal in its $T$-dependence.*

At the same time, our result is fully compatible with the well-known $\Omega(d\sqrt{T})$ minimax lower bounds for *contextual dueling bandits*, such as the linear setting studied by Bengs et al. (2022). In that literature the learner selects a *pair* of actions $(a_t^{(1)}, a_t^{(2)})$ at each round, observes a single noisy comparison between them, and performance is measured by a dueling-regret notion (weak or strong) over such pairs—roughly, how often the chosen pair loses or fails to beat the best arm. The $\Omega(d\sqrt{T})$ lower bound is minimax for this pair-action, dueling-regret problem. By contrast, in our setting the algorithm selects a *single* action $y_t$ at each round (or per state in the MDP), may query preferences involving $y_t$, and regret is the standard single-action cumulative regret $\mathrm{Regret}(T) = \sum_{t=1}^{T} (r^*(x_t, y^*(x_t)) - r^*(x_t, y_t))$. The contextual dueling lower bound does not provide a lower bound for $\mathrm{Regret}(T)$ in this single-action setting, just as Proposition I.2 does not make any claim about dueling regret over pairs of actions. The two results address different minimax problems and can hold simultaneously without contradiction.

In summary, Theorem 5.1 should be read as a *uniform fast-rate* $\widetilde{\mathcal{O}}(d_{\mathrm{eluder}} \log T)$ bound for single-action regret under our structural assumptions (realizability, boundedness, Lipschitz continuity, finite eluder dimension, and BTL preferences), and Proposition I.2 shows that its logarithmic dependence on $T$ is essentially optimal within this class.

**Proposition I.2** (Logarithmic lower bound in the BTL preference setting). *Consider the non-contextual special case of our model with a finite action set $\mathcal{Y} = \{1, \ldots, K\}$ and Bradley–Terry–Luce preferences generated from a latent reward vector $r^* \in [0,1]^K$. At each round $t$, the algorithm*

*chooses a single action $y_t \in \mathcal{Y}$ and observes one bit of preference feedback comparing $y_t$ to a fixed baseline action $y_0 \in \mathcal{Y}$, according to*

$$\mathbb{P}(y_t \succ y_0 \mid y_t) = \sigma\big(r^*(y_t) - r^*(y_0)\big),$$

*where $\sigma$ is the logistic link. Regret is the standard single-action regret*

$$\text{Regret}(T) = \sum_{t=1}^{T} \big(r^*(y^\star) - r^*(y_t)\big), \quad y^\star \in \arg\max_{y \in \mathcal{Y}} r^*(y).$$

*Assume there is a unique optimal action $y^\star$ and that all gaps $\Delta_y := r^*(y^\star) - r^*(y)$ for $y \neq y^\star$ are strictly positive. Let $p_y := \sigma\big(r^*(y) - r^*(y_0)\big)$ and let $\text{KL}(\cdot \| \cdot)$ denote the Bernoulli Kullback–Leibler divergence. Then, for any (possibly randomized) algorithm $\mathcal{A}$ that is uniformly good in the sense of Lai and Robbins (1985),*

$$\liminf_{T \to \infty} \frac{\mathbb{E}_{r^*}\big[\text{Regret}_{\mathcal{A}}(T)\big]}{\log T} \geq \sum_{y \neq y^\star} \frac{\Delta_y}{\text{KL}(p_y \| p_{y^\star})}. \tag{I.1}$$

*In particular, since $\{r^*(y)\}_y$ are bounded and the logistic logit range is therefore bounded, there exists a constant $C_{\text{KL}} < \infty$ such that $\text{KL}(p_y \| p_{y^\star}) \leq C_{\text{KL}} \Delta_y^2$ for all $y \neq y^\star$, and hence for any fixed instance there is a $c_{\text{low}}(r^*) > 0$ such that*

$$\mathbb{E}_{r^*}\big[\text{Regret}_{\mathcal{A}}(T)\big] \geq c_{\text{low}}(r^*) \log T \qquad \text{for all sufficiently large } T.$$

*Proof.* Fix an instance specified by a latent reward vector $r^* \in [0,1]^K$ and a baseline action $y_0 \in \mathcal{Y}$. Recall that by assumption there is a unique optimal action $y^\star \in \arg\max_{y \in \mathcal{Y}} r^*(y)$ and that the gaps $\Delta_y := r^*(y^\star) - r^*(y)$ are strictly positive for all $y \neq y^\star$.

**Step 1: Reduction to a Bernoulli bandit.** At each round $t$, the algorithm chooses a single action $y_t \in \mathcal{Y}$. The feedback is one bit indicating whether $y_t$ is preferred to the fixed baseline $y_0$; under the BTL model equation 2.1 this bit is

$$Z_t = \mathbf{1}\{y_t \succ y_0\}, \quad Z_t \mid (y_t = y) \sim \text{Bernoulli}(p_y),$$

with

$$p_y := \sigma\big(r^*(y) - r^*(y_0)\big), \qquad \sigma(z) = (1 + e^{-z})^{-1}.$$

Thus, from the viewpoint of the learning algorithm, this non-contextual preference problem is exactly a $K$-armed stochastic bandit with Bernoulli rewards $\{p_y\}_{y=1}^{K}$: on each round the algorithm chooses an arm $y$ and observes an independent Bernoulli sample with mean $p_y$.

Let $N_y(T) := \sum_{t=1}^{T} \mathbf{1}\{y_t = y\}$ denote the number of times arm $y$ is played up to time $T$. By definition of the regret in the proposition,

$$\text{Regret}(T) = \sum_{t=1}^{T} \big(r^*(y^\star) - r^*(y_t)\big) = \sum_{y \neq y^\star} \Delta_y N_y(T).$$

Taking expectations under the fixed instance $r^*$ gives

$$\mathbb{E}_{r^*}\big[\text{Regret}(T)\big] = \sum_{y \neq y^\star} \Delta_y \mathbb{E}_{r^*}[N_y(T)]. \tag{I.2}$$

**Step 2: Applying the Lai–Robbins lower bound.** The family of Bernoulli distributions $\{\text{Bernoulli}(p_y) : y \in \mathcal{Y}\}$ is a one-parameter exponential family, with canonical parameter $\theta_y = \log\big(p_y/(1 - p_y)\big)$ and mean $p_y$. The classical theorem of Lai and Robbins (Lai-Robbins bound) Lai & Robbins (1985) applies to this setting. In the notation of that theorem, an algorithm is *uniformly good* if, for every bandit instance, its regret grows slower than any power of $T$: for all $\alpha > 0$ and all arms $y$,

$$\mathbb{E}[N_y(T)] = o(T^\alpha) \quad \text{as } T \to \infty.$$

Under this condition, Lai and Robbins show that for each suboptimal arm $y \neq y^\star$,

$$\liminf_{T \to \infty} \frac{\mathbb{E}_{r^*}[N_y(T)]}{\log T} \geq \frac{1}{\mathrm{KL}(\nu_y \| \nu_{y^\star})}, \tag{I.3}$$

where $\nu_y$ and $\nu_{y^\star}$ are the reward distributions of arms $y$ and $y^\star$. In the Bernoulli case, $\nu_y$ is fully determined by $p_y$, and $\mathrm{KL}(\nu_y \| \nu_{y^\star}) = \mathrm{KL}(p_y \| p_{y^\star})$ is the usual Bernoulli Kullback–Leibler divergence.

We now combine equation I.2 and equation I.3. For each $T$,

$$\frac{\mathbb{E}_{r^*}[\mathrm{Regret}(T)]}{\log T} = \sum_{y \neq y^\star} \Delta_y \frac{\mathbb{E}_{r^*}[N_y(T)]}{\log T}.$$

Because the sum is finite (over the $K - 1$ suboptimal arms) and all terms in the sum are nonnegative, we may pass the $\liminf$ through the sum:

$$\liminf_{T \to \infty} \frac{\mathbb{E}_{r^*}[\mathrm{Regret}(T)]}{\log T} = \liminf_{T \to \infty} \sum_{y \neq y^\star} \Delta_y \frac{\mathbb{E}_{r^*}[N_y(T)]}{\log T}$$

$$\geq \sum_{y \neq y^\star} \Delta_y \liminf_{T \to \infty} \frac{\mathbb{E}_{r^*}[N_y(T)]}{\log T}$$

$$\geq \sum_{y \neq y^\star} \frac{\Delta_y}{\mathrm{KL}(p_y \| p_{y^\star})},$$

which is exactly the bound stated in equation I.1.

**Step 3: Positivity and logarithmic growth.** We now argue that the right-hand side is strictly positive and finite, which yields the claimed logarithmic growth rate.

First, because $r^*(y^\star) > r^*(y)$ for all $y \neq y^\star$, we have $\Delta_y > 0$ for all $y \neq y^\star$. The BTL link $\sigma$ is strictly increasing, so $p_y < p_{y^\star}$ for each $y \neq y^\star$, and therefore $\mathrm{KL}(p_y \| p_{y^\star}) > 0$ for all $y \neq y^\star$.

Second, the rewards are bounded in $[0, 1]$, so for any $y$ we have $r^*(y) - r^*(y_0) \in [-1, 1]$ and hence $p_y = \sigma(r^*(y) - r^*(y_0))$ lies in the compact interval $[\sigma(-1), \sigma(1)] \subset (0, 1)$. Thus the pairs $(p_y, p_{y^\star})$ all belong to the compact set

$$\big[\sigma(-1), \sigma(1)\big]^2 \subset (0, 1)^2.$$

The function

$$F(p, q) := \begin{cases} \dfrac{\mathrm{KL}(p \| q)}{(p - q)^2}, & p \neq q, \\ \lim_{u \to p} \dfrac{\mathrm{KL}(u \| p)}{(u - p)^2}, & p = q, \end{cases}$$

is continuous and finite on $(0, 1)^2$, and hence on the compact subset $[\sigma(-1), \sigma(1)]^2$. In particular, there exists a finite constant $C_{\mathrm{KL}} < \infty$ such that

$$\mathrm{KL}(p_y \| p_{y^\star}) \leq C_{\mathrm{KL}} (p_y - p_{y^\star})^2 \qquad \text{for all } y.$$

Since $\sigma$ is smooth and strictly monotone on $[-1, 1]$, the mean-value theorem gives, for each $y \neq y^\star$,

$$p_{y^\star} - p_y = \sigma\big(r^*(y^\star) - r^*(y_0)\big) - \sigma\big(r^*(y) - r^*(y_0)\big) = \sigma'(\xi_y) \Delta_y$$

for some $\xi_y$ between $r^*(y^\star) - r^*(y_0)$ and $r^*(y) - r^*(y_0)$. The derivative $\sigma'(z) = \sigma(z)(1 - \sigma(z))$ is strictly positive and continuous on $\mathbb{R}$, so on the compact interval $[-1, 1]$ it attains a positive minimum $\lambda_{\min} > 0$ and a finite maximum $\lambda_{\max} < \infty$. Hence

$$\lambda_{\min} \Delta_y \leq p_{y^\star} - p_y \leq \lambda_{\max} \Delta_y \qquad \text{for all } y \neq y^\star.$$

Combining the two displays, we obtain

$$\mathrm{KL}(p_y \| p_{y^\star}) \leq C_{\mathrm{KL}}(p_{y^\star} - p_y)^2 \leq C_{\mathrm{KL}} \lambda_{\max}^2 \Delta_y^2,$$

and thus

$$\frac{\Delta_y}{\mathrm{KL}(p_y \| p_{y^\star})} \geq \frac{1}{C_{\mathrm{KL}} \lambda_{\max}^2} \cdot \frac{1}{\Delta_y} > 0 \qquad \text{for each } y \neq y^\star.$$

Since there are finitely many suboptimal arms, the sum

$$L(r^*) := \sum_{y \neq y^\star} \frac{\Delta_y}{\mathrm{KL}(p_y \| p_{y^\star})}$$

is strictly positive and finite for every fixed instance $r^*$. From equation I.1 we have

$$\liminf_{T \to \infty} \frac{\mathbb{E}_{r^*}[\mathrm{Regret}(T)]}{\log T} \geq L(r^*).$$

By the definition of the $\liminf$, there exists $T_0(r^*) < \infty$ such that $\mathbb{E}_{r^*}[\mathrm{Regret}(T)]/\log T \geq \frac{1}{2}L(r^*)$ for all $T \geq T_0(r^*)$. Setting $c_{\mathrm{low}}(r^*) := \frac{1}{2}L(r^*) > 0$ yields

$$\mathbb{E}_{r^*}[\mathrm{Regret}(T)] \geq c_{\mathrm{low}}(r^*) \log T \qquad \text{for all } T \geq T_0(r^*),$$

which is the claimed logarithmic lower bound. $\qquad\square$

