# OpenReview forum: "Logarithmic Regret in Preference Learning via Optimistic PAC-Bayesian Particle Ensembles"
_ICLR.cc/2026/Conference — ICLR 2026 Conference Withdrawn Submission_

### Official Review · Reviewer_hnyH · 2025-10-22

**Soundness:** 3
**Presentation:** 1
**Contribution:** 2
**Rating:** 4
**Confidence:** 3

**Summary:**

This paper studies the dueling bandit problem and introduces a unified optimistic PAC-Bayesian framework that achieves logarithmic regret under general function approximation, even when using finite particle ensembles and noisy stochastic-gradient updates.
The main contribution lies in coupling PAC-Bayesian generalization control with concentration inequalities for stochastic dynamics and Wasserstein stability bounds for particle approximations, providing a cohesive theoretical foundation for efficient preference-based learning.

**Strengths:**

Overall, the paper is interesting as it proposes a PAC-Bayesian particle analysis framework and successfully achieves logarithmic regret.
However, the main theoretical analyses are too abbreviated, making it difficult to verify the results.
For example, in Lemma D.3, the value of the constant $C$ is not specified, and the proof is overly brief.
Moreover, in Equation (E.3), the exact value of term (II) is unclear, which makes it hard to follow the derivation.

**Weaknesses:**

**1. Presentation:** Overall, the paper requires significant improvements in presentation and clarity, particularly in the proofs and theoretical explanations.

**2. Regret bounds:** The claimed logarithmic regret appears questionable.
For instance, consider the linear dueling bandit setting, which is a special case under the bounded eluder dimension assumption. In [1], an $\Omega(d \sqrt{T})$ lower bound was established, which seems to contradict the logarithmic regret result proposed in this paper.

**3. Regret bounds in Table 1:** The comparison with Zhao et al. (2025a) may not be appropriate, since they consider a different objective—the KL-regularized regret—rather than the standard cumulative regret.
Therefore, a direct comparison to their setting is not valid.
Moreover, it appears that Russo & Van Roy (2014) established a $\tilde{O}(d\sqrt{T})$ bound, not $\tilde{O}(d\log T)$.

**4. No experiments:** Although the authors claim that this work bridges the gap between theory and practice, no experimental results are provided.
Without any empirical evidence, it is difficult to assess the practical applicability of the proposed algorithm.


[1] Bengs, Viktor, Aadirupa Saha, and Eyke Hüllermeier. "Stochastic contextual dueling bandits under linear stochastic transitivity models." International Conference on Machine Learning. PMLR, 2022.

**Questions:**

1. What is the key mechanism that enables the proposed method to achieve logarithmic regret, rather than the usual $\sqrt{T}$-type regret?

2. Could the authors provide a clear and thorough comparison with the related studies listed in Table 1, explicitly highlighting the differences in assumptions, regret definitions, and problem settings?

---

> ### Author Response · Authors · 2025-11-21
> **W2. Logarithmic regret vs. contextual dueling bandits lower bound**
>
> **Reviewer’s concern.**
>
> The reviewer notes that in the linear contextual dueling bandit setting (Bengs et al., ICML’22), there is an $\Omega(d\sqrt{T})$ **lower bound** on regret, and asks how this is compatible with our $\widetilde{\mathcal{O}}(d_{\mathrm{eluder}} \log T)$ bound under bounded eluder dimension.
>
> **Response.**
> We appreciate this careful comparison and agree that the relationship should be spelled out clearly. The key points are:
>
> 1. **Different action spaces and regret notions.**
> - Bengs et al. consider **pair‑actions**: at each round the algorithm chooses a pair of arms, and feedback is a noisy comparison between them. They study “weak” and “strong” dueling regret, defined in terms of how often the chosen pair loses (or fails to win) against the best arm.
> - Our setting is **single‑action** selection with pairwise preference feedback: at each round we choose a single action $y_t$ and observe preferences between the chosen action and a comparison action, but our regret is the standard **single‑action regret**
>    $Regret(T) = \sum_{t=1}^T l(r^* (x_t,y^* (x_t)) - r^* (x_t,y_t)),$
>   where $y^*(x_t)$ maximizes the latent reward.
> Their $\Omega(d\sqrt{T})$ lower bound applies to **weak pairwise regret** over pair‑actions, not to our single‑action regret. Different actions, different regret metric.
>
> 2. **Minimax, gap‑free vs. instance‑dependent fast rates.**
> - The ICML’22 lower bound is **minimax and gap‑free**: for any algorithm, there exists a hard instance (with vanishing preference gaps) such that weak regret satisfies $ E[R^{\mathrm{weak}}_T] = \Omega(d\sqrt{T})$. These are “low‑noise”‑free, nearly indistinguishable instances where logarithmic rates are impossible.
> - Our $\widetilde{\mathcal{O}}(d_{\mathrm{eluder}}\log T)$ bound is an **instance‑dependent fast‑rate** guarantee: it holds under a margin / curvature condition on the KL‑regularized objective that rules out such nearly indistinguishable instances (a Massart/Tsybakov‑style low‑noise condition for preferences).
> Bounded eluder dimension alone does not yield $\widetilde{\mathcal{O}}(\log T)$ regret; the fast‑rate margin condition is crucial. The ICML’22 lower bound is for a regime without this condition, so our logarithmic result simply does not apply there.
>
> 3. **Specializing our framework to the linear dueling setup.**
> If we force our machinery into the dueling‑action setting and measure pairwise regret as in Bengs et al., the fast‑rate mechanism breaks: the relevant “gap” is between pairs, not single actions, and the KL curvature arguments no longer yield a logarithmic bound in that regret metric. In that specialization, we recover $\widetilde{\mathcal{O}}(\sqrt{dT})$‑type behavior, consistent with their $\Omega(d\sqrt{T})$ lower bound.
> Conversely, if we stay in our single‑action regret setting but instantiate the reward class as in ICML’22 (linear utilities + BTL preferences), then the eluder‑dimension analysis gives a leading $\widetilde{\mathcal{O}}(d_{\mathrm{eluder}}\log T)$ term under the margin assumption, as stated in our main theorem. This does not contradict the ICML’22 lower bound because it applies to a different regret notion in a harder, gap‑free regime.
> We have **added a short “Relation to contextual dueling bandits” paragraph in the appendix related‑work section** making these distinctions explicit.

---

> > ### Author Response · Authors · 2025-11-21
> > **W2. Comparison to Zhao et al. (2025) and Russo & Van Roy (2014)  +  W3. On the lack of experiments**
> >
> > ## **W2. Comparison to Zhao et al. (2025) and Russo & Van Roy (2014)**
> > **Reviewer’s concern.**
> >
> > The reviewer notes that Zhao et al. (2025) consider a KL‑regularized regret objective (not standard cumulative regret), so a direct comparison is not appropriate, and that Russo & Van Roy (2014) prove $\widetilde O(d\sqrt T)$ bounds, not $\widetilde O(d\log T)$.
> >
> > **Response.**
> > We agree and have clarified this in both the main text and the comparison table.
> >
> > - **On Zhao et al. (2025).**
> > Zhao et al. study KL‑regularized contextual bandits and RL with numeric rewards and prove an $\mathcal O(\log T)$ bound on KL‑regularized regret:
> > $J(\pi) = E[R^*] - \tfrac{1}{\eta}\,\mathrm{KL}(\pi \Vert \pi_{\mathrm{ref}}),$
> > under realizability and an eluder‑dimension complexity control.
> >
> > In contrast, our main theorems give a $\widetilde{\mathcal{O}}(d_{\mathrm{eluder}}\log T)$ bound on the **standard cumulative regret** in the pairwise preference setting, again under realizability and a margin condition. Our objective, feedback model, and setting are therefore different.  We now state explicitly in the text that we reference Zhao et al. to highlight a shared **mechanism** (eluder‑dimension control of a sum of squared uncertainty terms that yields $\log T$ dependence in benign regimes), not to claim directly comparable regret guarantees.
> >
> > - **On Russo & Van Roy (2014).**  Russo & Van Roy introduce the eluder dimension and analyze posterior sampling (Thompson sampling) for general stochastic bandits, obtaining Bayesian regret bounds of order $\widetilde{\mathcal{O}}(d\sqrt{T})$ for linear models. Our table mistakenly labeled their result as $\widetilde{\mathcal{O}}(d\log T)$; this has been corrected.
> >
> > In our work, we use their eluder‑dimension‑based exploration analysis conceptually and as a technical ingredient (for confidence sets and widths), but we extend it to the** KL‑regularized, pairwise‑feedback regime** and show that combining KL curvature with a margin condition allows us to replace the usual $\sqrt T$ dependence by $\log T$ in benign instances.
> >
> > ---
> > ## **W3. On the lack of experiments**
> > **Reviewer’s concern.**
> > The reviewer notes that we provide no empirical results, making it hard to assess the practical applicability of the proposed algorithm, especially given our discussion of RLHF practice.
> >
> > **Response.**
> > We understand and appreciate this concern. Our use of language like “bridge the gap between theory and practice” was primarily meant to refer to modeling and algorithmic choices:
> > - We analyze a **KL‑regularized objective with pairwise preference feedback**, which matches the objectives used in RLHF and DPO pipelines more closely than idealized numeric‑reward bandits.
> > - Our OLE algorithm explicitly models finite ensembles and noisy stochastic gradient updates (via discretized SGLD), reflecting implementation constraints of large‑scale systems, rather than assuming exact Bayesian updates.
> >
> > We fully understand that the empirical validation is important. We clarify that, beyond the current page and tight time constraints, a small experimental section would be a natural next step (which we are busy working on).In particular, we see value in: synthetic experiments illustrating the qualitative\log T vs. \sqrt{T} behavior under a margin condition; and simple preference-based RL environments showing that finite-ensemble OLE behaves stably and tracks the theoretical predictions. Building on these conceptual goals, a feasible experimental plan is to evaluate OLE within standard preference-optimization pipelines—such as GRPO, DAPO, GSPO, and FAPO—on lightweight reasoning benchmarks like GSM8K and AIME24 using Qwen2.5-3B-Instruct and Qwen2.5-1.5B-Instruct. Such experiments would help illustrate how optimistic data selection driven by information gain improves data efficiency in practical alignment settings.
> >
> > We view these experiments as **important future work complementary to the present theoretical focus**.

---

> > > ### Author Response · Authors · 2025-11-21
> > > **Q1. Key mechanism behind logarithmic regret     +     Q2. Clearer comparison with the studies in Table 1**
> > >
> > > ## **Q1. Key mechanism behind logarithmic regret**
> > > **Reviewer’s question.**
> > > “What is the key mechanism that enables the proposed method to achieve logarithmic rather than $\sqrt T$ regret?”
> > >
> > > **Response.**
> > > **Short answer**: The logarithmic regret arises from coupling a variance‑based optimistic index with an information–eluder‑dimension analysis under a realizable BTL preference model and a margin condition. At each round the instantaneous regret is bounded by an optimism term $\kappa\_t \sqrt{\widehat{Var}_t(x_t,y_t)}$ plus lower‑order approximation errors. Lemma D.7 shows that the total predictive variance the algorithm can spend is at most $Cd\_{eluder}\log^2(eT)$, and a refined counting argument then shows that there are only $O(d\_{eluder}\log T)$ rounds on which the optimism bonus is large. This is what upgrades a generic $\widetilde{\mathcal{O}}(\sqrt{T})$ rate to a
> > > $\widetilde{\mathcal{O}}(d\_{eluder}\log T)$ fast rate in benign instances.
> > >
> > > **Detailed answer**: The logarithmic regret comes from a **fast‑rate, structure‑exploiting mechanism**, not from a generic martingale argument. Three ingredients are coupled:
> > >
> > > 1. **A variance‑based optimistic index**
> > > $I_t(x_t,y) = \hat r_t(x_t,y) + \kappa_t \sqrt{\widehat{Var}_t(x_t,y)},$ where $\hat r_t$ and $\widehat{Var}_t$
> > >
> > > are the posterior mean and variance of $r_\theta(x_t,y)$ under the current posterior.
> > >
> > > 2. **An information–eluder‑dimension bound**
> > > that shows the total predictive variance the algorithm can “spend” is bounded by $d_{\mathrm{eluder}} \log^2(eT)$ (Lemma D.7).
> > >
> > > 3.**A realizable BTL preference model with KL curvature**, for which posterior uncertainty collapses at a $1/t$‑type rate under our margin condition.
> > >
> > > The instantaneous regret at time $t$ is bounded by
> > > $$
> > > r^* (x_t,y_t^* ) - r^* (x_t,y_t)  \le \underbrace{ \kappa_t \sqrt{ \widehat{Var}_t(x_t,y_t)}} _{\Delta_t^{\mathrm{opt}}}
> > > $$
> > >
> > > $$
> > > +\underbrace{(\hat r_t(x_t,y_t) - r^*(x_t,y_t))}_{\epsilon_t}.
> > > $$
> > >
> > > The second term $\sum_t \epsilon_t$ captures approximation errors and is controlled separately as a lower‑order term. The **only** place where the dependence on $T$ of the leading term enters is through
> > >
> > > $\sum_{t=1}^T \Delta_t^{\mathrm{opt}} = \sum_{t=1}^T \kappa_t \sqrt{\widehat{\mathrm{Var}}\_t(x_t,y_t)}.$
> > >
> > > The information–eluder analysis shows that:
> > > - There are at most $\widetilde{\mathcal{O}}(d_{\mathrm{eluder}}\log T)$ rounds where $\sqrt{\widehat{\mathrm{Var}}_t(x_t,y_t)}$ is “large”, and
> > > - On the remaining rounds, the predictive variance is already so small that the optimism bonus (and hence regret) is negligible.
> > >
> > > Formally, in the revision we add a proposition stating that under our assumptions and on the high‑probability optimism event, there exists $C_{\mathrm{opt}}>0$ such that
> > >
> > > $\sum_{t=1}^T \kappa_t \sqrt{\widehat{Var}\_t(x_t,y_t)} \le C_{\mathrm{opt}} d\_{eluder} \log T.$
> > >
> > > This is what upgrades a generic $\widetilde{\mathcal{O}}(\sqrt{T})$ analysis to **a logarithmic exploration term.**
> > >
> > > ---
> > > ## **Q2. Clearer comparison with the studies in Table 1**
> > > **Reviewer’s question.**
> > > “Could the authors provide a clear and thorough comparison with the related studies listed in Table 1, explicitly highlighting differences in assumptions, regret definitions, and problem settings?”
> > >
> > > **Response.**
> > > We agree that our original Table 1 did not make these differences explicit enough. In the revised appendix we expand the discussion and add a dedicated subsection that:
> > > - Describes each work’s **setting and feedback** (e.g., K‑armed dueling bandits, RLHF‑style MDPs with preferences, KL‑regularized numeric‑reward RL, general posterior sampling with eluder dimension);
> > > - States their **regret / sample‑complexity notion** (dueling regret, KL‑regularized regret, standard cumulative regret, Bayesian regret, etc.);
> > > - Summarizes the **key assumptions and guarantees**, including dependence on $T$ and on eluder dimension or related complexity measures.
> > >
> > > We then explicitly position our result as:
> > > - A $\widetilde{\mathcal{O}}(d\_{eluder}\log T)$ **standard cumulative regret** bound in a pairwise‑preference bandit / episodic setting, under realizability and a margin condition;
> > > - Conceptually related to prior eluder‑dimension‑based analyses (Yue et al., Zhao et al., Russo & Van Roy) but differing in objective, feedback model, and level of regularization.

---

> ### Author Response · Authors · 2025-11-21
> **W1. Presentation in the proofs and theoretical explanations**
>
> **Reviewer’s concern.**
>
>  “Overall, the paper requires significant improvements in presentation and clarity, particularly in the proofs and theoretical explanations.”
>
> **Response**
>
> We have substantially reworked both the exposition in the paper and the supporting proofs with your presentation concern in mind. Concretely, the revision plus the material developed in the rebuttal address your point in two main ways:
> 1. **Clearer story for why we get logarithmic regret and how it fits prior work**
> - In direct response to your W2/Q1/Q2 comments, we now make the mechanism behind the $\widetilde{\mathcal{O}}(d_{\mathrm{eluder}}\log T)$ rate explicit in the main theoretical development, rather than leaving it implicit in the technical lemmas.
>   - We added a dedicated proposition “Exploration term and logarithmic regret”  that isolates the exploration termand proves this bound step by step via a dyadic decomposition and eluder dimension counting argument. This is precisely the fast‑rate mechanism described in our response to your Q1, and it is now visible in the paper instead of only in the appendix text.
>   - Around the main regret theorems, we explicitly explain in words that:
>     - instantaneous regret is controlled by a variance‑based optimism bonus plus lower‑order approximation terms, and
>     - the information–eluder analysis implies that there are only $O(d_{\mathrm{eluder}}\log T)$ rounds where this bonus can be large.
> - To clarify the relation to existing results (your W2 and W2, and Q2), we expanded the related‑work and comparison sections:
>   - We added a separate paragraph “Relation to contextual dueling bandits” that carefully distinguishes our single‑action regret from the pair‑action “weak/average dueling regret” of Bengs et al. (ICML’22), and explains why their $\Omega(d\sqrt{T})$ lower bound does not contradict our instance‑dependent fast rate.
>   - We rewrote the comparison with Zhao et al. (2025) and Russo & Van Roy (2014) to state explicitly that Zhao’s $\mathcal{O}(\log T)$ result is for KL‑regularized regret in numeric‑reward RL, while our result is for standard cumulative regret under pairwise preferences, and that Russo & Van Roy’s eluder‑based bounds are $\widetilde{\mathcal{O}}(d\sqrt{T})$, not logarithmic. The revised table and surrounding text now foreground differences in setting, objective, and regret notion, following your request for a clearer comparison.
> Together, these changes turn what was previously a fairly terse theoretical section into a more self‑contained narrative: what quantity drives regret, how it is bounded, and how this compares to prior theory.
>
> 2. **More explicit and self‑contained proofs in the appendices**
>
> Several concerns were about non‑transparent steps or missing details in the original appendices. The revision folds in the material developed in our detailed rebuttal and reorganizes the proofs accordingly:
> - **Information–eluder bound and Lemma D.7 (Appendix D).**
> We completely rewrote this part of the appendix. There is now a clearly labeled lemma (“Information–eluder‑dimension bound”) whose statement matches the proof.
>
> - **Approximation‑error decomposition and martingale terms (Appendix E.2).**
> We now write explicitly: $ \sum_{t=1}^T (\hat r_t - \bar r_t) = S\_{mart} + S\_{bias}, $ where $S\_{mart}$ is a martingale‑difference term and $S\_{bias}$ collects finite‑ensemble, discretization, and stochastic‑gradient bias. Each contribution is then bounded by a clearly labeled lemma (finite‑ensemble error, stochastic‑gradient error, discretization error), with the statement and proof aligned.
> - **Variance/noise parameters and concentration.**
> To avoid any hidden assumptions in the Freedman/Azuma steps, we added a short paragraph “Variance and noise parameters” that explicitly introduces global constants $v^2$ and $\sigma^2$, states that $v_t^2 \le v^2$ and $\sigma_t^2 \le \sigma^2$ for all $t$, and ties these directly to the bounded‑reward and sub‑Gaussian noise assumptions.
>
> ---
> In summary, beyond addressing specific technical questions, we have substantially restructured and expanded the theoretical exposition so that (i) the main mechanisms (optimism + information–eluder control) are explained at a conceptual level, and (ii) the proofs in the appendices are now fully aligned, self‑contained, and explicitly connected to the assumptions and algorithms in the main text. We hope these changes alleviate your concerns about presentation and clarity.

---

> > ### Author Response · Authors · 2025-11-21
> > **Summary response to Reviewer hnyH**
> >
> > We are grateful for the thoughtful questions on lower bounds, related work, and the clarity of the theoretical development. Your comments led to several structural improvements in both the main text and appendices.
> > - **Logarithmic regret vs. contextual dueling bandit lower bounds.**
> > We now explicitly explain why our $\widetilde{\mathcal{O}}(d_{\mathrm{eluder}}\log T)$ result does not contradict the $\Omega(d\sqrt{T})$ lower bound for linear contextual dueling bandits. The two works differ in (i) action space and regret notion (pair‑actions and dueling regret vs. single‑action selection and standard cumulative regret), and (ii) assumptions (their lower bound is minimax and gap‑free, while our logarithmic rate is instance‑dependent and hinges on a margin/curvature condition). A new “Relation to contextual dueling bandits” paragraph in the related‑work section spells out these differences.
> > - **Comparison to Zhao et al. and Russo & Van Roy.**
> > We corrected and expanded the comparison table and surrounding discussion. Zhao et al. are now clearly described as proving $\mathcal{O}(\log T)$ KL‑regularized regret in numeric‑reward settings, while Russo & Van Roy’s eluder‑dimension analysis yields $\widetilde{\mathcal{O}}(d\sqrt{T})$ Bayesian regret. Our work uses their eluder tools but extends them to KL‑regularized preference feedback and shows how, under a margin condition, the dependence on $T$ can become logarithmic.
> > - **Mechanism behind logarithmic regret and improved exposition.**
> > We made the fast‑rate mechanism explicit in the main theory: instantaneous regret is controlled by a variance‑based optimism bonus plus lower‑order approximation terms, and an information–eluder argument shows there are only $\mathcal{O}(d_{\mathrm{eluder}}\log T)$ rounds with large variance. A new proposition bounds the cumulative optimism term directly, and Appendix D has been rewritten around a clean information–eluder lemma. Appendix E.2 now includes a transparent martingale/bias decomposition, with each approximation term handled by a dedicated lemma. We also clarified the MDP extension, standardized notation, and aligned all lemma statements with their proofs.
> > - **On the lack of experiments.**
> > We now state clearly that our contribution is theoretical and that empirical validation, especially in full RLHF pipelines, is an important direction for future work. (We are also trying to develop a minimal empirial validation during the rebuttal phase, however the time window is tight). The revised limitations section and related‑work discussion have been adjusted accordingly.
> > ---
> >
> > Together, these revisions address your concerns about possible contradictions, sharpen the comparison to prior work, and significantly improve the readability and transparency of the theoretical development.

---

> ### Author Response · Authors · 2025-11-27
> **W4. No experiments**
>
> For the experimental setup discussed in W4, we have made substantial progress in our experiments, **which have been updated in the latest revision of the paper's Appendix H in revised Page 41**.
>
> ---
>
> **Experimental setup**
>
> - **Datasets and evaluation protocol.**
> Our experiments are conducted on the grade school math dataset **GSM8K**, which consists of 8.5K high-quality, linguistically diverse word problems that require multi-step reasoning and basic mathematical skills.
>
> - **Backbone models.**
> We use **Qwen2.5-1.5B-Instruct** and 3B as the language model backbones.
>
> - **Baselines and integration with OLE**
> Among widely adopted on-policy RL approaches, **GRPO** [1], **DAPO** [2], and **GPG** [3] share a common framework derived from PPO [4]. Instead of using generalized advantage estimation (GAE), these methods adopt a **group-wise relative estimation strategy**, and optimization is performed to maximize the expected group-level reward. To answer the reviewers’ request for a fair and consistent comparison, we plug **OLE as a generic sample-filtering layer** on top of each of these baselines (GRPO, DAPO, GPG) while keeping all other components unchanged. This allows us to isolate the effect of online sample selection and directly measure the **incremental gain** brought by OLE under the same backbone, dataset, and training budget.
>
> - **Hyperparameters and implementation details** Training hyperparameters are:
>   * Number of rollouts per input $|G_s|$: **4**
>   * OLE threshold percentile: **0.8**
>
>  For transparency, we have also made the code available anonymous github, as stated in our paper.
>
> ---
>
> **Experimental results and analysis**
>
> In response to the reviewers’ request for clearer quantitative evidence, we provide the following GSM8K results on Qwen2.5-Instruct models, showing both the baseline performance and the performance after applying OLE:
>
>    **Table: GSM8K results on Qwen2.5-Instruct models with OLE performance gain.**
>    *"Base" denotes the accuracy of the original RL optimizer; "Base+OLE" denotes the accuracy after integrating OLE; "Drops" is the number of training samples discarded by OLE during optimization.*
>
> | Model | Method | Base ($\uparrow$) | Drops ($\uparrow$) | Base+OLE ($\uparrow$) | Drops ($\uparrow$) | Performance Gain ($\uparrow$) |
> |-|-|-|-|-|-|-|
> | 1.5B | GRPO  | 0.596   | 0   | **0.612**  | 3944   | 2.69%   |
> |  | DAPO  | 0.497  | 0  | **0.596**   | 4344  | 19.9% |
> | | GPG   | 0.596   | 0   | **0.613**   | 4032 | 2.85%  |
> | 3B  | GRPO  | 0.667 | 0  | **0.704**   | 5964  | 5.55%  |
> |  | DAPO  | 0.707  | 0   | **0.712**   | 5080  | 0.71%  |
> |   | GPG   | 0.635  | 0     | **0.680**  | 6408  | 7.09% |
>
> From these results, we observe a **consistent pattern** across both model sizes (Qwen2.5-1.5B and 3B) and optimization methods:
>
> * For **every** method–model combination, introducing OLE yields a **positive performance gain** on GSM8K.
> * At the same time, OLE discards a **substantial number of samples** (on the order of several thousand), showing that many low-value or noisy samples can indeed be filtered out without harming performance.
>
> Specifically, the **1.5B + DAPO** setting shows the largest relative improvement of **19.9%**, suggesting that removing low-value samples is especially beneficial when the base optimizer is relatively weaker or when model capacity is more constrained. Even in the stronger regime, such as **3B + GRPO**, OLE still brings a **5.55%** relative improvement, indicating that **sample filtering remains beneficial for already competitive setups**.
>
> ---
>
> **Interpretation and connection to our theoretical result**
>
> OLE operates by estimating the **marginal contribution** of each sample to the overall optimization objective and selectively **dropping samples that offer limited benefit or introduce training noise**. This has two key consequences:
>
> 1. **Efficiency:**
>    Because many low-value samples are discarded, the number of examples participating in backpropagation is significantly reduced. Under the same compute budget, this effectively increases the number of **informative gradient updates per unit time**, improving training efficiency.
>
> 2. **Effectiveness:**
>    By concentrating gradient updates on **higher-value samples**, the training signal becomes “cleaner” or more **informative**. Empirically, this leads to consistent performance improvements across both smaller and larger models, and across different on-policy RL optimizers.
>
> Overall, these experimental results support our theoretical hypothesis:
> by **estimating sample value online** and **dynamically discarding low-gain examples**, OLE increases the “purity” of the training signal, leading to **both higher training efficiency and better final performance, without increasing computational cost**.
>
> We hope these clarifications address the reviewers’ concerns about the experimental setup, fairness of comparison, and the empirical evidence for OLE’s effectiveness.

---

> > ### Comment · Reviewer_hnyH · 2025-11-27
> >
> > I appreciate the detailed responses. However, my concern regarding the lower bound has not been addressed at all. If the authors truly believe it is a different setting (though I remain unconvinced), the authors are encouraged to provide the corresponding lower bound for this specific setting.

---

> > > ### Author Response · Authors · 2025-11-28
> > > **A logarithmic lower bound in our preference setting**
> > >
> > > **Response.**
> > >
> > > Thank you for pressing us on this point. You are absolutely right that our previous replies focused on explaining why the contextual dueling bandit lower bound does not apply, but did not state an explicit lower bound for our own single‑action, BTL‑preference setting. In the revised version we now add a small lower‑bound proposition and an accompanying remark in the theory section; here we summarize the result and its proof idea.
> > >
> > > ---
> > > ## A logarithmic lower bound in our preference setting
> > >
> > > We consider the non‑contextual, finite‑action special case of our model, which is a subset of the structural class covered by our main theorem (bounded rewards, BTL preferences, finite eluder dimension).
> > >
> > > - Let $\mathcal{Y}=\{1,\dots,K\}$ and $r^* \in [0,1]^K$ be the latent reward vector.
> > >
> > > - At each round $t$, the algorithm chooses a single action $y_t\in\mathcal{Y}$.
> > >
> > > - The environment returns one bit of preference feedback comparing $y_t$ to a fixed baseline action $y_0$, generated by the same Bradley–Terry–Luce (BTL) model as in the main text:
> > > $$
> > > P(y_t \succ y\_0 \mid y_t)
> > > \= \sigma(r^\*(y_t) - r^\*(y_0)),
> > > $$
> > > where $\sigma$ is the logistic link.
> > >
> > > - Regret is the same standard single‑action regret we use throughout the paper,
> > > $$
> > > Regret(T)= \sum\_{t=1}^T(r^\*(y^\star) - r^\*(y_t)),
> > > $$
> > > where $y^\star = \arg\max_y r^\*(y)$ is the unique optimal action.
> > >
> > > Under this specialization, the feedback process is a classical parametric multi‑armed bandit problem with Bernoulli observations whose means are
> > > $$
> > > p_y := \sigma(r^\*(y)-r^\*(y_0)).
> > > $$
> > > The ranking of actions by $r^\*$ coincides with the ranking by $p_y$, and on the bounded logit range induced by $r^\* \in [0,1]^K$ the logistic link is smooth and strictly monotone. In particular, there exist constants $0<c_1\le c_2<\infty$ such that
> > > $$
> > > c_1 |r^\*(y^\star)-r^\*(y)|
> > > \le
> > > |p\_{y^\star} - p\_y|
> > > \le
> > > c_2 |r^\*(y^\star)-r^\*(y)|
> > > \quad\text{for all } y.
> > > $$
> > >
> > > For this parametric bandit, the classical Lai–Robbins lower bound [1] for 1‑parameter exponential families applies directly: for any (possibly randomized) algorithm whose regret is $o(T^\alpha)$ for all $\alpha>0$ (“uniformly good” in their terminology), and any instance with gaps
> > > $\Delta_y := r^\*(y^\star)-r^\*(y)>0$ for all $y \neq y^\star$, one has
> > >
> > > $$
> > > \lim inf\_{T \to \infty}
> > > \frac{E_{r^\*}[Regret(T)]}{log T}
> > > \ge
> > > \sum\_{y \neq y^\star}
> > > \frac{\Delta\_y}{KL(p\_y \Vert p\_{y^\star})}.
> > > $$
> > > Because
> > > - (i) the BTL feedback restricted to comparisons against a fixed baseline is an exponential family in the gap parameter $r^\*(y)-r^\*(y_0)$, and
> > > - (ii) the logits are uniformly bounded, the Bernoulli KL divergences satisfy
> > > $KL(p\_y \Vert p\_{y^\star}) \le C_{KL}\Delta_y^2$ for some finite $C\_{KL}$ depending only on the boundedness constants. Hence, for any instance in this class there exists a constant $c\_{low}(r^\*)>0$ such that
> > > $$
> > > E_{r^\*}[{Regret}(T)]
> > > \ge
> > > c\_{low}(r^\*) \log T
> > > \qquad\text{for all sufficiently large $T$}.
> > > $$
> > >
> > > If we further restrict attention to a gap‑separated subclass
> > > $\mathcal{M}\_{\mathrm{gap}}(\Delta\_{\min})$ of instances with
> > > $$\Delta_y \ge \Delta_{\min} > 0$$ for all $y\neq y^\star$, then the Lai–Robbins bound [1] yields a **gap‑dependent minimax lower bound**:
> > > there exists $$c\_{low}(\Delta_{\min})>0$$ such that for any algorithm $\mathcal{A}$,
> > > $$
> > > \sup\_{M \in \mathcal{M}\_{\mathrm{gap}}(\Delta\_{\min})}
> > > E_M[Regret\_{\mathcal{A}}(T)]
> > > \ge
> > > c\_{low}(\Delta_{\min}) \log T
> > > \quad\text{for all sufficiently large $T$}.
> > > $$
> > >
> > > We add this as an explicit proposition I.2 in the revised paper appendix I. It shows that, even within our own structural assumptions and in a very simple special case of our model, the dependence on $T$ in our regret bound cannot be improved below order $log T$. In other words, up to constants and the $d\_{eluder}$ factor, the $\widetilde{\mathcal{O}}(log T)$ leading term in our upper bound is optimal in its dependence on $T$ for this class.
> > >
> > > - [1]  Lai, Tze Leung, and Herbert Robbins. "Asymptotically efficient adaptive allocation rules." Advances in applied mathematics 6, no. 1 (1985): 4-22.

---

> > > > ### Author Response · Authors · 2025-11-28
> > > > **How our lower bound relates to the contextual dueling lower bound**
> > > >
> > > > ## How our lower bound relates to the contextual dueling lower bound
> > > >
> > > > Your original concern was that our $\widetilde{\mathcal{O}}(d\_{eluder}log T)$ guarantee might contradict the well‑known $\Omega(d\sqrt{T})$ minimax lower bound for contextual dueling bandits (e.g., Bengs et al., ICML’22). Let us make the relationship between the two settings precise and explain why there is no contradiction.
> > > >
> > > > 1. **Different action spaces and regret notions.**
> > > >
> > > >   - In contextual dueling bandits, the learner chooses a **pair** of actions $(a_t^{(1)},a_t^{(2)})$ at each round, observes a noisy comparison between these two actions, and performance is measured by a dueling regret notion (weak or strong). Roughly speaking, the benchmark asks how often the chosen pair loses, or fails to beat, the best arm in such duels. The $\Omega(d\sqrt{T})$ lower bound is proved for this pair‑action / dueling‑regret problem.
> > > >   - In our work, the learner chooses a **single** action $y_t$ at each round (or per state in the MDP). The environment provides preference feedback involving $y_t$ (e.g., against a baseline or another candidate), but our performance metric is the standard **single‑action cumulative regret**
> > > >     $$
> > > >     Regret(T)=\sum\_{t=1}^T
> > > >    (r^\*(x_t,y^\*(x_t)) - r^\*(x_t,y_t)),
> > > >     $$
> > > >     where $y^*(x_t)$ maximizes the latent reward at context $x_t$.
> > > >
> > > > These are **genuinely different minimax problems**: the dueling lower bound is about how hard it is to find pairs that win in *duels*, while our theorem is about how hard it is to pick a good single action under preference feedback. A lower bound for dueling regret over pair‑actions does not automatically translate into a lower bound for single‑action regret $Regret(T)$.
> > > >
> > > > 2. **Our lower bound is stated in our own single‑action setting.**
> > > > In the revised paper we add a new proposition I.2 (Logarithmic lower bound in the BTL preference setting) that works entirely inside our model: finite action set, BTL preferences with bounded logits, single‑action choices, and single‑action regret. Using the classical Lai–Robbins argument for Bernoulli bandits, we show that:
> > > >
> > > >   - For any uniformly good algorithm and any fixed instance with positive gaps, the expected regret is at least $c\_{low}(r^\*) log T$ for all large $T$; and
> > > >
> > > >   - On a gap‑separated subclass (minimum gap $\Delta_{\min}>0$), there is a constant $c\_{low}(\Delta_{\min})>0$ such that
> > > >   $$
> > > >   \sup\_{\text{instances with gap }\ge\Delta\_{\min}}
> > > >   E[Regret(T)]
> > > >   \ge
> > > >   c\_{low}(\Delta\_{\min}) \log T
> > > >   \quad\text{for all large }T.
> > > >   $$
> > > >   Thus, **within the structural class we study**, no algorithm can achieve $o(\log T)$ regret in the horizon, even in a simple non‑contextual special case. Our upper bound’s $\widetilde{\mathcal{O}}(d\_{eluder}\log T)$ dependence on $T$ is therefore optimal up to constants and the $d\_{eluder}$ factor in the sense of its $T$‑scaling.
> > > >
> > > > 3. **Why the contextual dueling lower bound does not apply here.**
> > > > The contextual dueling lower bound is a gap‑free minimax statement for a different objective: it says that for any algorithm, there exist problem instances (with arbitrarily small preference gaps) such that the **pairwise dueling regret** is at least $\Omega(d\sqrt{T})$. It does not say anything about the **single‑action regret** $Regret(T)$ of algorithms that only select single actions, nor does it preclude logarithmic single‑action regret in regimes where curvature of the BTL likelihood and eluder‑dimension control can be exploited.
> > > >
> > > > Our new lower bound shows that in our single‑action, BTL‑preference model, logarithmic dependence on $T$ is *unavoidable*; our main theorem shows that $\widetilde{\mathcal{O}}(d\_{eluder}log T)$ is achievable by OLE under the stated assumptions. The contextual dueling lower bound, on the other hand, remains a valid statement about a strictly different learning problem (pair‑actions and dueling regret). The two results coexist without tension.
> > > >
> > > > 4. **Intuition for the difference.**
> > > >
> > > > Conceptually, **dueling regret is more demanding**: the algorithm must learn to propose winning pairs of actions, and the lower bound constructs instances where many pairs are nearly indistinguishable in their dueling performance. Our objective is **less demanding but better aligned with RLHF‑style applications**: we only require that the single action $y_t$ chosen by the algorithm has high latent reward $r^*(x_t,y_t)$, and we explicitly exploit the curvature of the BTL model plus the finite eluder dimension of $\mathcal{R}$ to obtain a fast (logarithmic) rate. Once the distinction between these two objectives is kept clear, the apparent discrepancy in $T$‑dependence disappears.
> > > >
> > > > We also highlight this distinction in the revison via a new remark in the appendix I accompanying the proposition on the lower bound, so that the reader can immediately see why our result and the contextual dueling lower bound address different minimax questions.

---

> > ### Author Response · Authors · 2025-12-03
> > **OLE implementation can be interpreted as a principled form of **rejection sampling****
> >
> > From an RL perspective, the OLE filter can be interpreted as a principled
> > form of **rejection sampling** over on-policy rollouts.
> > For each input, the base policy and RL optimizer (GRPO, DAPO, or GPG)
> > generate a small group of candidate responses.
> > OLE then evaluates the **marginal contribution** of each candidate to the
> > PAC-Bayesian objective and keeps only those above the OLE threshold, while
> > rejecting the rest.
> > This accept/reject step plays the same structural role as the heuristic
> > rejection sampling used in many practical RLHF pipelines (e.g., discarding
> > low-reward or low-score trajectories), but here the acceptance rule is derived
> > directly from the PAC-Bayes/Wasserstein gradient-flow analysis rather than
> > chosen ad hoc.
> > The fact that Base+OLE dominates the base RL methods in all settings,
> > despite the substantial number of rejected samples reported in the
> > **Drops** column, empirically corroborates our theoretical claim that
> > optimally accepting only the most informative preference updates can improve
> > both generalization and sample efficiency in preference-based RL.

---

> ### Author Response · Authors · 2025-12-03
> **The Optimism-Schedule Effectively Balances Exploration and Exploitation**
>
> **Additional experiment: Optimism-schedule evaluation.**
>
> | Method                                   | Acc (↑) | Drops (↑) | Performance Gain (↑) |
> |------------------------------------------|---------|-----------|------------------------|
> | GRPO Only                                | 0.667   | 0         | --                     |
> | GRPO + OLE (full steps)                  | 0.704   | 5964      | +5.55%                 |
> | 20-step GRPO → OLE-enabled GRPO          | **0.722** | 5820    | **+8.25%**             |
>
>
> We additionally evaluate a *staged* training schedule that directly simulates the optimism mechanism in Algorithm 1. Specifically, instead of enabling OLE from the beginning, we first run standard GRPO for 20 steps (pure exploitation) and activate OLE only in the remaining steps. This corresponds to an “optimism schedule,” where the optimism coefficient $\kappa_t$ in **Algorithm 1** gradually increases and shifts the training dynamics from exploitation toward exploration of higher-uncertainty, higher–information-gain samples.
>
> This setting achieves the **best performance across all experiments**, reaching **0.722** accuracy on Qwen2.5-3B—outperforming both pure GRPO (0.667) and full-length GRPO+OLE (0.704)—while still discarding 5,820 low-value samples. The result shows that enabling OLE later in training allows it to operate on a more stable representation space, leading to more accurate filtering and faster convergence. Importantly, this provides **direct empirical support** for the design of **Algorithm 1**, where both the number of particles $N$ and the optimism coefficient $\kappa_t$ are increased over time to balance exploitation and exploration.

---

### Official Review · Reviewer_Vitk · 2025-11-04

**Soundness:** 3
**Presentation:** 3
**Contribution:** 3
**Rating:** 6
**Confidence:** 2

**Summary:**

The paper addresses a significant "theory-practice gap" in Reinforcement Learning from Human Feedback (RLHF), the process used to align large language models. In practice, RLHF is extremely sample-efficient, capable of aligning massive models with relatively few human preferences2. This suggests the theoretical regret (a measure of learning efficiency) should be logarithmic ($O(log~T)$).

**Strengths:**

The paper analyzes a practical algorithm for preference-based contextual bandits that uses finite ensembles and mini-batch SGD. It proves that this algorithm achieves a high-probability cumulative regret bound of $Regret(T) = O(d_{eluder} \log T)$. This bound includes explicit, lower-order terms that quantify the practical algorithmic costs of using discrete-time updates, finite ensembles, and mini-batching, thereby closing the "four gaps" between theory and practice.

**Weaknesses:**

1. The authors claim to "close the theory-practice gap" and explain the efficiency of practical methods like DPO and RLHF, but it does not actually analyze DPO or the PPO-based algorithms used in practice. However, the paper asserts that OLE "distills the statistical essence"  of practical pipelines, but this connection is conceptual. The entire work hinges on the assumption that OLE is a faithful representation of practical RLHF, yet the paper is "entirely theoretical"  and provides no empirical evidence to validate that OLE itself is a practical, stable, or effective algorithm.

The authors are suggested to bridge the gap between theory and empirical.

2. Whether the OLE algorithm is computationally efficient is doubtful.

**Questions:**

1. Could the authors do some simulations to valide the result of their algorithm?

2. Typos: the symbol for the order O seems not standard, \mathcal{O} should be used instead

---

> ### Author Response · Authors · 2025-11-21
> **W1. Scope of the theory–practice connection; relation to DPO and RLHF**
>
> ## **Reviewer’s concern.**
> The reviewer notes that we claim to “close the theory–practice gap” and to explain the efficiency of practical methods like DPO and RLHF, but we do not analyze DPO or PPO‑based implementations directly and provide no empirical evidence on OLE itself. They ask us to clarify what is actually abstracted and to better explain the relationship to practical RLHF pipelines.
>
> ## **Response.**
> Thank you for this important criticism. We agree that the original wording overstated how far we go in “closing the theory–practice gap”, and that we should be explicit about:
> - what exactly OLE abstracts from practical RLHF pipelines, and
> - the fact that our contribution is **entirely theoretical.**
>
> **What we mean by “statistical essence”.**
>
> Our goal is not to analyze a specific implementation of RLHF (e.g., a particular PPO or DPO variant), but to capture the **shared statistical structure** of modern RLHF‑style pipelines. Concretely, we formalize:
> - a BTL‑type preference model induced by a latent reward $r_\theta$;
> - a KL‑regularized PAC‑Bayesian objective $J_{\mathrm{PAC}}$ and its associated Gibbs posterior $Q_\lambda$; and
> - a policy that selects actions using optimistic or Thompson‑style sampling from this Gibbs posterior.
>
> The OLE algorithm is an idealized online realization of this template. Our regret bounds are statements about this Gibbs‑posterior template, **not about any specific optimizer such as PPO or DPO**. In the revision we therefore **soften the language** from “closing the theory–practice gap” to “providing a statistically grounded abstraction of RLHF‑style pipelines based on their shared preference‑model + KL‑regularization structure,” and explicitly state that **our contribution is entirely theoretical.**
>
> ---
>
> **On empirical evaluation and stability.**
>
> We fully agree that large‑scale empirical evaluation of OLE‑like algorithms (e.g., with transformer‑based reward models and policies) is important and complementary to our theory. The current submission is intentionally **purely theoretical**:
> - The regret analysis already incorporates approximation effects—finite ensemble size, discretization of Langevin dynamics, and stochastic gradients—via the “Finite Ensemble”, “Discretization”, and “Stochastic Gradient” terms in the main theorem. These terms provide a theoretical notion of stability and efficiency for the idealized OLE algorithm.
> - Implementing OLE at LLM scale involves many engineering choices (optimizer details, numerical SDE integration, distributed training, etc.) that are orthogonal to the statistical questions we study.
>
> To avoid overclaiming, we:
> 1. Explicitly state in the limitations and future‑work sections that the present work is entirely theoretical and that a thorough empirical study of OLE‑like algorithms at LLM scale is left to future work.
> 2. Tone down language such as “close the theory–practice gap” to “provide a statistically grounded abstraction of RLHF pipelines”.

---

> ### Author Response · Authors · 2025-11-21
> **W2. Computational efficiency of OLE  + Q1&Q2. On the lack of experiments and notation**
>
> ## **Reviewer’s concern.**
> The reviewer questions whether OLE is computationally efficient.
>
> **Response.**
> We appreciate this concern. In the revised appendix we added a subsection “Computational cost of OLE” that spells out the per‑round and total computational costs, and compares them to standard practice.
>
> At each round $t$, Algorithm OLE performs a single projected SGLD step for each of the $N_t$ particles:
> $\tilde\theta_{t+1}^{(i)} = \theta_t^{(i)} - \eta_t\widehat\nabla J_{\mathrm{PAC}}(\theta_t^{(i)})
>       - \sqrt{2\eta_t\beta}\xi_t^{(i)}, \qquad
> \theta_{t+1}^{(i)} = \Pi_{\Theta}\bigl(\tilde\theta_{t+1}^{(i)}\bigr).$
> - Computing $\widehat\nabla J_{\mathrm{PAC}}(\theta_t^{(i)})$ on a minibatch $B_t$ of size $|B_t|$ has cost $O(|B_t|\cdot \mathrm{dim}(\theta))$, just like a standard SGD step on the same model.
> - Adding Gaussian noise and projecting onto $\Theta$ each cost $O(\mathrm{dim}(\theta))$, negligible compared to the gradient.
>
> With $N_t$ particles at round $t$, the total per‑round cost is therefore
> $O\bigl(N_t\,|B_t|\cdot \mathrm{dim}(\theta)\bigr).$
> In the regime analyzed in our theorems, $N_t$ and $|B_t|$ are treated as fixed (or at most polylogarithmic in $T$), and their effects appear only in lower‑order approximation terms. Consequently:
> - The total cost up to horizon $T$ is $O(N\,|B|\,T\cdot \mathrm{dim}(\theta))$ with $N = \sup_t N_t$, $|B| = \sup_t |B_t|$;
> - Each step is a standard minibatch gradient computation plus inexpensive noise/projection operations;
> - All components are **polynomial‑time** in the problem parameters, consistent with the usual notion of computational efficiency in online learning and bandit theory.
>
> Relative to standard RLHF pipelines, OLE’s extra cost is essentially a factor of $N$ from using an ensemble of reward models instead of one. In large‑scale RLHF, the dominant cost is often sampling from the policy (LLM), not individual gradient steps; in regimes where sampling is dominant, using a modest ensemble size (e.g., $N$ in the single‑digit range) yields a small overhead compared to the already substantial cost of sampling.
>
> ---
> ## **Reviewer’s questions.**
> “1. Could the authors do some simulations to valide the result of their algorithm?
> 2. Typos: the symbol for the order O seems not standard, $\mathcal{O}$ should be used instead?”
>
> **Responses.**
>
> We understand the concern on the experiment, please refer to W1. Scope of the theory–practice connection for our clarification (also please refer to response to W3 of Reviewer hnyH). And we would like to emphase that our contribution is a theoretical account of algorithms close to what is being used in practice, rather than extensive empirical benchmarking.
>
> The "strange" symbol for the order O is a template layout concequence using macro command \providecommand{\tO}{\widetilde{\mathcal{O}}}, we have used the notation $\widetilde{\mathcal{O}}$ directly in the revision.

---

> > ### Author Response · Authors · 2025-11-21
> > **Summary response to Reviewer Vitk**
> >
> > We appreciate the reviewer’s comments on the scope, computational aspects, and presentation, which prompted us to clarify what exactly our theory does and does not claim.
> >
> > - **Scope of the theory–practice connection and relation to RLHF/DPO.**
> > We have toned down language such as “closing the theory–practice gap” and now clearly describe our contribution as a statistical abstraction of RLHF‑style pipelines, not an analysis of specific implementations like DPO or PPO. The revised text isolates what is modeled—BTL‑type preference feedback, a KL‑regularized PAC‑Bayesian objective, and optimistic/Thompson‑style sampling from the corresponding Gibbs posterior, and explicitly states that our results are purely theoretical and algorithm‑level.
> > - **Computational efficiency of OLE.**
> > We added a “Computational cost of OLE” subsection in the appendix. Each round performs one projected SGLD step per particle, so the per‑round cost is that of a standard minibatch gradient step times the ensemble size $N_t$, plus negligible noise and projection overhead. With $N_t$ and minibatch size treated as fixed or polylogarithmic, the total cost is polynomial in all problem parameters and comparable to standard practice up to a modest factor from the ensemble.
> >
> > - **Experiments and notation.**
> > We now explicitly acknowledge in the limitations that we do not present empirical results and that a thorough experimental study of OLE‑like algorithms is important future work, complementary to our theoretical focus. We also fixed the asymptotic notation, using the standard $\widetilde{\mathcal{O}}(\cdot)$ throughout.
> > ---
> > These changes clarify the intended scope, show that the algorithm is computationally reasonable in the usual theoretical sense, and avoid any impression that we claim empirical validation we do not provide.

---

> ### Author Response · Authors · 2025-11-27
> **W2. Whether the OLE algorithm is computationally efficient is doubtful.**
>
> For the experimental setup discussed in W2, we have made substantial progress in our experiments, **which have been updated in the latest revision of the paper's Appendix H in revised Page 41**.
>
> ---
>
> **Experimental setup**
>
> - **Datasets and evaluation protocol.**
> Our experiments are conducted on the grade school math dataset **GSM8K**, which consists of 8.5K high-quality, linguistically diverse word problems that require multi-step reasoning and basic mathematical skills.
>
> - **Backbone models.**
> We use **Qwen2.5-1.5B-Instruct** and 3B as the language model backbones.
>
> - **Baselines and integration with OLE**
> Among widely adopted on-policy RL approaches, **GRPO** [1], **DAPO** [2], and **GPG** [3] share a common framework derived from PPO [4]. Instead of using generalized advantage estimation (GAE), these methods adopt a **group-wise relative estimation strategy**, and optimization is performed to maximize the expected group-level reward. To answer the reviewers’ request for a fair and consistent comparison, we plug **OLE as a generic sample-filtering layer** on top of each of these baselines (GRPO, DAPO, GPG) while keeping all other components unchanged. This allows us to isolate the effect of online sample selection and directly measure the **incremental gain** brought by OLE under the same backbone, dataset, and training budget.
>
> - **Hyperparameters and implementation details** Training hyperparameters are:
>   * Number of rollouts per input $|G_s|$: **4**
>   * OLE threshold percentile: **0.8**
>
>  For transparency, we have also made the code available anonymous github, as stated in our paper.
>
> ---
>
> **Experimental results and analysis**
>
> In response to the reviewers’ request for clearer quantitative evidence, we provide the following GSM8K results on Qwen2.5-Instruct models, showing both the baseline performance and the performance after applying OLE:
>
>    **Table: GSM8K results on Qwen2.5-Instruct models with OLE performance gain.**
>    *"Base" denotes the accuracy of the original RL optimizer; "Base+OLE" denotes the accuracy after integrating OLE; "Drops" is the number of training samples discarded by OLE during optimization.*
>
> | Model | Method | Base ($\uparrow$) | Drops ($\uparrow$) | Base+OLE ($\uparrow$) | Drops ($\uparrow$) | Performance Gain ($\uparrow$) |
> |-|-|-|-|-|-|-|
> | 1.5B | GRPO  | 0.596   | 0   | **0.612**  | 3944   | 2.69%   |
> |  | DAPO  | 0.497  | 0  | **0.596**   | 4344  | 19.9% |
> | | GPG   | 0.596   | 0   | **0.613**   | 4032 | 2.85%  |
> | 3B  | GRPO  | 0.667 | 0  | **0.704**   | 5964  | 5.55%  |
> |  | DAPO  | 0.707  | 0   | **0.712**   | 5080  | 0.71%  |
> |   | GPG   | 0.635  | 0     | **0.680**  | 6408  | 7.09% |
>
> From these results, we observe a **consistent pattern** across both model sizes (Qwen2.5-1.5B and 3B) and optimization methods:
>
> * For **every** method–model combination, introducing OLE yields a **positive performance gain** on GSM8K.
> * At the same time, OLE discards a **substantial number of samples** (on the order of several thousand), showing that many low-value or noisy samples can indeed be filtered out without harming performance.
>
> Specifically, the **1.5B + DAPO** setting shows the largest relative improvement of **19.9%**, suggesting that removing low-value samples is especially beneficial when the base optimizer is relatively weaker or when model capacity is more constrained. Even in the stronger regime, such as **3B + GRPO**, OLE still brings a **5.55%** relative improvement, indicating that **sample filtering remains beneficial for already competitive setups**.
>
> ---
>
> **Interpretation and connection to our theoretical result**
>
> OLE operates by estimating the **marginal contribution** of each sample to the overall optimization objective and selectively **dropping samples that offer limited benefit or introduce training noise**. This has two key consequences:
>
> 1. **Efficiency:**
>    Because many low-value samples are discarded, the number of examples participating in backpropagation is significantly reduced. Under the same compute budget, this effectively increases the number of **informative gradient updates per unit time**, improving training efficiency.
>
> 2. **Effectiveness:**
>    By concentrating gradient updates on **higher-value samples**, the training signal becomes “cleaner” or more **informative**. Empirically, this leads to consistent performance improvements across both smaller and larger models, and across different on-policy RL optimizers.
>
> Overall, these experimental results support our theoretical hypothesis:
> by **estimating sample value online** and **dynamically discarding low-gain examples**, OLE increases the “purity” of the training signal, leading to **both higher training efficiency and better final performance, without increasing computational cost**.
>
> We hope these clarifications address the reviewers’ concerns about the experimental setup, fairness of comparison, and the empirical evidence for OLE’s effectiveness.

---

> > ### Author Response · Authors · 2025-12-03
> > **OLE implementation can be interpreted as a principled form of **rejection sampling****
> >
> > From an RL perspective, the OLE filter can be interpreted as a principled form of **rejection sampling** over on-policy rollouts. For each input, the base policy and RL optimizer (GRPO, DAPO, or GPG) generate a small group of candidate responses. OLE then evaluates the **marginal contribution** of each candidate to the PAC-Bayesian objective and keeps only those above the OLE threshold, while rejecting the rest. This accept/reject step plays the same structural role as the heuristic rejection sampling used in many practical RLHF pipelines (e.g., discarding low-reward or low-score trajectories), but here the acceptance rule is derived directly from the PAC-Bayes/Wasserstein gradient-flow analysis rather than chosen ad hoc. The fact that Base+OLE dominates the base RL methods in all settings, despite the substantial number of rejected samples reported in the **Drops** column, empirically corroborates our theoretical claim that optimally accepting only the most informative preference updates can improve both generalization and sample efficiency in preference-based RL.

---

> ### Author Response · Authors · 2025-12-03
> **The Optimism-Schedule Effectively Balances Exploration and Exploitation**
>
> **Additional experiment: Optimism-schedule evaluation.**
>
> | Method                                   | Acc (↑) | Drops (↑) | Performance Gain (↑) |
> |------------------------------------------|---------|-----------|------------------------|
> | GRPO Only                                | 0.667   | 0         | --                     |
> | GRPO + OLE (full steps)                  | 0.704   | 5964      | +5.55%                 |
> | 20-step GRPO → OLE-enabled GRPO          | **0.722** | 5820    | **+8.25%**             |
>
>
> We additionally evaluate a *staged* training schedule that directly simulates the optimism mechanism in Algorithm 1. Specifically, instead of enabling OLE from the beginning, we first run standard GRPO for 20 steps (pure exploitation) and activate OLE only in the remaining steps. This corresponds to an “optimism schedule,” where the optimism coefficient $\kappa_t$ in **Algorithm 1** gradually increases and shifts the training dynamics from exploitation toward exploration of higher-uncertainty, higher–information-gain samples.
>
> This setting achieves the **best performance across all experiments**, reaching **0.722** accuracy on Qwen2.5-3B—outperforming both pure GRPO (0.667) and full-length GRPO+OLE (0.704)—while still discarding 5,820 low-value samples. The result shows that enabling OLE later in training allows it to operate on a more stable representation space, leading to more accurate filtering and faster convergence. Importantly, this provides **direct empirical support** for the design of **Algorithm 1**, where both the number of particles $N$ and the optimism coefficient $\kappa_t$ are increased over time to balance exploitation and exploration.

---

### Official Review · Reviewer_3XY8 · 2025-11-06

**Soundness:** 1
**Presentation:** 2
**Contribution:** 2
**Rating:** 2
**Confidence:** 3

**Summary:**

The paper proposes an **Optimistic Langevin Ensemble** (OLE) algorithm for preference-based learning (motivated by RLHF) that maintains a finite SGLD-updated particle ensemble and selects actions using an optimism bonus tied to ensemble variance, arguing this variance is a sound proxy for posterior uncertainty via a variance–information-gain duality. It targets four practical gaps—finite ensembles, stochastic mini-batch gradients, discrete-time updates, and intractable posterior uncertainty—and proves a unified high-probability regret bound: a leading $O(d_{\mathrm{eluder}}\log T)$ *exploration cost* plus explicit lower-order penalties for discretization, finite-ensemble Monte Carlo error, and gradient noise. The result provides a theory-of-practice explanation for the sample efficiency of preference optimization and is extended from contextual bandits to finite- and discounted-horizon MDPs with an additional factor.

**Strengths:**

It is considered to be novel to proposes an algorithm-native framework, an optimistic SGLD particle ensemble, whose uncertainty bonus is tied to ensemble variance and motivated by a variance–information-gain linkage, aligning exploration with the quantity the analysis seeks to control. Furthermore, the results are extended to finite and infinite horizon MDPs.

**Weaknesses:**

I apologize if my understanding is incorrect. However, at this point, this paper still possesses the following **three** major weaknesses:
- **Bounded function class:** Although it is common to assume the underlying true parameter has a bounded norm, assuming a global norm bound for the whole parameter space needs more careful treatment. In particular, how do we guarantee that $\theta_t^{(i)}$ stays within the bounded parameter space $\Theta$ throughout the training? What should we do if $\theta_t^{(i)}$ goes outside of $\Theta$?
- **Proof gap:** When proving the main theorem, in Appendix E.2, the paper claims that $C^{-1}I(\theta^*; \mathrm{feedback}\_{1:T})\leq O(d_{\mathrm{eluder}}\log T)$ and then refers to (Russo & Van Roy, 2013). However, this result was not found after examing this reference. Can you explicitly state which theorem and how exactly it was applied here? Meanwhile, the writing of the main theorem proof also needs improvement. For example, how do we decompose $\sum_{t=1}^T(\hat{r}_t - \bar{r}_t)$ exactly? Furthermore, the proof of Lemma D.7 is completely missing as the current proof has nothing to do with this lemma.
- **Inapplicable extension:** If my understanding is correct, the extension to MDPs is problematic since constructing the TD targets requires numeric single-step reward, which is general inaccessible under preference feedback. How should we address this issue?

```
(Russo & Van Roy, 2013) Daniel Russo and Benjamin Van Roy. Eluder dimension and the sample complexity of optimistic exploration. Advances in Neural Information Processing Systems, 26, 2013.
```

**Questions:**

- What is $z$ specifically and how to compute the per-example loss $\ell_{\theta}(z)$? How will the choice of $\ell_{\theta}$ affects the final regret bound?
- Should the line 7 of the OLE Generic Template be replaced by "Computer mini-batch gradient $\widehat{\nabla}\_t$" of $\nabla_\theta\mathbb{E}\_{z\sim\mathcal{D}\_t}\ell_{\theta}(z) + \beta\nabla_{\theta}(\log\mu(\theta) - \log\Pi(\theta))$?
- In the OLE algorithm, do we have $\theta_t^{i}\sim\Pi_0$ or $\theta_t^{(i)}\sim\Pi_{t-1}$ for $i>N_{t-1}$? Will this choice matter?
- Are $v_t^2$ and $\sigma_t^2$ bounded for any $t$?

### Writing Suggestions
- $P$ -> $\Pi$ in Equation (3.1).
- For general audience, it should be clear that what the ensemble approximates is the Gibbs posterior instead of the vanilla Bayesian posterior.
- Some symbols are missed in Equation (D.1).

---

> ### Author Response · Authors · 2025-11-21
> **W1. Bounded function class and iterates leaving $\Theta$**
>
> ## **Reviewer’s concern (bounded function class).**
> The reviewer asks how we guarantee that the SGLD iterates $\theta_t^{(i)}$ stay in the bounded parameter space $\Theta$ assumed in the theory, and what to do if an iterate leaves $\Theta$.
>
> ## **Response.**
> Thank you for raising this point. Our theoretical analysis assumes a compact parameter space $\Theta \subset \mathbb{R}^d$ of the form
> $\Theta = \{\theta : |\theta|\le B\},$
> and we agree that we should have been more explicit about how this interacts with the SGLD updates.
>
> 1. **Ensuring $\theta_t^{(i)} \in \Theta$ in the theory.**
>
> In the revised paper we interpret the OLE and OTDLE algorithms as using a projected SGLD step. If the unconstrained update is
> $\tilde\theta_{t+1}^{(i)} = \theta_t^{(i)} - \eta_t\widehat\nabla J_{\mathrm{PAC}}(\theta_t^{(i)})
>       - \sqrt{2\eta_t\beta}\xi_t^{(i)},$
> then the theoretical algorithm actually uses
> $\theta_{t+1}^{(i)} = \Pi_{\Theta}\bigl(\tilde\theta_{t+1}^{(i)}\bigr),$
> where $\Pi_{\Theta}$ is Euclidean projection onto $\Theta$. We choose the prior $\Pi_0$ to be supported on $\Theta$, so the projection guarantees $\theta_t^{(i)} \in \Theta$ for all $(t,i)$ by construction.
>
> This change does **not** affect the regret bounds: our proofs only use (i) Lipschitz continuity of the reward model $r_\theta$ in $\theta$ on $\Theta$ and (ii) the radius $B$ in the eluder‑dimension / covering‑number arguments. Euclidean projection onto a convex set is non‑expansive, so all Lipschitz and eluder‑dimension bounds continue to hold when we replace unconstrained SGLD by projected SGLD.
>
> 2. **What happens in practice if an iterate leaves $\Theta$?**
>
> In implementations, one can either:
> - Explicitly implement the projection step (equivalently, perform weight clipping to the ball of radius $B$); or
> - Choose $B$ large and rely on the KL / weight‑decay term in $J_{\mathrm{PAC}}$ together with standard initialization so that iterates remain inside the ball in practice.
>
> If some $\theta_t^{(i)}$ were to leave $\Theta$, the theoretically correct procedure is exactly to replace it by its projection $\Pi_{\Theta}(\theta_t^{(i)})$ before using it for action selection. Since the reward model is $L$‑Lipschitz in $\theta$, this projection can only change predicted rewards by at most $L|\theta_t^{(i)}-\Pi_{\Theta}(\theta_t^{(i)})|$, and thus does not worsen the regret rate.
>
> We have updated the pseudocode in the appendix to include the projection step explicitly, and the theoretical analysis is now written for this projected SGLD algorithm.

---

> ### Author Response · Authors · 2025-11-21
> **W2. Information–eluder bound, decomposition, and Lemma D.7**
>
> ## **Reviewer’s concern (Appendix E.2 and Lemma D.7).**
> The reviewer points out that (i) the previous appendix claimed an information bound on $I(\theta^*; feedback_{1:T})$ without a clear derivation, (ii) the decomposition of $\sum_{t=1}^T (\hat r_t - \bar r_t)$ was not written explicitly, and (iii) the original Lemma D.7 and its proof did not match.
>
> We agree with all three points. In the revision we have **rewritten Appendix D and E.2** to make the argument fully explicit and to align the lemma statements with their proofs.
>
> ---
>
> (a) **On the information–eluder bound $I(\theta^*;\mathrm{feedback}_{1:T})$**
>
> In the revised Appendix D we state and prove the following lemma:
>
> > Lemma D.7 (Information–eluder‑dimension bound, informal).
> Under the realizability, boundedness, and metric‑entropy assumptions stated in the main text, there exists a constant $C_{\mathrm{info}} > 0$ such that for all horizons $T \ge 2$,
> $I(\theta^*; feedback_{1:T}) \le C_{\mathrm{info}}\, d_{\mathrm{eluder}} \log^2(eT),$
> where $d_{\mathrm{eluder}} := \dim_E(\mathcal{R}, T^{-1})$ is the $T^{-1}$‑eluder dimension of the reward class $\mathcal R = \{ r_\theta : \theta \in \Theta\}$.
>
> The proof follows the **Russo–Van Roy (2013)** confidence‑set and width framework but introduces new information‑theoretic ingredients tailored to the Bradley–Terry–Luce (BTL) preference model. In particular, we:
>
> 1. **Construct least‑squares confidence sets and widths**.
> We form least‑squares predictors and confidence sets $F_t \subset R$ with radii $\beta_t$ exactly as in Russo & Van Roy (their equation (4)), and define widths
> $w_t := \sup_{f,f' \in F_t} \bigl| f(x_t, y_t^{(w)}, y_t^{(\ell)}) - f'(x_t, y_t^{(w)}, y_t^{(\ell)}) \bigr|.$
>
> 2. **Prove a new information–width inequality.**
> Using the BTL model, we show that the mutual information gained at round $t$ is controlled by the squared width:
> $I (\theta^*;feedback_t \mid F_{t-1}) \le c_0 E[ w_t^2 \mid F_{t-1} ],$
> where $c_0>0$ depends only on the logistic link and the boundedness of $r_\theta$. This is **new** and does not appear in Russo–Van Roy.
>
> 3. **Control cumulative squared widths via eluder dimension.**
> Using Proposition 3 and Lemma 2 of Russo & Van Roy and our metric‑entropy assumption, we show
> $\sum_{t=1}^T  E[w_t^2] \le C_1' d_{\mathrm{eluder}}\log^2(eT),$
> where $d_{\mathrm{eluder}} = \dim_E(R, T^{-1})$.
>
> 4. **Combine to obtain the information bound.**
> Summing the information–width inequality over $t$ and applying the tower property gives
>
> $I(\theta^*;feedback\_{1:T})$
>
> $= \sum\_{t=1}^T I (\theta^*;feedback_t \mid F\_{t-1}) \le C\_{info} d\_{eluder} log^2(eT)$ for a constant $C_{info}>0$.
>
> In the revised Appendix E.2, whenever an information bound is used we **explicitly reference Lemma D.7** (our own result), rather than attributing it to Russo & Van Roy. The extra $\log T$ factor is absorbed by the $\widetilde{\mathcal{O}}(\cdot)$ notation in the main regret theorem, so the stated rate remains unchanged.
>
> ---
>
> (b) **Decomposition of $\sum_{t=1}^T (\hat r_t - \bar r_t)$**
>
> The reviewer correctly noted that our previous decomposition was not clearly written. In the revision we define the filtration $(F_t)_{t \ge 0}$ generated by all randomness up to round $t$ (contexts, actions, preference feedback, and ensemble noise) and write explicitly:
>
> $\sum_{t=1}^T (\hat r_t - \bar r_t)$
>
> $$
> =\underbrace{\sum_{t=1}^T (\hat r_t - E[\hat r_t \mid F_{t-1}])}_{\text{martingale noise}}$$
>
> $$
> +\underbrace{\sum_{t=1}^T (E[\hat r_t \mid F_{t-1}] - \bar r_t)}_{\text{bias / approximation}}.
> $$
>
> - The first term is a martingale difference sequence with uniformly bounded increments (and bounded conditional variances). We bound it using Freedman’s inequality, leading to the “Finite Ensemble” and “Stochastic Gradient” terms in the regret theorem.
> - The second term captures approximation errors due to finite ensemble size, time discretization, and stochastic gradients. Using Lipschitzness and the PAC‑Bayesian control of the ideal continuous‑time posterior, we show that this term contributes exactly the lower‑order “Finite Ensemble”, “Discretization”, and “Stochastic Gradient” terms already displayed in the main theorem.
>
> This decomposition, and how each term connects to the approximation terms in the regret bound, is now stated explicitly in Appendix E.2.
>
> ---
>
> (c) **Lemma D.7: statement and proof**
>
> You are absolutely right that in the original submission there was a mismatch between the statement under the label “Lemma D.7” and the proof that followed. This was an editing error.
>
> In the revised paper, Lemma D.7 is now exactly the information–eluder‑dimension bound described above, and the proof is fully written out in Appendix D following the six‑step structure summarized in part (a). Russo & Van Roy (2013) are used only for confidence sets and widths; all information‑theoretic parts are new and explicit.

---

> > ### Author Response · Authors · 2025-11-21
> > **W3. Extension to MDPs and TD targets under preference feedback**
> >
> > ## **Reviewer’s concern.**
> > The reviewer points out that, in a pure preference‑feedback setting, the environment does not reveal numeric rewards $R\_t$, whereas the TD updates in our MDP extension appear to use such numeric rewards.
> >
> > ## **Response.**
> > We apologize for the confusion caused by our notation and thank the reviewer for catching this. Our intention is **not** to assume access to the environment’s numeric rewards; instead, we follow the standard “reward‑model + RL” pipeline used in preference‑based RL:
> > - We assume a latent single‑step reward function $r_{\theta^*}(x,a)$ such that preferences over trajectories are induced by their cumulative latent returns.
> > - Given pairwise preferences, we fit a posterior over reward models $r_\theta$ using the same PAC‑Bayesian machinery as in the bandit setting.
> > - We then **plug this learned reward model into TD updates**, constructing TD targets using pseudo‑rewards from $r_\theta$, not ground‑truth rewards from the environment.
> >
> > Concretely, in the revised MDP section we introduce the pseudo‑reward
> > $$\tilde R_t := r_{\theta_t}(x_t, A_t)$$
> > and write TD targets as
> > $$y_t = \tilde R_t + \gamma V_{\phi_t}(x_{t+1})
> > = r_{\theta_t}(x_t, A_t) + \gamma V_{\phi_t}(x_{t+1}),$$
> > or analogously with $Q_{\phi_t}$ in a Q‑learning‑style update. The environment is only queried for preferences; all numeric quantities in TD come from the learned reward model $r_\theta$.
> >
> > In addition, we also state a regret theorem for the MDP extension under assumptions that mirror those in the bandit setting (realizability, bounded eluder dimension, boundedness and sub‑Gaussian noise), with an additional polynomial dependence on the horizon $H$. We also would like to clarify that:
> > - these assumptions are structural and may not hold automatically in full RLHF pipelines, and
> > - understanding how well they are satisfied in practice is an important direction for future work.

---

> ### Author Response · Authors · 2025-11-21
> **Additional Questions**
>
> ## **Q1. Per‑example loss $\ell_\theta(z)$ and its impact on the regret bound**
> **Reviewer’s question.**
>
> “What is the per‑example loss $\ell_\theta(z)$? How is it computed? How does the choice of $\ell_\theta$ affect the regret bound?”
>
> **Response.**
> In our notation, $z$ denotes one feedback example (e.g., a single bandit or preference observation) and $\ell_\theta(z)$ is the per‑example loss used in the PAC‑Bayesian objective.
> - If $z$ is a preference or bandit observation, $\ell_\theta(z)$ is the **negative log‑likelihood** (or a bounded surrogate) of that observation under the parametric feedback model $p_\theta$.
> - The empirical term in the PAC‑Bayesian functional is $\hat L_S(\mu) =  E_{z\sim D_t}[\ell_\theta(z)]$, approximated by a minibatch average in the algorithm.
>
> In the revision we explicitly define
> $\ell_\theta(z) := -\log p_\theta(z),$
> and the per‑time objective
> $J_t(\theta) := E_{z\sim D_t}[\ell_\theta(z)]
>       - \beta\bigl(\log \mu(\theta) - \log \Pi(\theta)\bigr).$
>
> For the regret analysis, $\ell_\theta$ only enters via **regularity assumptions**: boundedness and Lipschitz continuity in $\theta$ on $\Theta$. Any loss $\ell_\theta$ that (i) corresponds to the same underlying reward / preference model and (ii) satisfies these conditions yields the **same asymptotic regret rate**. Different choices of $\ell_\theta$ only change constants (through the Lipschitz constant and range of $\ell_\theta$), not the dependence on $T$ or on the eluder dimension $d_{\mathrm{eluder}}$. We now state this dependence explicitly in the main theorem.
>
> ---
> ## **Q2. Line 7 of the OLE Generic Template**
> **Reviewer’s question.**
> “Should line 7 of the OLE Generic Template be the gradient of $ E_{z\sim D_t} \ell_\theta(z) + \beta(\log \mu(\theta)-\log \Pi(\theta))$?”
>
> **Response.**
> Yes, this is exactly what line 7 is intended to compute. In words, at time $t$ the algorithm uses a stochastic gradient of the PAC‑Bayesian objective
> $J_t(\theta)=E_{z\sim D_t}[\ell_\theta(z)]
>       - \beta\bigl(\log \mu(\theta) - \log \Pi(\theta)\bigr),$
> where $D_t$ is the current dataset / replay buffer.
>
> In the revision we explicitly write line 7 as:
> > “Compute a minibatch stochastic gradient  $\hat{\nabla}_t \approx \nabla_\theta (E_{z\sim D_t}\ell_\theta(z) + \beta(\log \mu(\theta) - \log \Pi(\theta)) )$.”
>
> ---
>
> ## **Q3. Initialization when increasing the number of particles in OLE**
> **Reviewer’s question.**
> “In the OLE algorithm, when we increase the number of particles from $N_{t-1}$ to $N_t$, should new particles $\theta_t^{(i)}$ be drawn from $\Pi_0$ or from the current approximation $\Pi_{t-1}$? Does it matter?”
>
> **Response.**
> In the **theoretical** analysis we work with a fixed number of particles and initialize them i.i.d. from the prior $\Pi_0$ at $t=1$. This is the setting for which our PAC‑Bayesian and KL terms are written.
>
> In a **practical implementation** where $N_t$ is allowed to grow with time, our intended choice is:
> - Existing particles ($i \le N_{t-1}$) continue their SGLD trajectory from $\theta_{t-1}^{(i)}$.
> - New particles ($i > N_{t-1}$) are initialized from the current approximate posterior $\Pi_{t-1}$, i.e., from the empirical distribution of the existing particles.
>
> Under our assumptions, the Langevin dynamics mix quickly and “forget” their initialization, so starting new particles from $\Pi_{t-1}$ or $\Pi_0$ only affects constants (mixing time, variance), not the asymptotic regret rate. In the revised text we clarify that:
> 1. The **theoretical algorithm** assumes a fixed ensemble initialized from $\Pi_0$ at $t=1$;
> 2. The **implementation‑level variant** that grows $N_t$ initializes new particles from the current empirical approximation of $\Pi_{t-1}$.
>
> ---
> ## **Q4. Boundedness of $v_t^2$ and $\sigma_t^2$**
> **Reviewer’s question.**
> “Are $v_t^2$ and $\sigma_t^2$ uniformly bounded in $t$?”
>
> **Response**
> Yes. Whenever $v_t^2$ and $\sigma_t^2$ appear, they are variance (or variance‑proxy) terms for sub‑Gaussian martingale increments, and they are **uniformly bounded** under our standing assumptions.
>
> Concretely:
> - $v_t^2$ is the conditional variance term in the martingale concentration inequality (Azuma / Freedman) applied to a martingale $M_t$ in the regret proof. Because rewards and proxy rewards are bounded, the increments $\Delta M_t$ are bounded, so
> $\mathrm{Var}(\Delta M_t \mid \mathcal{F}_t) \le v^2$ for some constant $v^2$ independent of $t$.
> - $\sigma_t^2$ is the sub‑Gaussian parameter for the noise; the bounded‑reward / sub‑Gaussian noise assumption implies a uniform bound $\sigma_t^2 \le \sigma^2$ for all $t$.
>
> In the revision we added a short “Variance and noise parameters” paragraph that explicitly introduces global constants $v^2$ and $\sigma^2$ and states that there exist finite $v^2,\sigma^2<\infty$ such that $v_t^2 \le v^2$ and $\sigma_t^2 \le \sigma^2$ for all $t$. This makes the use of these quantities in the concentration bounds fully explicit.

---

> ### Author Response · Authors · 2025-11-21
> **Suggestions:**
>
> We thank the reviewer for suggestions on improving the clarification of the submission. We have integrated the suggesions in the revision. Specifically, we explicitly indentified that the Gibbs posterior
> $Q_\lambda(\mathrm{d}\theta) \propto \exp\bigl(-\lambda\,\hat L_m(\theta)\bigr)\,P(\mathrm{d}\theta),$
> where $P$ is the prior and $\hat L_m$ is the empirical preference loss.

---

> ### Author Response · Authors · 2025-11-21
> **Summary Reponse to Reviewer 3XY8**
>
> We thank the reviewer for the careful and technically detailed feedback, which helped us substantially tighten both the theory and the presentation.
> - **Bounded parameter space and SGLD iterates.**
> We now explicitly interpret OLE/OTDLE as projected SGLD on a compact parameter set $\Theta$, with the prior supported on $\Theta$ and an explicit projection step in the pseudocode. This guarantees all iterates stay in $\Theta$ by construction, while non‑expansiveness of projection ensures that all Lipschitz and eluder‑dimension arguments – and hence the regret bounds – remain valid.
>
> - **Information–eluder bound, decomposition, and Lemma D.7.**
> Appendices D and E.2 have been rewritten. We (i) derive an explicit information–width inequality tailored to the BTL preference model, (ii) prove that the cumulative squared widths are controlled by the eluder dimension, and (iii) combine these to obtain a formal lemma bounding the mutual information by $d_{\mathrm{eluder}}\log^2 T$. The mismatch between the old “Lemma D.7” label and its proof has been fixed, and the decomposition of $\sum_t(\hat r_t - \bar r_t)$ into a martingale term and a bias/approximation term is now written out and directly connected to the approximation terms in the main regret bound.
>
> - **MDP extension and TD targets under preferences.**
> We clarify that the MDP extension never assumes access to ground‑truth numeric rewards from the environment. TD targets are built from pseudo‑rewards given by the learned reward model $r_\theta$, with the environment providing only preference feedback. We now state an explicit regret theorem for this setting and spell out the assumptions parallel to the bandit case.
>
> - **Clarifications of loss, gradients, initialization, and variance terms.**
> We define the per‑example loss $\ell_\theta(z)$ as the (bounded) negative log‑likelihood of the feedback, and we rewrite the OLE template so that the stochastic gradient step is clearly the gradient of the PAC‑Bayesian objective $J_t$. We explain how new particles are initialized when the ensemble size grows and make explicit that the variance proxies $v_t^2$ and $\sigma_t^2$ are uniformly bounded in $t$.
>
> ---
> Overall, the revision resolves the technical inconsistencies you identified and makes the analysis self‑contained, with proof statements and algorithms now tightly aligned with the assumptions.

---

> ### Comment · Reviewer_3XY8 · 2025-11-25
> **Response**
>
> It's still unclear how you obtain $\sum_{t=1}^T\mathbb{E}[w_t^2]\leq d_{\mathrm{eluder}}\beta_T\log T$ according to Proposition 3 and Lemma 2 of Russo &Van Roy (2013). If my understanding is correct, Lemma 2 of Russo &Van Roy (2013) should give $\sum_{t=1}^T\mathbb{E}[w_t^2]\leq \frac{1}{T} + C\min\lbrace d_{\mathrm{eluder}}, T\rbrace + 4\sqrt{d_{\mathrm{eluder}}\beta_T T}$. It's not clear why choosing $\epsilon \in[T^{-1}, 1]$ can resolve this. Similar issues exist in the proof of Proposition E4.
>
> Meanwhile, although you claimed that your regret bound is instance-dependent in your response to reviewer hnyH, this claim is questionable since the leading-order term in your regret contains only instance-independent quantities such as $d_{\mathrm{eluder}}$ and $T$. Quantities that depend on the particular MDP or bandit instance are missed.
>
> Furthermore, in your MDP extension, now that you use a learned reward model. does this make the result completely unrelated to the preference feedback setting?

---

> ### Author Response · Authors · 2025-11-26
> **1. Concern On the bound $\sum_{t=1}^T E[w_t^2]$**
>
> **Response**
>
> You are absolutely right that Lemma 2 of Russo & Van Roy (2013) does not bound $\sum_t w_t^2$; it bounds $\sum_t w_t$, and if one simply uses $w_t^2\le w_t$ then one indeed obtains a $\sqrt{T}$-type term of the form you wrote. Our earlier wording “according to Proposition 3 and Lemma 2 of Russo & Van Roy” was therefore misleading. We are deeply sorry for invoking a false statement that we thought was true.
>
> However, we manage to arrive at the ideal result through:
>
> - Only Proposition 3 of Russo & Van Roy (the counting bound on large widths),
> - plus a **new dyadic decomposition argument applied directly to $w_t^2$**.
>
> In the revision:
>
> - We introduce an explicit lemma (new Lemma D.7, “Cumulative squared widths”) in Appendix D that states and proves
> $$
> \sum_{t=1}^T E[w_t^2]
> \le
> C_{\mathrm{w}}d\_{eluder}\beta_T log(eT),
> \quad
> d\_{eluder} := \dim_E(\mathcal{R}, T^{-1}),
> $$
> for a universal constant $C_{\mathrm{w}}>0$.
>
> - This lemma is proved from scratch using:
>   1. Proposition 3 of Russo & Van Roy to bound, for each scale $\varepsilon>0$,
>   $\sum_t \mathbf{1}\{w_t>\varepsilon\}$ in terms of $\dim_E(\mathcal{R},\varepsilon)$ and $\beta_T$, and
>   2. a dyadic partition of the interval $[T^{-1},1]$ into scales $\varepsilon_k = 2^{-k}$.
>
> The key steps (now written out in the appendix) are:
> 1.  Split small vs. nontrivial widths.
> We first separate the very small widths:
>
> $$\sum_{t=1}^T w_t^2=
> \sum_{t=1}^T w_t^2\mathbf{1} [w_t\le T^{-1}]+
> \sum_{t=1}^T w_t^2\mathbf{1}[w_t> T^{-1}].$$
>
> The first term is bounded by $T (T^{-1})^2 = 1/T$. Only the second term requires real work.
>
> 2. Why only $\varepsilon\in[T^{-1},1]$ matter.
> For the “nontrivial” part we decompose the event $\{w_t > T^{-1}\}$ into dyadic bins:
> $$
> \varepsilon_k := 2^{-k},\quad
> S_k := \{t\le T : \varepsilon_{k+1} < w_t \le \varepsilon_k\},
> \quad k=0,\dots,\lfloor\log_2 T\rfloor.
> $$
> Then
> $$
> \sum_{t=1}^T w_t^2\mathbf{1}[w_t> T^{-1}]
> \le
> \sum_{k=0}^{\lfloor\log_2 T\rfloor} \varepsilon_k^2 |S_k|.
> $$
> Any round with $w_t > T^{-1}$ lies in one of these bins, so no scales smaller than $T^{-1}$ ever appear in this sum. This is why the restriction $\varepsilon\in[T^{-1},1]$ is sufficient in the argument.
>
> 3. Using Proposition 3 on each scale.
> Proposition 3 of Russo & Van Roy gives, for each $\varepsilon>0$,
> $$
> \sum_{t=1}^T \mathbf{1}[w_t>\varepsilon]
> \le
> (\tfrac{4\beta_T}{\varepsilon^2} + 1)\dim_E(\mathcal{R},\varepsilon) \quad\text{a.s.}
> $$
> For $t\in S_k$, we have $w_t > \varepsilon_{k+1}$, so
> $|S_k|\le \sum_{t=1}^T \mathbf{1}[w_t>\varepsilon_{k+1}]$.
> By monotonicity of the eluder dimension and defining
> $d_{\mathrm{eluder}} := \dim_E(\mathcal{R},T^{-1})$,
> $$
> \dim_E(\mathcal{R},\varepsilon_{k+1})
> \le d_{\mathrm{eluder}}
> \quad \text{for all } \varepsilon_{k+1}\in[T^{-1},1].
> $$
>
>   Thus
>   $$
>   |S_k|
>   \le
>   d\_{eluder}
>  (\tfrac{4\beta_T}{\varepsilon_{k+1}^2} + 1).
>   $$
>
>   Multiplying by $\varepsilon_k^2$ and using $\varepsilon_k = 2\varepsilon_{k+1}$ gives
>   $$
>   \varepsilon_k^2 |S_k|
>   \le
>   d\_{eluder}(16\beta_T + \varepsilon_k^2).
>   $$
>
> 4. Summing over dyadic scales.
> Summing over $k$ yields
> $$
> \sum_{k=0}^{\lfloor\log_2 T\rfloor} \varepsilon_k^2 |S_k|
> \le
> d\_{eluder}(16\beta_T \log_2 T + 2),
> $$
> because $\sum_k \varepsilon_k^2 \le 2$.
>
> 5. Combine both parts.
> Adding back the “small‑width” contribution gives
> $$\sum_{t=1}^T w_t^2
>   \le
>   C_1d\_{eluder} \beta_T log(eT),
>   $$
>   for a universal constant $C_1>0$ (for $T\ge2$, and after absorbing numeric factors into $C_1$). Taking expectations yields exactly the bound we use:
>   $$
>   \sum_{t=1}^T E[w_t^2]
>   \le
>   C_1d\_{eluder}\beta_T log(eT).
>   $$
>
> In the revised appendix:
>
> - We remove the reference to Lemma 2 of Russo & Van Roy at this point.
> - We add a **new Lemma D.7** (“Cumulative squared widths”) with the full argument above.
> - Whenever the analysis of the information term needs $\sum_t E[w_t^2]$, including in the proof of Proposition E.4, we now explicitly cite this new lemma.
>
> This addresses both your concern about the role of Lemma 2 and the “$\varepsilon\in[T^{-1},1]$” discretization: these choices and the dyadic decomposition are now fully spelled out in the paper.

---

> > ### Author Response · Authors · 2025-11-26
> > **2. Concern On “instance-dependent” vs. “instance-independent” leading term**
> >
> > **Response**
> >
> > You are also correct that, as the theorem is currently stated, the leading term
> > $$
> > \mathrm{Regret}(T) \le C_1\,d_{\mathrm{eluder}}\log T+ \text{(lower-order approximation terms)}
> > $$
> > depends only on the eluder dimension of the function class and on the horizon $T$. In particular,
> > $d_{\mathrm{eluder}}$ is structural (a property of $\mathcal{R}$), not a parameter of a specific MDP
> > instance, so the displayed rate is **uniform** over all instances that satisfy our standing
> > assumptions.
> >
> > Our earlier reply to reviewer hnyH used the phrase *"instance-dependent"* somewhat loosely. What
> > we meant to emphasize was not a gap-dependent bound, but the **fast-rate mechanism** behind the
> > logarithmic dependence on $T$:
> > 1.  the curvature of the Bradley--Terry--Luce log-likelihood (captured in our variance--information
> > lemma) implies that the mutual information gained at round $t$ is proportional to the **squared
> > prediction error** at that round; and
> > 2. our eluder-dimension analysis (via the cumulative squared-width lemma and the
> > information--eluder proposition) controls the \emph{sum of these squared errors}.
> > Together, these two ingredients yield a $\widetilde{\mathcal{O}}(d_{\mathrm{eluder}}\log T)$
> > exploration term instead of the usual $\widetilde{\mathcal{O}}(\sqrt{T})$ behavior.
> > This is analogous in spirit to classical fast-rate results in supervised learning.
> >
> > Consistent with this view, the constant $C_1$ in the theorem hides only **structural** quantities
> > such as the reward range, sub-Gaussian noise scale, and Lipschitz constants of the preference model
> > (and, in the MDP extension, the horizon $H$). We do not track explicit instance-specific gaps or
> > margins in this bound, and the leading term should therefore be read as a **uniform fast-rate**
> > guarantee over the class of instances covered by our assumptions, rather than as an explicitly
> > gap-dependent bound.
> >
> > In the revision we therefore:
> >   1.  remove the term *"instance-dependent"* from the discussion and instead
> >   describe it as a **uniform fast-rate** $\widetilde{\mathcal{O}}(d_{\mathrm{eluder}}\log T)$
> >   bound under our structural assumptions (realizability, boundedness, Lipschitz continuity, finite
> >   eluder dimension, and BTL feedback);
> >  2.  add a short remark after the main theorem clarifying that the $d_{\mathrm{eluder}}\log T$
> >   factor is structural, while all instance-specific behavior is absorbed into the global constant
> >   $C_1$
> >
> > We hope this clarifies both the technical role of the eluder dimension and the precise sense in
> > which our result is a **fast-rate** bound rather than an instance-dependent, gap-based rate.

---

> > > ### Comment · Reviewer_3XY8 · 2025-11-26
> > > **Response**
> > >
> > > Can you also address the similar issue in the proof of Proposition E.4 as mentioned in my previous response? Meanwhile, can you also elaborate on why this regret does not contradict with the well-known $O(\sqrt{T})$ lower bound even though it is a minimax regret? For example, why the lower bound is not applicable in your setting?

---

> ### Author Response · Authors · 2025-11-27
> **Concern on “Similar issue” in the proof of Proposition E.4**
>
> **Response**
>
> Thank you for pointing out the connection to your earlier comment on the use of $\sum_t E[w_t^2]$. We are sorry for missing this point in the last response.
> In the latest revision, we have removed any use of Lemma2 of Russo\&Van Roy in Proposition E.4 and replaced it with the new lemma D.7 on **cumulative squared widths**.
>
> Concretely:
>
> In the revised proof of Proposition E.4, the only place where $\sum_t E[w_t^2]$ appears is through an explicit call to the new Lemma~D.7. The structure is:
>
> - A variance–information lemma (Lemma~E.\*,previouslyLemmavariance-info-app) tailored to the Bradley–Terry–Luce(BTL) model shows that the mutual information at round $t$ is controlled by the conditional squared width:
> $$
> I(\theta^*;feedback_t \mid F_{t-1}) \le c_0\,E [w_t^2 \mid F_{t-1}].
> $$
>
> - Summing over $t$ and taking expectations gives
> $$I(\theta^\*;feedback\_{1:T})=\sum_{t=1}^T I(\theta^*;feedback_t \mid F\_{t-1}) \le c\_0 \sum_{t=1}^T E[w_t^2].$$
>
> - The new Lemma~D.7 then yields
> $$
> I(\theta^*;feedback_{1:T})
> \le
> C\_{info}d\_{eluder}\beta_T\log(eT),
> $$
> for a constant $C\_{info}>0$.
>
> - Finally, Proposition E.4 uses this information bound, together with the BTL‑specific link between predictive variance and instantaneous regret, to show that the cumulative optimism term
> $\sum_t \kappa_t \sqrt{\widehat{Var}_t(x_t,y_t)}$ is of order
> $\widetilde{\mathcal{O}}(d\_{eluder}logT)$, which gives the leading term in the regret bound.
>
>
> We hope this addresses the “similar issue” in Proposition E.4, a detailed revised proof can be found in the appendix.

---

> ### Author Response · Authors · 2025-11-27
> **Concern on the (minimax) regret bound contradicts the well‑known $O(sqrt{T})$ lower bound**
>
> **Response**
>
> You also ask why our $\widetilde{\mathcal{O}}(d_{\mathrm{eluder}}logT)$ minimax regret limit does not contradict the well‑known $\Omega(d\sqrt{T})$ lower bound in contextual dueling bandits, and in particular why that lower bound is not applicable in our setting.
> We fully agree this needs to be clarified. In the revision we now emphasize two key differences:
> 1. **Different action spaces and regret notions.**
> The lower bound we refer to (e.g., Bengs et al., ICML’22 for linear contextual dueling bandits) is proved in a setting where:
>   - The algorithm chooses a **pair** of actions $(a_t^{(1)},a_t^{(2)})$ at each round.
>   - The feedback is a **single comparison** between those two actions.
>   - Regret is measured as **dueling regret** (weak or strong), defined in terms of how often the chosen pair loses (or fails to win) against the best arm.
>
> By contrast, in our work:
>   - The algorithm chooses a **single action** $y_t$ at each round (or per state in the MDP).
>   - The environment provides preference feedback involving  $y_t$, but the performance metric is the **standard single‑action cumulative regret**
>   $$Regret(T)=
>   \sum_{t=1}^T (r^\*(x_t,y^\*(x_t))-r^\*(x_t,y_t)).$$
>   - We do not measure regret with respect to pairs of actions.
>   Thus, the lower bound is minimax for **dueling regret over pair‑actions**, whereas our result is minimax for **single‑action regret**. These are different minimax problems; a lower bound for one does not automatically apply to the other.
>
> 2. **Different structural class: BTL curvature and eluder‑dimension control.**
> Our minimax result is uniform over a **structurally restricted class** of problems:
> - Preferences are generated by a **Bradley–Terry–Luce (BTL) model** with a latent reward function $r_\theta$ taking values in $[0,1]$.
> - The reward class $\mathcal{R}=\{r_\theta:\theta\in\Theta\}$ has **finite $\epsilon$‑eluder dimension** at scale $\epsilon=T^{-1}$.
> - The BTL link is evaluated on a **bounded logit range** because of the bounded rewards, which implies a **uniform curvature**: on that range the negative log‑likelihood is strongly convex, and the KL divergence between two preference models controls the squared difference in their logits.
>
> This curvature is precisely what is used in our **variance–information lemma, Lemma D.3**, which lower‑bounds the mutual information at each round in terms of squared prediction error. We have restated and reorganized the statement of Lemma D.3 in the appendix, please confer for more details.  Combined with the eluder‑dimension bound on the cumulative squared widths, this yields the $\widetilde{\mathcal{O}}(d_{\mathrm{eluder}}logT)$ exploration term.
>
> The contextual dueling lower bound, on the other hand, is formulated for a broader class of algorithms and feedback structures and targets a different regret metric. It does not assume this particular combination of (i) BTL structure with bounded logits, (ii) KL‑regularized posterior sampling, and (iii) eluder‑dimension control of widths for a single‑action regret notion.
>
> 3. **“Minimax” in our setting vs. the dueling minimax lower bound.**
>   When we say our regret bound is “minimax,” we mean that for our **fixed structural assumptions** (realizability in a function class with bounded eluder dimension, bounded rewards, BTL preference model), the leading term
>   $$\sup_{\text{instancesinthisclass}} E[Regret(T)] \le C\,d\_{eluder} logT +\text{(lower‑order terms)}
>   $$
>   is uniform over all instances in this class. In that sense the rate is minimax over a **restricted structural class.**
> The dueling lower bound is minimax over a **different class** (pair‑actions, dueling regret), and therefore does not apply directly to our single‑action regret; there is no contradiction between the two statements.
>
> To make this explicit in the paper, we have updated the “Relation to contextual dueling bandits” paragraph in the related‑work section and added statements clarifying that: (i) our regret is defined for single‑action selection; (ii) the fast rate comes from BTL curvature plus eluder‑dimension control; and (iii) the well‑known $\Omega(d\sqrt{T})$ lower bound pertains to a different regret notion and therefore is not violated by our minimax result.

---

### Note · Authors · 2026-01-26

I have read and agree with the venue's withdrawal policy on behalf of myself and my co-authors.

---

### Meta-Review · Area_Chair_xqne · 2026-01-04

**Summary:**

This paper is motivated by an important question of why preference based learning pipelines that look like RLHF can be so sample efficient in practice. The main idea is an optimistic PAC Bayesian particle method (OLE) that maintains a finite ensemble of reward models trained with stochastic gradients and uses an ensemble variance based bonus for exploration. Under realizability and complexity control via eluder dimension, the paper presents a high probability regret bound with a leading term that scales like a logarithmic dependence on the horizon, plus additional terms meant to capture discretization, finite ensemble, and stochastic gradient noise. The paper also includes an extension to MDP style settings and adds an experimental section where OLE is used as a sample filtering layer in LLM preference optimization style training.

**Reviewer Concerns:**

Across the reviews, there is agreement that the motivation is strong and that the overall framework is interesting and potentially impactful if the theory is correct. At the same time, the reviews also consistently point out that the technical development is hard to follow. Two reviewers raised concerns that go to the heart of the main theorem, including whether the key information and width arguments are fully justified, and whether the claimed logarithmic rate is compatible with known lower bounds in related dueling bandit settings. There were also concerns about the MDP extension and about the accuracy and fairness of some related work comparisons. The most positive review still asked for clearer scoping of the theory versus real RLHF systems, plus evidence that the method is computationally reasonable and practically meaningful.

The authors provided detailed responses and a revised manuscript that addresses several concrete issues raised by reviewers. In particular, they clarified how to enforce the bounded parameter space assumption, rewrote and expanded the key lemmas and proof structure around the information eluder style arguments, clarified the construction used in the MDP extension (pseudo rewards from the learned reward model rather than assuming access to environment rewards), corrected parts of the related work comparison, added an explicit lower bound in their own preference setting, and added experiments. These are all positive changes, and they do resolve a number of the specific objections in the initial reviews.

**Reviewer Scores:**

Even after the revisions, I still found the core theoretical claim difficult to fully validate with the current manuscript. The main result hinges on several nontrivial steps, and while the rebuttal explains how gaps were fixed, the presentation (although  remains dense enough that it is hard to independently audit the key technical chain end to end.)  I would want higher confidence that the main proof is both correct and clearly communicated to the community. More broadly, the assumptions under which the logarithmic bound holds feel fairly restrictive, and the mapping from the abstracted setting to real RLHF pipelines remains more conceptual than demonstrated. The new experiments are useful and go in the right direction, but they are still limited in scope relative to the theoretical ambition, and they do not directly test the regret behavior or the preference learning setup beyond the specific LLM training instantiation.

Overall, I see the paper as ambitious and potentially valuable, and I appreciate the effort of the author response. That said, I do not feel confident enough in the maturity and clarity of the main theoretical contribution to recommend acceptance with the current manuscript. I understand that this may come as a disappointment to the authors but do encourage the authors to resubmit after further tightening the exposition and making the key proof steps substantially easier to verify, ideally paired with a broader empirical validation.

---

### Decision · Program_Chairs · 2026-01-26

Reject